# Clapping: Removing Per-sample Storage for Pipeline Parallel Distributed Optimization with Communication Compression

## Abstract

Pipeline-parallel distributed optimization is essential for large-scale machine learning but is challenged by significant communication overhead from transmitting high-dimensional activations and gradients between workers. Existing approaches often depend on impractical unbiased gradient assumptions or incur sample-size memory overhead. This paper introduces **Clapping**, a **C**ommunication compression algorithm with **LA**zy sam**P**ling for **P**ipeline-parallel learn**ING**. Clapping adopts a lazy sampling strategy that reuses data samples across steps, breaking sample-wise memory barrier and supporting convergence in few-epoch or online training regimes. Clapping comprises two variants including Clapping-**FC** and Clapping-**FU**, both of which achieve convergence without unbiased gradient assumption, effectively addressing compression error propagation in multi-worker settings. Numerical experiments validate the performance of Clapping across different learning tasks.

## 1 Introduction

Large-scale optimization and learning have become essential tools in numerous applications. Addressing these complex and large problems presents a substantial challenge, frequently necessitating extensive computation over many days or even months. Consequently, distributed algorithms are crucial for accelerating large-scale optimization and learning processes. In distributed optimization, multiple workers collaborate to solve a global problem with the help of communication between workers. Most existing research focuses on data-parallel distributed optimization (Li et al., 2014; Nedic & Ozdaglar, 2009; Chen & Sayed, 2012; Yuan et al., 2016; Konevcnỳ et al., 2015; Alistarh et al., 2017). In this paradigm, each worker maintains a complete replica of the model, independently samples training data, and exchanges models or gradients at each iteration, thereby achieving significant acceleration of large-scale optimization and learning tasks.

As model parameters in contemporary optimization and learning problems have grown to hundreds of billions (Radford et al., 2019; Brown et al., 2020; Koroteev, 2021; Rae et al., 2021; Zhang et al., 2022; Smith et al., 2022; Liu et al., 2024), these models have exceeded single-worker memory capacity, necessitating model-parallel distributed optimization. This paradigm partitions the model across multiple workers, with each worker managing only a subset of parameters, thereby enabling the training of massive models that would be intractable on a single worker. Model-parallel distributed optimization is particularly prevalent in the pre-training and fine-tuning of Large Language Models (LLMs) (Touvron et al., 2023; Meta, 2024). For instance, consider an LLM architecture comprising 24 transformer layers that exceeds the memory capacity of a single GPU cluster. The model can be efficiently segmented, with the initial 12 layers allocated to one GPU cluster and the remaining layers to another. This strategic partitioning ensures each GPU maintains only a fraction of the model's parameters, substantially reducing per-device memory requirements. Another prominent application of model-parallel distributed optimization is split learning (Thapa et al., 2022; Lin et al., 2024; Wang et al., 2022a), where the entire machine learning model is divided into smaller network segments and trained independently across multiple edge computing devices.

**Problem statement.** This paper examines pipeline-parallel distributed optimization, a specific form of model-parallel optimization that partitions model parameters in a pipeline-like manner (Narayanan

Figure 1: Illustration of the pipeline-parallel distributed optimization with communication compression.

et al., 2019; Huang et al., 2019; Narayanan et al., 2021; Ryabinin et al., 2023; Wan et al., 2025). Consider a computing cluster consisting of $E \geq 2$ workers. Model parameter $\boldsymbol{w}$ is partitioned into $E$ parts, denoted as $\boldsymbol{w} = (w_1, \cdots, w_E) \in \mathbb{R}^{d_{w_1} + \cdots + d_{w_E}}$, where each block component $w_e \in \mathbb{R}^{d_{w_e}}$ is maintained by worker $e$. Pipeline-parallel optimization can be formulated as follows:

$$\min_{\boldsymbol{w} \in \mathbb{R}^d} \quad \mathbb{E}_{x \in \mathcal{D}}\Big[ L(x; \boldsymbol{w}) := y_E \Big], \tag{1a}$$

$$\text{s.t.} \quad y_e = a_e(y_{e-1}, w_e), \quad \forall e \in \{1, \cdots E\}. \tag{1b}$$

In the above problem, the random variable $x$ denotes the data sample following distribution $\mathcal{D}$. The loss $L(x, \boldsymbol{w})$ is a composite function consisting of $E$ operators $a_e(y_{e-1}, w_e) : \mathbb{R}^{d_{e-1}} \times \mathbb{R}^{d_{w_e}} \to \mathbb{R}^{d_e}$ with $d_E = 1$, where $y_e$, following the terminology in LLMs, refers to the activation. For initialization, we let $y_0 = x$. An illustration for problem (1) is shown in Fig. 1, where each worker participates in the computation in a pipeline fashion. In LLMs, each operator $a_e(y_{e-1}, w_e)$ corresponds to a transformer layer. In split learning, there are two operators: $a_1(y_0, w_1)$, which represents the client-side sub-network, and $a_2(y_1, w_2)$, which corresponds to the server-side sub-network.

Pipeline-parallel distributed algorithms are effective for large-scale problems with multiple devices but incur significant communication overhead from transmitting high-dimensional activation and gradient vectors over low-bandwidth networks (Diskin et al., 2021; Ryabinin et al., 2023; Wang et al., 2022b), as shown in Figure 2. This overhead is also exacerbated in split learning due to wireless communication (Thapa et al., 2022; Lin et al., 2024). To mitigate this, we study pipeline-parallel algorithms with **communication compression**, which transmit compressed activation and gradient vectors rather than original ones to reduce cost.

**Fundamental challenges.** While communication compression has been extensively studied for data-parallel distributed optimization (Alistarh et al., 2017; Stich et al., 2018; Vogels et al., 2019; Richtárik et al., 2021; Huang et al., 2022; Fatkhullin et al., 2024), it remains largely unexplored for pipeline-parallel distributed optimization. Compressing activations and gradients, as illustrated in Fig.1, presents two fundamental challenges. First, each stochastic gradient computed through forward-backward propagation requires $2E - 2$ rounds of compression ($E - 1$ in the forward pass and $E - 1$ in the backward pass), introducing substantial errors in gradient estimation that may lead to non-convergence. Second, the composite structure of activations (see Eq. (1b)) results in error propagation during pipeline communication, causing a non-separable entanglement of gradient information and compression errors (detailed in Sec.B.1). These challenges render most effective techniques from data-parallel optimization infeasible for pipeline-parallel optimization.

## 1.1 LIMITATIONS IN EXISTING WORKS

Several studies have emerged to address these challenges (Evans & Aamodt, 2021; Fu et al., 2020; Wang et al., 2022b), but they exhibit critical limitations.

**(L1) Impractical unbiased gradient assumption.** The unbiased gradient assumption is crucial for ensuring the convergence of optimization algorithms. In line with this, existing works (Evans & Aamodt, 2021; Fu et al., 2020) assume that unbiased activation compression leads to unbiased gradient errors, but this fails due to the composite and non-linear structure of the operator $a_e(y_{e-1}, w_e)$. For example, even if $\mathbb{E}[\tilde{y}_{e-1}] = y_{e-1}$, it is not generally true that $\mathbb{E}[a_e(\tilde{y}_{e-1}, w_e)] = a_e(y_{e-1}, w_e)$.

**(L2) Sample-size memory overhead.** Wang et al. (2022b) proposes AQ-SGD, which achieves convergence without requiring the unbiased gradient assumption. Unlike approaches in (Evans & Aamodt, 2021; Fu et al., 2020) that directly compress activations, AQ-SGD compresses the change in activations for identical training samples across epochs. However, AQ-SGD requires storing activations for **all training samples**, resulting in memory overhead proportional to the sample size.

**(L3) Multiple-epoch training requirement.** Another technique behind the convergence of AQ-SGD is error compensation (Seide et al., 2014; Stich et al., 2018; Richtárik et al., 2021), which can progressively mitigate activation compression errors. However, this approach requires extensive training epochs to eliminate errors, leading to non-convergence in few-epoch training regimes. This limitation is particularly significant for large-scale LLM training/fine-tuning, where practical workloads typically involve only a few epochs (Rae et al., 2021; Zhang et al., 2022; Meta, 2024; Song et al., 2022; Nakamoto et al., 2024; Song et al., 2024; Guo et al., 2024).

**(L4) Limited scalability beyond two-worker setup.** The convergence analysis in (Evans & Aamodt, 2021; Fu et al., 2020) is designed for two-worker configurations and does not extend to larger pipeline systems. While AQ-SGD offers convergence guarantees for multi-worker setups, it assumes **no error accumulation** in the multi-worker pipeline scenario, which is generally **NOT** true in practice.

## 1.2 Contributions

This paper develops novel algorithms to address the aforementioned limitations. Our contributions are as follows:

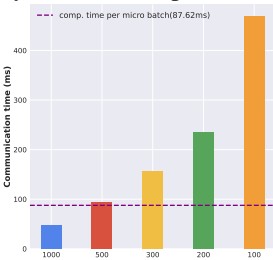

**(C1).** We propose `Clapping`, a **C**ommunication compression framework with **LA**zy sam**P**ling for **P**ipeline-parallel Learn**ING**. The `Clapping` framework is versatile and can be implemented in two variants: `Clapping`-**FC** and `Clapping`-**FU** , which differ in their approaches to the compression strategy in forward- and backward-propagation. A core technique underpinning `Clapping` is its novel lazy sampling strategy. By retaining the same data sample across multiple steps, this strategy enables `Clapping` to overcome the sample-size memory overhead associated with AQ-SGD and achieve convergence under few-epoch or even single-epoch training regimes, effectively addressing **Limitations (L2)** and **(L3)**.

Figure 2: Communication time for GPT2-xl with different bandwidth on 4 Nvidia A100 GPUs.

**(C2).** We demonstrate that `Clapping`-**FC** and `Clapping`-**FU** asymptotically achieves a **convergence rate** of $\mathcal{O}(1/\sqrt[3]{T})$ and $\mathcal{O}(1/\sqrt{T})$ respectively, where $T$ represents the number of algorithm iterations, without requiring **the unbiased gradient assumption** and **the reliance on multi-epoch training**. Furthermore, our analysis naturally extends to multi-worker setup with $E > 2$, explicitly accounting for the propagation and accumulation of compression errors in pipeline-parallel settings. These results directly address **Limitations (L1)** and **(L4)**. Our analysis also reveals key factors that influence the convergence rate, providing guidelines to boost performance.

**(C3).** We conduct extensive experiments across split learning, LLMs fine-tuning and pre-training. `Clapping` is comparable to AQ-SGD in several fine-tuning with far less memory cost and has a wider applicability than the other communication compression algorithms.

All theoretical results and existing algorithms for communication compression in pipeline-parallel distributed optimization are summarized in Table 1. Notably, `Clapping` achieves convergence without the unbiased gradient assumption, overcomes sample-size memory overhead, ensures convergence in few-epoch training regimes, and scales effectively to multi-worker pipeline scenarios.

**Notations.** We present the notations in this paper in Appendix A.

## 2 Prior Arts

Here we display some prior works of pipeline-parallel distributed optimization. More related works can be found in Appendix B.2.

### 2.1 Pipeline-parallel SGD

To solve pipeline-parallel optimization problem (1), we define $v_e := \nabla_{y_e} L(x; \boldsymbol{w}) \in \mathbb{R}^{d_e}$ as the gradient of the loss function with respect to the activation $y_e$, referred to as the activation gradient. Similarly, we define $u_e := \nabla_{w_e} L(x; \boldsymbol{w}) \in \mathbb{R}^{d_{w_e}}$ as the gradient with respect to the weight $w_e$, referred to as the weight gradient. Pipeline-parallel SGD performs the following steps at iteration $t$:

• **Forward**: Each worker $e$ computes activation $y_e^{(t)} = a_e(y_{e-1}^{(t)}, w_e^{(t)})$ following the forward order $e = 1, \cdots, E$. We let $y_0^{(t)}$ be the random sample $x^{(t)}$ for initialization.

Table 1: Comparison between different communication compression algorithms for pipeline-parallel optimization. Notation $T$ denotes the number of iterations, $x$ denotes the training sample, $N$ is the size of training data, $B$ is the batch size, $q$ denotes the dimension of the communicated parameters. We also list the result of Momentum SGD without compression in the bottom line for reference.

| Algorithms | C. Rate$^{\diamond}$ | No Unbiased G. Assum.$^{\dagger}$ | C. Mem.$^{\ddagger}$ | Few-epo. Rate$^{\triangleright}$ | Multi. Workers$^{\triangleright}$ |
|---|---|---|---|---|---|
| AC-GC (Evans & Aamodt, 2021) | $\frac{1}{\sqrt{T}}$ | ✗ | 0 | N.A. | ✗ |
| TinyScipt (Fu et al., 2020) | $\frac{1}{\sqrt{T}}$ | ✗ | 0 | N.A. | ✗ |
| AQ-SGD (Wang et al., 2022b) | $\frac{N}{\sqrt{T}}$ | ✔ | $\mathcal{O}(Nq)$ | ✗ | ☹ |
| **Clapping-FC (Ours)** | $\frac{1}{\sqrt[3]{T}}$ | ✔ | $\mathcal{O}(Bq)$ | ✔ | ✔ |
| **Clapping-FU (Ours)** | $\frac{1}{\sqrt{T}}$ | ✔ | $\mathcal{O}(Bq)$ | ✔ | ✔ |
| Momentum SGD (Nesterov, 2013) | $\frac{1}{\sqrt{T}}$ | ✔ | N.A. | ✔ | ✔ |

$^{\diamond}$ The convergence rate with respect to total steps $T \to \infty$ (smaller is better).

$^{\dagger}$ No additional assumptions regarding the unbiased gradients.

$^{\ddagger}$ Extra memory overhead incurred during a single communication compression operation. N.A. indicates no compression.

$^{\triangleright}$ The algorithm can achieve convergence in the few-epoch learning tasks, i.e., $N = \mathcal{O}(T)$.

$^{\triangleleft}$ Convergence analysis can be extended to the multiple-worker setup. ☹ indicates the analysis is with impractical assumptions.

- **Backward**: Each worker $e$ computes activation gradient $v_{e-1}^{(t)} = \nabla_1 a_e(y_{e-1}^{(t)}, w_e^{(t)})^\top v_e^{(t)}$ following the backward order $e = E, \cdots, 2$. We initialize $v_E^{(t)} = 1$.
- **Update**: Each worker $e$ computes weight gradient $u_e^{(t)} = \nabla_2 a_e(y_{e-1}^{(t)}, w_e^{(t)})^\top v_e^{(t)}$ following the backward order $e = E, \cdots, 1$. We update parameter $w_e^{(t+1)} = w_e^{(t)} - \gamma u_e^{(t)}$.

Pipeline-parallel SGD iteratively repeats the aforementioned steps until convergence (see Fig. 1). This requires transmitting all activations and their corresponding gradients between workers. To reduce communication overhead, we can apply direct compression of activations and gradients (Evans et al., 2020; Fu et al., 2020). However, such a compression may cause non-convergence due to biased gradient estimation and error propagation, even with unbiased compressors (see Appendix B.1).

## 2.2 AQ-SGD

To eliminate convergence bias in direct compression, Wang et al. (2022b) proposes AQ-SGD, an algorithm designed to ensure exact convergence to the stationary solution without relying on the impractical assumption of unbiased gradients. The key mechanism employed by AQ-SGD is error feedback (Richtárik et al., 2021). For a specific random sample $x$, we let $y_{x,e}$ represent the activation associated with that sample on each worker $e$, where $y_{x,0} = x$ serves as the initialization. Rather than compressing each activation $y_{x,e}$ directly, AQ-SGD compresses the changes in activations as follows:

$$y_{x,e}^{(t)} = a_e(\tilde{y}_{x,e-1}^{(t)}, w_e^{(t)}), \quad \tilde{y}_{x,e}^{(t)} = \tilde{y}_{x,e}^{(t-1)} + \mathcal{C}(y_{x,e}^{(t)} - \tilde{y}_{x,e}^{(t-1)}). \tag{2}$$

Suppose the compressor $\mathcal{C}(\cdot)$ is contractive, i.e., $\|\mathcal{C}(y) - y\| \leq \omega\|y\|$ for some $\omega \in (0, 1)$ (see Assumption 3 for details), it then holds that

$$\|\tilde{y}_{x,e}^{(t)} - y_{x,e}^{(t)}\| = \|\mathcal{C}(y_{x,e}^{(t)} - \tilde{y}_{x,e}^{(t-1)}) - (y_{x,e}^{(t)} - \tilde{y}_{x,e}^{(t-1)})\| \leq \omega\|\tilde{y}_{x,e}^{(t-1)} - y_{x,e}^{(t)}\|.$$

If we further assume that $y_{x,e}^{(t)} \to y_{x,e}^{\star} := a_e(y_{x,e-1}^{\star}, w_e^{\star})$, it follows that $\|\tilde{y}_{x,e}^{(t)} - y_{x,e}^{\star}\| \leq \omega\|\tilde{y}_{x,e}^{(t-1)} - y_{x,e}^{\star}\|$, implying that $\tilde{y}_{x,e}^{(t)}$ converges asymptotically to $y_{x,e}^{\star}$.

While error feedback progressively eliminates the compression bias in the activations, it requires repeatedly executing (2) with the same sample $x$ over multiple iterations. To enable the error feedback update, AQ-SGD passes all data samples across multiple epochs and stores each $\tilde{y}_{x,e}$ for every data sample $x$ and worker $e$. When a data sample $x$ is selected at iteration $t$, AQ-SGD updates $\tilde{y}_{x,e}^{(t)}$ according to (2); otherwise, it retains the previous value, ensuring $\tilde{y}_{x,e}^{(t)} = \tilde{y}_{x,e}^{(t-1)}$. Specifically, AQ-SGD operates as follows.

- **Forward**: Suppose sample $x$ is selected at the current iteration $t$. Each worker $e$ computes activation $y_{x,e}^{(t)}$ and compresses $\tilde{y}_{x,e}^{(t)}$ according to (2), following the forward order $e = 1, \cdots, E$. We let $y_0^{(t)} = x^{(t)}$. Finally, we update $\tilde{y}_{x',e}^{(t)} = \tilde{y}_{x',e}^{(t-1)}$ for any $x' \neq x$.

---

**Algorithm 1** LazySampling$(\mathcal{D}, t, p)$

---

**if** $t = 1$ **then**

 Sample data $x^{(1)}$ randomly from distribution $\mathcal{D}$.

**else**

 Retain $x^{(t)} = x^{(t-1)}$ with probability $1 - p$ and let $f_{\mathsf{FU}}^{(t)} = \mathtt{False}$.

 Sample $x^{(t)} \sim \mathcal{D}$ with probability $p$ and let $f_{\mathsf{FU}}^{(t)} = \mathtt{True}$.

**end if**

**Return:** $x^{(t)}, f_{\mathsf{FU}}^{(t)}$.

---

**Algorithm 2** Forward$_e(\tilde{y}_e^{(t-1)}, \tilde{y}_{e-1}^{(t)}, w_e^{(t)}, f_{\mathsf{FU}}^{(t)})$

---

In worker $e$: $y_e^{(t)} = a_e(\tilde{y}_{e-1}^{(t)}, w_e^{(t)})$,

**if** $\mathtt{Clapping}$-**FU and** $f_{\mathsf{FU}}^{(t)} = \mathtt{True}$ **then**

 Send $y_e^{(t)}$ from worker $e$ to $e + 1$,

 $\tilde{y}_e^{(t)} = y_e^{(t)}$.

**else**

 Send $\mathcal{C}(y_e^{(t)} - \tilde{y}_e^{(t-1)})$ from worker $e$ to $e + 1$,

 $\tilde{y}_e^{(t)} = \tilde{y}_e^{(t-1)} + \mathcal{C}(y_e^{(t)} - \tilde{y}_e^{(t-1)})$.

**end if**

---

**Algorithm 3** Backward$_e(\tilde{v}_{e-1}^{(t-1)}, \tilde{v}_e^{(t)}, w_e^{(t)}, f_{\mathsf{FU}}^{(t)})$

---

In worker $e$: $v_{e-1}^{(t)} = \nabla_1 a_e(\tilde{y}_{e-1}^{(t)}, w_e^{(t)})^\mathsf{T} \tilde{v}_e^{(t)}$,

**if** $\mathtt{Clapping}$-**FU and** $f_{\mathsf{FU}}^{(t)} = \mathtt{True}$ **then**

 Send $v_{e-1}^{(t)}$ from worker $e$ to $e - 1$,

 $\tilde{v}_{e-1}^{(t)} = v_{e-1}^{(t)}$.

**else**

 Send $\mathcal{C}(v_{e-1}^{(t)} - \tilde{v}_{e-1}^{(t-1)})$ from worker $e$ to $e - 1$,

 $\tilde{v}_{e-1}^{(t)} = \tilde{v}_{e-1}^{(t-1)} + \mathcal{C}(v_{e-1}^{(t)} - \tilde{v}_{e-1}^{(t-1)})$.

**end if**

---

- **Backward**: Each worker $e$ computes activation gradient $v_{e-1}^{(t)} = \nabla_1 a_e(\tilde{y}_{x,e-1}^{(t)}, w_e^{(t)})^\mathsf{T} \tilde{v}_e^{(t)}$ and compresses $\tilde{v}_{e-1}^{(t)} = \mathcal{C}(v_{e-1}^{(t)})$ following the order $e = E, \cdots, 2$.
- **Update**: Each worker $e$ computes weight gradient $u_e^{(t)} = \nabla_2 a_e(\tilde{y}_{x,e-1}^{(t)}, w_e^{(t)})^\mathsf{T} \tilde{v}_e^{(t)}$ following the order $e = E, \cdots, 1$. We update parameter $w_e^{(t+1)} = w_e^{(t)} - \gamma u_e^{(t)}$.

AQ-SGD operations are illustrated in Fig. 1, with error feedback applied only during the forward pass (implementation details in (Wang et al., 2022a, Algorithm 1)). Each worker $e$ must store activations $\tilde{y}_{x,e-1}$ for every data sample $x$, limiting the method to finite datasets and making it unsuitable for online learning with infinite data streams. This requires memory proportional to sample size, resulting in significant overhead and posing a substantial challenge in large-scale optimization, especially with extensive datasets and large models.

## 3 Clapping ALGORITHM

Here, we introduce $\mathtt{Clapping}$ , a novel approach designed to overcome the limitations of AQ-SGD.

**Lazy sampling.** As discussed in Sec. 2.2, error feedback is critical in communication compression for pipeline-parallel distributed optimization. The primary challenge lies in the fact that the update (2) must be repeated a sufficient number of times to progressively remove the compression bias. To facilitate error feedback, AQ-SGD stores $\tilde{y}_{x,e}$ for every data sample $x$ on each worker $e$. This design is the key reason for its limitations, which include a sample-size memory overhead and the inability to handle infinite datasets.

We propose a lazy sampling strategy to address this challenge. Unlike AQ-SGD, which stores $\tilde{y}_{e,x}$ and updates it only when data $x$ is re-sampled during multiple-epoch training, our approach employs a fixed sample $x$ for gradient evaluation across multiple consecutive steps, combined with error compensation. Once the compression bias is sufficiently reduced, we proceed to sample the next data point. As outlined in Algorithm 1, we retain the previous sample with probability $1 - p$; otherwise, a new sample is drawn from $\mathcal{D}$. This strategy enables sample reuse across iterations without storing $y_{e,x}$, thereby eliminating sample-size memory overhead. Moreover, lazy sampling facilitates error feedback updates even in settings with infinitely many data samples.

**Error feedback.** $\mathtt{Clapping}$ employs error feedback in both forward and backward processes, as detailed in Algorithms 2 and 3. In contrast, AQ-SGD applies error feedback exclusively to the forward process. Furthermore, leveraging the lazy sampling strategy, our approach eliminates the need to maintain $\{\tilde{y}_{e,x}\}$ and $\{\tilde{v}_{e,x}\}$ for each sample $x \in \mathcal{D}$. As shown in Algorithms 2 and 3, we only maintain $\tilde{y}_e$ and $\tilde{v}_e$ across all samples and iterations.

---

**Algorithm 4** `Clapping`

---

**Require:** Initialize $\tilde{y}_e^{(0)} = 0, \tilde{v}_e^{(0)} = 0, \tilde{u}_e^{(0)} = 0$ for $e = 1, \cdots, E - 1$. Initialize dataset $\mathcal{D}$, learning rate $\gamma_t$, compressor $\mathcal{C}$, and lazy sampling rate $\{p_t\}_{t=1}^T$.

    **for** $t = 1, \cdots, T$ **do**

        $x^{(t)}, f_{\mathsf{FU}}^{(t)} = \mathsf{LazySampling}(\mathcal{D}, t, p_t)$, initialize $\tilde{y}_0^{(t)} = x^{(t)}$, and let $\tilde{v}_E^{(t)} = 1$.

        **for** $e = 1, 2, \cdots, E - 1$ **do**

            $\mathsf{Forward}_e(\tilde{y}_e^{(t-1)}, \tilde{y}_{e-1}^{(t)}, w_e^{(t)}, f_{\mathsf{FU}}^{(t)})$,

        **end for**

        **for** $e = E, E - 1, \cdots, 1$ **do**

            Update $\tilde{u}_e^{(t)}$ and $w_e^{(t+1)}$ by (3), and take $\mathsf{Backward}_e(\tilde{y}_e^{(t-1)}, \tilde{y}_{e-1}^{(t)}, w_e^{(t)}, f_{\mathsf{FU}}^{(t)})$ **if** $e \neq 1$.

        **end for**

    **end for**

---

**Incorporation of momentum.** Momentum is a widely used technique to accelerate SGD convergence, acting as a surrogate for large-batch gradients and reducing gradient variance. Recent studies underscore its theoretical benefits: Cheng et al. (2023) highlight its role in mitigating data heterogeneity, while Fatkhullin et al. (2024) demonstrate its effectiveness in enhancing error feedback. As algorithm 4 illustrates, we adopt the following momentum update to mitigate the gradient bias caused by inaccurate activation gradients:

$$\tilde{u}_e^{(t)} = (1 - m_t)\tilde{u}_e^{(t-1)} + m_t \nabla_2 a_e(\tilde{y}_{e-1}^{(t)}, w_e^{(t)})^\mathsf{T} \tilde{v}_e^{(t)}, \quad w_e^{(t+1)} = w_e^{(t)} - \gamma \tilde{u}_e^{(t)}. \tag{3}$$

where $m_t \in (0, 1)$ is the momentum coefficient.

**Clapping algorithm.** Combining lazy sampling, error feedback, and momentum updates, the complete `Clapping` framework is presented in Algorithm 4. While `Clapping` is formulated with momentum SGD and a batch size of 1, it can be extended to optimizers like Adam (Kingma, 2014) and AdamW (Loshchilov, 2017). We present `Clapping` with Adam optimizer as well as the extended theoretical convergence analysis in Appendix E, and we evaluate `Clapping` with Adam-based optimizers in Section 5. For large-batch scenarios, lazy sampling can be adapted batch-wise, simplifying implementation. The detailed lazy sampling strategy and algorithmic formulation for large batches are provided in Appendix F.1.

**Clapping-FC and Clapping-FU.** Clapping can be implemented in two variants according to whether the compression takes during communication when the data batch is firstly sampled. Specifically, `Clapping` with First step Compressed (**Clapping-FC**) takes the compression operation during the whole process of learning. Meanwhile, `Clapping` with First step Uncompressed (**Clapping-FU**) does not take compression when the data $x^{(t)}$ is randomly sampled from $\mathcal{D}$ so as to reduce the error introduced by sample variance, as shown in Algorithms 2 and 3. `Clapping`-**FU** deferred compression mechanism maintains competitive performance (e.g., achieving 60% higher communication improvement than `Clapping`-**FC** when $p_t = 0.4$). Meanwhile, `Clapping`-**FC** can also deliver superior accuracy in practical optimization tasks as evidenced in Section 5.

**Memory Overhead.** With the error feedback technique, `Clapping` caches the current batch's activations and gradients, resulting in $\mathcal{O}(B)$ memory overhead for batch size $B$. In contrast, (Wang et al., 2022b)'s sample-wise error compensation incurs $\mathcal{O}(N)$ memory requirement for sample size $N$. The memory cost reduction is significant: while Wang et al. (2022b) theoretically requires TBs for a single communication compression, `Clapping` only needs several GBs for pre-training models with billions of parameters like LLaMA-2 7B or LLaMA-3 8B, acceptable in practice (See Appendix H for details). This advantage is more pronounced with larger models and more extensive datasets.

## 4 THEORETICAL ANALYSIS

This section presents theoretical analysis for `Clapping`-**FC** and `Clapping`-**FU**.

### 4.1 ASSUMPTIONS

We first introduce assumptions used throughout this paper.

**Assumption 1.** *There exist constants $L_{\nabla\ell}, C_a, L_{\nabla a}, L_a$ such that:*

1. $\nabla\ell$ *is $L_{\nabla\ell}$-Lipschitz continuous;*

2. *For $e = 1, 2, \cdots, E$, the gradient of $a_e$ can be bounded by $C_a$, i.e. $\|\nabla a_e(y, \boldsymbol{w})\| \le C_a$;*

3. *For $e = 1, 2, \cdots, E-1$, $a_e(y, \boldsymbol{w}), \nabla a_{e+1}(y, \boldsymbol{w})$ are $L_a, L_{\nabla a}$ Lipschitz continuous with respect to $y$ and $\boldsymbol{w}$, respectively.*

We remark that Assumption 1 is weaker than the smoothness assumption used in Wang et al. (2022b).

**Assumption 2.** *The stochastic gradient $\nabla L(x; \boldsymbol{w})$ is an unbiased estimate of $\nabla\ell(\boldsymbol{w})$ with bounded variances $\sigma^2$.*

**Assumption 3.** *For the compressor $\mathcal{C}$, there exist constants $\omega_F, \omega_B \in [0, 1)$ such that:*

$$\mathbb{E}\left[\|x - \mathcal{C}(x)\|^2 \big| x\right] \le \begin{cases} \omega_F^2 \|x\|^2, \textit{forward propagation}, \\ \omega_B^2 \|x\|^2, \textit{backward propagation}. \end{cases}$$

Compressors satisfying the above assumption are referred to as contractive compressors. In general, more aggressive compression leads to greater information distortion, corresponding to a larger $\omega$. This assumption applies to numerous compressors, including top-$K$ and low-rank projection (Alistarh et al., 2017; 2018; Vogels et al., 2019; Beznosikov et al., 2023; Stich et al., 2018), and is widely adopted in communication-efficient algorithms (Koloskova et al., 2019; Richtárik et al., 2021; Fatkhullin et al., 2024; Wang et al., 2023a; Huang et al., 2022).

The assumption below is critical for lazy sampling:

**Assumption 4.** *For each $x_1, x_2 \sim \mathcal{D}$, there exists $\varphi > 0$ such that $\mathbb{E}_{x_1, x_2 \in \mathcal{D}}\left[\|x_1 - x_2\|^2\right] \le \varphi^2$.*

It is important to note that Assumption 4 is not overly restrictive for most optimization and learning tasks. Indeed, Assumption 4 holds for **all finite datasets**. Moreover, it is likely to be satisfied even for infinite datasets, particularly **when normalization techniques are applied**.

## 4.2 Clapping CONVERGENCE

Firstly, we present the convergence result of Clapping-**FC** is as follows.

**Lemma 1.** *Suppose $x^{(t)}$ is the sampled data at iteration $t$, if we let $p_2 = 1$ and $p_3 = \cdots = p_T = p$ as a constant, then for Clapping-**FC**, under Assumptions 1–3 the following holds.*

$$\frac{1}{T}\sum_{t=1}^{T}\mathbb{E}\left[\|\nabla\ell(\boldsymbol{w}^{(t)})\|^2\right] \lesssim_{T,p,m} \frac{1}{\gamma T} + \frac{1}{mT} + \frac{1}{T}\sum_{t=1}^{T}\mathbb{E}\left[\|x^{(t+1)} - x^{(t)}\|^2\right] + \sigma^2\frac{(2-p)m - (1-p)m^2}{1 - (1-p)(1-m)^2}$$

$$+ \frac{1}{T}\left(\frac{1}{m^2} - \frac{1}{\gamma^2}\right)\sum_{e=1}^{E}\sum_{t=1}^{T}\mathbb{E}\left[\|w_e^{(t+1)} - w_e^{(t)}\|^2\right]. \tag{4}$$

In inequality (4), the term $\sum_{t=1}^{T}\mathbb{E}[\|x^{(t+1)} - x^{(t)}\|^2]$ arises from error feedback across iterations. If different data samples are selected at iterations $t$ and $t + 1$, the term $y_e^{(t+1)} - \tilde{y}_e^{(t)}$ in Algorithm 2 introduces an error due to the discrepancy between $x^{(t+1)}$ and $x^{(t)}$. When $\sum_{t=1}^{T}\mathbb{E}[\|x^{(t+1)} - x^{(t)}\|^2] = \mathcal{O}(T)$, Algorithm 4 converges to an $\mathcal{O}(1)$ bias, as indicated by (4). This highlights the necessity to introduce lazy sampling to mitigate this term.

As outlined in Algorithm 1, lazy sampling ensures $x^{(t+1)} = x^{(t)}$ with probability $1 - p_t$, thus the term $\sum_{t=1}^{T}\mathbb{E}[\|x^{(t+1)} - x^{(t)}\|^2]$ can be reduced to $\mathcal{O}(Tp)$. With this, we can present the convergence of Clapping-**FC** as follow:

**Theorem 1.** *Suppose $x^{(t)}$ is the sampled data at iteration $t$, there exist properly chosen constant step sizes $\gamma$, a momentum coefficient $m$, and lazy sampling coefficient $p$ such that, for Clapping-**FC**, under Assumptions 1–4 the following holds.*

$$\frac{1}{T}\sum_{t=1}^{T}\mathbb{E}\left[\|\nabla\ell(\boldsymbol{w}^{(t)})\|^2\right] \lesssim \frac{\sigma^{\frac{4}{3}}}{T^{\frac{1}{3}}(1 - \omega_B)^{\frac{4(E-1)}{3}}(1 - \omega_F)^{\frac{4(E-1)}{3}}} + \frac{\delta}{T}, \tag{5}$$

*where $\delta$ is a constant only depends on $\omega_B, \omega_F, E$ as*

$$\delta \lesssim \frac{1}{(1 - \omega_F)^{E-1}(1 - \omega_B)^{E-1}} + \frac{\omega_F^2 + \omega_B}{(1 - \omega_F)^2(1 - \omega_B)^{2(E-1)}} + \frac{1}{(1 - \omega_F)^{2(E-2)-1}}. \tag{6}$$

Meanwhile, the convergence rate of Clapping-**FU** is as follows.

Table 2: The score (↑) of all the tasks for GLUE benchmark with communication compression algorithms.

| Algorithms | MNLI | SST-2 | MRPC | CoLA | QNLI | QQP | RTE | STS-B | Avg |
|---|---|---|---|---|---|---|---|---|---|
| No comp. | **90.01** | **96.41** | 88.25 | **64.69** | **94.59** | **92.03** | 82.35 | **92.52** | 87.61 |
| EF21 | 89.28 | 94.79 | 87.00 | 62.97 | 93.87 | 92.01 | 81.62 | 91.30 | 86.61 |
| Clapping-**FC** | 89.94 | 96.06 | **90.50** | 63.98 | 94.04 | 91.89 | **84.19** | 91.71 | **87.79** |

**Theorem 2.** *Suppose $x^{(t)}$ is the sampled data at iteration $t$, there exist properly chosen constant step sizes $\gamma$, a momentum coefficient $m$, and lazy sampling coefficient $p$ such that, for* Clapping-*$FU$, under Assumptions 1–3 the following holds.*

$$\frac{1}{T}\sum_{t=1}^{T} \mathbb{E}\left[\left\|\nabla\ell(\boldsymbol{w}^{(t)})\right\|^2\right] \lesssim \frac{\sigma}{\sqrt{T}} + \frac{1}{T(1-\omega_B)^{E-1}(1-\omega_F)^{E-1}}. \tag{7}$$

As illustrated in inequality (5) and (7), Clapping-**FC** and Clapping-**FU** can achieve an asymptotic convergence rate of $\mathcal{O}(1/\sqrt[3]{T})$ and $\mathcal{O}(1/\sqrt{T})$, respectively. We should note that this is the **FIRST** convergence result of communication compression algorithms for pipeline-parallel distributed optimization that holds in few-epoch learning without unbiased gradient assumption, which satisfies the need of modern large-scale optimization tasks.

**Trade-off in the selection of $p_t$.** A small $p_t$ in lazy sampling can guarantee the convergence and also reduce the communication overhead with Clapping-**FU**. Nevertheless, an excessively small $p_t$ may compromise the model's generalization ability, which occurs because some samples may be over-learned, while others are neglected. Meanwhile, the $\sigma^2$ term in inequality (4) equals to $\mathcal{O}(1)$ when $p \to 0$, causing a non-convergence of Clapping-**FC**. As a result, there is a trade-off in selecting $p_t$. In practice, a not-too-small $p_t$ like $0.5$ or $0.4$ can be beneficial to the experimental performance; see Sec. 5 for further details.

**Impact of compression error accumulation.** Evidently, more compression entails more accumulated error and results in slower convergence. It is also noteworthy that Clapping-**FU** requires $\mathcal{O}(1/\varepsilon^2 + 1/\varepsilon(1-\omega_B)^{E-1}(1-\omega_F)^{E-1})$ iterations to approach an $\varepsilon$-stationary point. Thus, the impact of compression in our proposed method can be asymptotically nullified since the term dominates the convergence rate, which aligns with the result in (Fatkhullin et al., 2024). However, the adverse influence of communication compression persists in the algorithms presented by (Wang et al., 2022b).

**Convergence with Adam.** We provide the convergence analysis for both Clapping-**FC** and Clapping-**FU** with the Adam optimizer in Appendix E.2. It can be shown that Clapping with Adam shares the same convergence rate as that with momentum SGD. Such a convergence guarantee illustrates that Clapping is suitable for LLM pre-training and fine-tuning tasks, which significantly extends the applicability of our proposed algorithm.

**Convergence in large batch scenario.** We provide the convergence analysis of Clapping with batch size $B > 1$ in Appendix F.2. It can be shown that Clapping-**FC** achieves a convergence rate of $\mathcal{O}(1/\sqrt[3]{BT})$ and Clapping-**FU** achieves an convergence rate of $\mathcal{O}(1/\sqrt{BT})$, with the impact of compression error remaining similar to the case when $B = 1$. Additionally, the selection of the lazy sampling coefficient can also extended to the large-batch scenario.

**Convergence and error propagation in multi-worker scenario.** Eq. (5) and (7) illustrate the convergence result of Algorithm 4 with multiple workers with the analysis of the error accumulation in the compression. And we also present a detailed analysis of error propagation in Appendix D. Such a result remedies the shortage of (Wang et al., 2022b) in multi-worker compression. See Appendix G for more detail. Furthermore, we provide an empirical analysis of error propagation across multiple workers; refer to Appendix I.3.2 for experimental details and results.

## 5 EXPERIMENTS

We present experiments to validate the performance of Clapping . **Unless otherwise specified, Clapping refers to Clapping-FC in this section**. Additional experimental details, extended results, and supplementary experiments are provided in Appendix I.

Table 3: The evaluation accuracy (↑) of different communication compression algorithms when fine-tuning different models by Wikitext with Top-5% compressor.

| Model | No comp. | Direct comp. | EF21 | AQ-SGD | Clapping-FC | | |
|-------|----------|--------------|------|--------|-------------|--|--|
| | | | | | $p = 0.3$ | $p = 0.4$ | $p = 0.5$ |
| LLaMA-2 7B | 0.5948 | 0.5473 | 0.5678 | 0.5696 | **0.5960** | 0.5920 | 0.5877 |
| LLaMA-3 8B | 0.5991 | 0.5671 | 0.5677 | 0.5683 | 0.5865 | **0.5969** | 0.5887 |

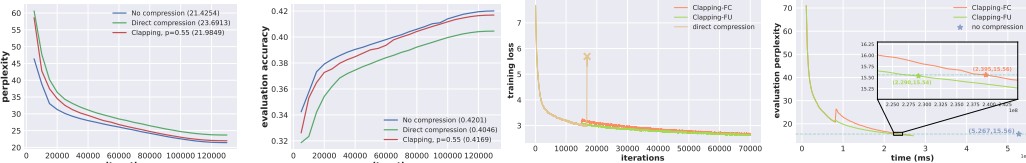

Figure 3: The evaluation perplexity (left) and accuracy (right) with various communication compression algorithms for pre-training GPT-2.

Figure 4: The training loss (left) and evaluation perplexity (right) with various communication compression algorithms for pre-training LLaMA-2 1B.

**Fine-tuning on GLUE benchmark.** We fine-tune pre-trained RoBERTa-large (Liu et al., 2019) on the GLUE benchmark (Wang et al., 2018) for 10 epochs using communication compression algorithms, including Clapping and direct compression with EF21 (Richtárik et al., 2021), and compare them with uncompressed fine-tuning on two NVIDIA A800 GPUs. A TopK compressor retains 30% of elements at the network midpoint. As shown in Table 2, Clapping outperforms EF21 in most tasks and achieves the highest average score.

**Fine-tuning on Wikitext.** We fine-tune pre-trained LLaMA-2-7B (Touvron et al., 2023) and LLaMA-3-8B (Grattafiori et al., 2024) models on the Wikitext-2 dataset (Merity et al., 2016). The model is split at the midpoint, and a Top-K compressor (Wangni et al., 2018) retains 5% of elements. Table 3 shows that Clapping outperforms other algorithms, including direct compression, EF21, and AQ-SGD. By tuning the low-rank coefficient $p$, **Clapping-FC achieves 95% communication saving with less than 0.5% error in practical fine-tuning tasks**.

**Pre-training GPT-2 model with multiple compression.** We pre-train a GPT-2 small model (Radford et al., 2019) on the OpenWebText dataset (Peterson et al., 2019) using natural compression with algorithms including uncompressed training, direct compression, and Clapping. We consider a harsh scenario that splits the 124M model to three parts and applies compression twice during forward and backward propagation. Figure 3 shows that Clapping mitigates communication errors in loss and perplexity, inducing approximately 1% decrease in accuracy to adapt to the compression scenario. AQ-SGD is inapplicable due to single-epoch training constraints.

**Pre-training LLaMA-2 1B model with end-to-end time.** We pre-train a LLaMA-2-1B model (Touvron et al., 2023) on the C4 dataset (Raffel et al., 2020) using a compressor combining TopK and quantization, comparing Clapping-**FU** and Clapping-**FC** under 100 MB/s bandwidth constraints. Following (Zhao et al., 2024)'s setup (see Appendix I.4.2 for more details), Figure 4 shows that both Clapping-**FU** and Clapping-**FC** achieve at least **2.2**× acceleration to reach the final perplexity reported in (Zhao et al., 2024), while direct compression fails to converge. Thus it demonstrates that Clapping can achieve the convergence without significant degradation of expressive capability. See Appendix I.4.2 for more discussion.

## 6 CONCLUSIONS

This paper proposes Clapping, a communication compression framework for pipeline-parallel distributed optimization. By introducing error feedback and lazy sampling techniques, both Clapping-**FC** and Clapping-**FU** achieve the state-of-the-art convergence rate compared to existing algorithms without the unbiased gradient assumption and sample-wise memory overhead, while Clapping-**FU** can achieve the $\mathcal{O}(1/\sqrt{T})$ convergence.

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

# Appendix

## A  NOTATIONS

In this paper, we define $\ell(\boldsymbol{w}) := \mathbb{E}_{x\sim\mathcal{D}}[L(x;\boldsymbol{w})]$ throughout this paper. We further denote its partial gradient with respect to the parameter $w_e$ as $\nabla_e\ell(\boldsymbol{w}) := \partial\ell(\boldsymbol{w})/\partial w_e \in \mathbb{R}^{d_{w_e}}$ for $e = 1, 2, \ldots, E$. For the operator $a_e(y_{e-1}, w_e) : \mathbb{R}^{d_{e-1}} \times \mathbb{R}^{d_{w_e}} \to \mathbb{R}^{d_e}$, we define $\nabla_1 a_e(y_{e-1}, w_e) \in \mathbb{R}^{d_e\times d_{e-1}}$ and $\nabla_2 a_e(y_{e-1}, w_e) \in \mathbb{R}^{d_e\times d_{w_e}}$ as the Jacobian matrix with respect to $y_{e-1}$ and $w_e$, respectively. The notation $\|\cdot\|$ represents the $\ell_2$-norm of vectors, and $\mathbf{1}_B \in \mathbb{R}^B$ denotes a vector with all elements equal to 1. For variables $\{y_e^{(t)}\}_{e=1,\ldots,E}^{t=0,\ldots,T+1}$, the subscript $e$ (resp. superscript $t$) denotes the index of the worker (resp. iteration). We use $a \lesssim b$ to indicate that there exists a constant $C \geq 0$ such that $a \leq Cb$, and $a \lesssim_d b$ indicates that there exists a $C \geq 0$ that is independent with $d$ such that $a \leq Cb$.

## B  MORE RELATED WORKS

### B.1  DIRECT COMPRESSION

To reduce communication overhead in pipeline-parallel optimization, we can directly compress the activations and their corresponding gradients, significantly reducing their size (Evans et al., 2020; Fu et al., 2020). Let $\mathcal{C}(\cdot)$ denote a compressor. The compressed pipeline-parallel SGD follows the same forward-backward procedure as described in Sec. 2.1, with several core operations slightly modified:

- **Forward**: Each worker $e$ computes activation $y_e^{(t)} = a_e(\tilde{y}_{e-1}^{(t)}, w_e^{(t)})$ and compresses $\tilde{y}_e^{(t)} = \mathcal{C}(y_e^{(t)})$ following the forward order $e = 1, \cdots, E$. We let $y_0^{(t)} = x^{(t)}$.

- **Backward**: Each worker $e$ computes activation gradient $v_{e-1}^{(t)} = \nabla_1 a_e(\tilde{y}_{e-1}^{(t)}, w_e^{(t)})^\top \tilde{v}_e^{(t)}$ and compresses $\tilde{v}_{e-1}^{(t)} = \mathcal{C}(v_{e-1}^{(t)})$ following the backward order $e = E, \cdots, 2$.

- **Update**: Each worker $e$ computes weight gradient $u_e^{(t)} = \nabla_2 a_e(\tilde{y}_{e-1}^{(t)}, w_e^{(t)})^\top \tilde{v}_e^{(t)}$ following the backward order $e = E, \cdots, 1$. We update parameter $w_e^{(t+1)} = w_e^{(t)} - \gamma u_e^{(t)}$.

Pipeline-parallel SGD with direct compression is also illustrated in Fig. 1. Compression errors introduce unique challenges to pipeline-parallel SGD, which differ significantly from those encountered in data-parallel SGD.

**Unbiased compressor leads to biased gradient.** Direct compression in pipeline-parallel optimization naturally results in biased gradient estimates. Suppose $\mathcal{C}(\cdot)$ is a unbiased compressor such that $\mathbb{E}[\tilde{y}_e] = y_e$, it cannot be guaranteed that $\mathbb{E}[\nabla_j a_e(\tilde{y}_{e-1}, w_e)] = \nabla_j a_e(y_{e-1}, w_e)$ $(j = 1, 2)$ due to the composite and non-linear structure of the activation operator $a_e(y_{e-1}, w_e)$, leading to biased activation gradient and weight gradient estimates. However, (Evans et al., 2020) and (Fu et al., 2020) impose the impractical assumption of an unbiased gradient to establish convergence guarantees. Without this assumption, direct compression cannot achieve convergence to stationary solutions.

**Error propagation.** Compression errors in pipeline-parallel SGD propagate in both the forward and backward processes. To illustrate this, we consider a linear neural network mapping $a_e(y_{e-1}, w_e) = W_e y_{e-1}$, where $W_e$ is the weight matrix reshaped from $w_e$. Let $\tilde{v}_e = v_e + \epsilon_e$, where $\epsilon_e$ represents the error incurred during the compression $\tilde{v}_e = \mathcal{C}(v_e)$. From the backward step in direct compression, it holds that

$$v_1 = W_1^\top \left( W_2^\top \left( \cdots \left( W_E^\top v_E + \epsilon_E \right) + \cdots \right) + \epsilon_2 \right) + \epsilon_1.$$

It is evident that the innermost error $\epsilon_E$ propagates through layers and can be significantly amplified by $W_1^\top\cdots W_{E-1}^\top$. This error propagation leads to a complex entanglement between the true gradient and the compressed one, severely impairing the performance and stability of the optimization.

## B.2 OTHER RELATED WORKS

**Communication compression in data parallelism.** Communication compression has demonstrated significant efficacy in data-parallel distributed optimization (Xu et al., 2020; Wang et al., 2023b), with two core strategies underpinning its success: *sparsification* and *quantization*. Classical sparsification methods include Top-K (Wangni et al., 2018; Alistarh et al., 2018) and Rand-K (Stich, 2018; Beznosikov et al., 2023), while quantization techniques encompass Sign-SGD (Seide et al., 2014; Bernstein et al., 2018), TurnGrad (Wen et al., 2017), and natural compression (Horvóth et al., 2022). However, compression inevitably introduces information distortion, which can hinder convergence rates or even lead to non-convergence. To mitigate these challenges, a variety of advanced techniques have been developed, including error feedback (Stich et al., 2018; Tang et al., 2019; Richtárik et al., 2021; Fatkhullin et al., 2024), hybrid compression (Wang et al., 2023a), and multiple-step compression (Huang et al., 2022; He et al., 2023). Furthermore, (Markov et al., 2023) introduced weight compression as a complementary approach. Despite these significant advancements, none of these results have been directly extended to pipeline-parallel distributed optimization.

**Activation compression.** A closely related technique is activation compression, which aims to reduce memory costs during LLM pre-training and fine-tuning (Jiang et al., 2018; 2022; Jin et al., 2021; Evans & Aamodt, 2021; Evans et al., 2020; Fu et al., 2020; Yu & Wu, 2023). In communication compression for pipeline-parallel optimization, activations must also be compressed to minimize communication overhead. The key distinction lies in the fact that, in memory-efficient settings, activations are computed precisely during forward propagation, with their compressed copies stored for backward propagation (Jiang et al., 2022), thereby introducing no additional errors during forward propagation. However, in pipeline-parallel communication compression, forward propagation is inherently error-prone, as compressed activations are immediately used for subsequent computations (Wang et al., 2022b), leading to error accumulation across both forward and backward propagation. This poses a significant challenge for algorithm design and theoretical analysis. Recently, He et al. (2025) also presented an effective activation compression algorithm in pipeline parallelism. While favorable compression performance is achieved on the LLMs fine-tuning tasks, it still lacks theoretical convergence guarantees.

## C PROOF OF THE CONVERGENCE RATE

In this section, we present the convergence analysis of Algorithm 4.

To begin with, we use different superscripts to represent variables with/without compression. Suppose the communication compression is taken in the $t$-th iteration, then we have:

- **Variables with a hat** (like $\hat{y}_e^{(t)}, \hat{v}_e^{(t)}$): The variable is obtained from standard momentum SGD algorithm **without any communication compression**.

- **Variables with a tilde** (like $\tilde{y}_e^{(t)}, \tilde{v}_e^{(t)}$): The variable which is used for the gradient evaluation during the $t$-th iteration. If the communication compression is taken during the $t$-th iteration, it denotes variabl obtained from Algorithm 1 - 4 and **has already been compressed**.

- **Variables without any additional superscript** (like $y_e^{(t)}, v_e^{(t)}$): The variable is obtained from Algorithm 1 - 4 and **has not been compressed** during the $t$-th iteration if the communication compression is taken.

If the communication compression is not taken in this iteration, same variables with different superscripts are all denote the variable without communication, such as $y_e^{(t)} = \hat{y}_e^{(t)} = \tilde{y}_e^{(t)}$.

Moreover, in the $t$-th iteration, we firstly denote $\mathcal{F}_e^{(t)}$ and $\mathcal{G}_e^{(t)}$ for $e = 1, 2, \cdots, E-1$ as follows:

- $\mathcal{F}_e^{(t)}$: the filtration before the communication from machine $e$ to machine $e + 1$ in **forward propagation**,

- $\mathcal{G}_e^{(t)}$: the filtration before the communication from machine $e + 1$ to machine $e$ in **backward propagation**.

### C.1 DESCENT LEMMA AND ANALYSIS OF THE EVALUATED GRADIENT

In this subsection, we firstly present the descent lemma.

**Lemma 2** (Descent Lemma). *Suppose Assumption 2 holds and $\gamma \le \dfrac{1}{2L_{\nabla \ell}}$, then in Algorithm 4 we have:*

$$\frac{1}{T} \sum_{t=1}^{T} \mathbb{E}\left[\left\|\nabla \ell(\boldsymbol{w}^{(t)})\right\|^2\right]$$

$$\le \frac{2}{\gamma T} \mathbb{E}\left[\ell(\boldsymbol{w}^{(1)}) - \inf_{\boldsymbol{w}} \ell(W)\right] - \frac{1}{2\gamma^2 T} \sum_{t=1}^{T} \mathbb{E}\left[\left\|\boldsymbol{w}^{(t+1)} - \boldsymbol{w}^{(t)}\right\|^2\right] + \frac{1}{T} \sum_{t=1}^{T} \sum_{e=1}^{E} \mathbb{E}\left[\left\|\tilde{u}_e^{(t)} - \nabla_e \ell(\boldsymbol{w}^{(t)})\right\|^2\right]. \tag{8}$$

*Proof.* According to the result in (Fatkhullin et al., 2024), we can get from Assumption 2 that:

$$\ell(\boldsymbol{w}^{(t+1)}) \le \ell(\boldsymbol{w}^{(t)}) - \frac{\gamma}{2} \left\|\nabla \ell(\boldsymbol{w}^{(t)})\right\|^2$$
$$- \left(\frac{1}{2\gamma} - \frac{L_{\nabla \ell}}{2}\right) \left\|\boldsymbol{w}^{(t+1)} - \boldsymbol{w}^{(t)}\right\|^2 + \frac{\gamma}{2} \sum_{e=1}^{E} \left\|\tilde{u}_e^{(t)} - \nabla_e \ell(\boldsymbol{w}^{(t)})\right\|^2. \tag{9}$$

Then, as $\gamma \le \dfrac{1}{2L_{\nabla \ell}}$, we have:

$$\ell(\boldsymbol{w}^{(t+1)}) - \inf_{\boldsymbol{w}} l(\boldsymbol{w}) \le \ell(\boldsymbol{w}^{(t)}) - \inf_{\boldsymbol{w}} \ell(\boldsymbol{w}) - \frac{\gamma}{2} \left\|\nabla \ell(\boldsymbol{w}^{(t)})\right\|^2$$
$$- \frac{1}{4\gamma} \left\|\boldsymbol{w}^{(t+1)} - \boldsymbol{w}^{(t)}\right\|^2 + \frac{\gamma}{2} \sum_{e=1}^{E} \left\|\tilde{u}_e^{(t)} - \nabla_e \ell(\boldsymbol{w}^{(t)})\right\|^2. \tag{10}$$

Taking expectation and summation on both sides over $t = 1, 2, \cdots, T$, we can get:

$$\mathbb{E}\left[\ell(\boldsymbol{w}^{(T+1)}) - \inf_{\boldsymbol{w}} \ell(\boldsymbol{w})\right] \le \mathbb{E}\left[\ell(\boldsymbol{w}^{(1)}) - \inf_{\boldsymbol{w}} l(\boldsymbol{w})\right] - \frac{\gamma}{2} \sum_{t=1}^{T} \mathbb{E}\left[\left\|\nabla \ell(\boldsymbol{w}^{(t)})\right\|^2\right]$$
$$- \frac{1}{4\gamma} \sum_{t=1}^{T} \mathbb{E}\left[\left\|\boldsymbol{w}^{(t+1)} - \boldsymbol{w}^{(t)}\right\|^2\right] + \frac{\gamma}{2} \sum_{t=1}^{T} \sum_{e=1}^{E} \mathbb{E}\left[\left\|\tilde{u}_e^{(t)} - \nabla_e \ell(\boldsymbol{w}^{(t)})\right\|^2\right]. \tag{11}$$

Finally, we have:

$$\frac{1}{T} \sum_{t=1}^{T} \mathbb{E}\left[\left\|\nabla \ell(\boldsymbol{w}^{(t)})\right\|^2\right]$$

$$\le \frac{2}{\gamma T} \mathbb{E}\left[\ell(\boldsymbol{w}^{(1)}) - \inf_{\boldsymbol{w}} \ell(W)\right] - \frac{1}{2\gamma^2 T} \sum_{t=1}^{T} \mathbb{E}\left[\left\|\boldsymbol{w}^{(t+1)} - \boldsymbol{w}^{(t)}\right\|^2\right] + \frac{1}{T} \sum_{t=1}^{T} \sum_{e=1}^{E} \mathbb{E}\left[\left\|\tilde{u}_e^{(t)} - \nabla_e \ell(\boldsymbol{w}^{(t)})\right\|^2\right].$$

$$\square$$

**Remark 1.** *It is noteworthy that one can get $\|\hat{v}_e^{(t)}\|$ is bounded for $e = 1, \cdots, E - 1$ from Assumption 1 and the definition of $\hat{v}_e^{(t)}$. If we let $a_e^{\circ}(v, y, w_e) = \nabla_2 a_e(y, w_e)^{\top} v$, and $L'_{\nabla a} = \mathcal{O}\left(\max\{L_{\nabla a}, L_{\nabla a}\|\hat{v}_e^{(t)}\|\}\right)$, then there exist $L_{\nabla a}^{\circ} = \mathcal{O}(C_a)$ such that*

$$\left\|a_e^{\circ}(\hat{v}_e^{(t)}, y, w_e) - a_e^{\circ}(v', y', w'_e)\right\|^2 = \left\|\nabla_2 a_e(y, w_e)^{\top} \hat{v}_e^{(t)} - \nabla_2 a_e(y', w'_e)^{\top} v'\right\|^2$$

$$\le 2 \left\|\nabla_2 a_e(y, w_e)^{\top} \hat{v}_e^{(t)} - \nabla_2 a_e(y', w'_e)^{\top} \hat{v}_e^{(t)}\right\|^2 + 2 \left\|\nabla_2 a_e(y', w'_e)^{\top} \hat{v}_e^{(t)} - \nabla_2 a_e(y', w'_e)^{\top} v'\right\|^2 \tag{12}$$

$$\le (L'_{\nabla a})^2 \left(\|y - y'\|^2 + \|w_e - w'_e\|^2\right) + (L_{\nabla a}^{\circ})^2 \left\|\hat{v}_e^{(t)} - v'\right\|^2,$$

*where the last inequality is due to the Lipschitz continuous of $\nabla_2 a_e$, the boundness of $\hat{v}_e^{(t)}$ and the boundness of $\nabla_2 a_e$.*

Next, we present the preliminary analysis of the error of stochastic gradient evaluation, i.e., the term $\left\|\tilde{u}_e^{(t)} - \nabla_e\ell(\boldsymbol{w}^{(t)})\right\|^2$ for $e = 1, 2, \cdots, E$.

**Lemma 3.** *Suppose Assumption 1 and 2 hold, and let $m_1 = \cdots = m_T = m_{T+1} = m$ as well as $p_3 = \cdots = p_T = p_{T+1} = p$. Moreover, we set $p_2 = 1$. Then, for all $t = 2, \cdots, T+1$ we have:*

$$
\sum_{e=1}^{E}\sum_{t=1}^{T+1}\mathbb{E}\left[\left\|\tilde{u}_e^{(t)} - \nabla_e\ell(\boldsymbol{w}^{(t)})\right\|^2\right]
$$

$$
\leq 32 L_{\nabla\ell}^2\left(\frac{p+m}{m^2(1-(1-p)(1-\frac{m}{2}))} + \frac{1}{m^2}\right)\sum_{e=1}^{E}\sum_{t=1}^{T}\mathbb{E}\left[\left\|w_e^{(t+1)} - w_e^{(t)}\right\|^2\right]
$$

$$
+ 8(L_{\nabla a}^\circ)^2\sum_{e=1}^{E-1}\sum_{t=2}^{T+1}\mathbb{E}\left[\left\|\tilde{v}_e^{(t)} - \hat{v}_e^{(t)}\right\|^2\right] + 8(L_{\nabla a}')^2\sum_{e=1}^{E-1}\sum_{t=2}^{T+1}\mathbb{E}\left[\left\|\tilde{y}_e^{(t)} - \hat{y}_e^{(t)}\right\|^2\right]
$$

$$
+ 4T\sigma^2\frac{(2-p)m - (1-p)m^2}{1-(1-p)(1-m)^2} + \frac{3}{m}\sum_{e=1}^{E}\mathbb{E}\left[\left\|\tilde{u}_e^{(1)} - \nabla_e\ell(\boldsymbol{w}^{(1)})\right\|^2\right].
$$

(13)

*Proof.* For $t = 2, 3, \cdots, T$, we denote $\psi(t)$ as the last moment in which the sample is randomly obtained with $\mathcal{D}$ as of the $t$-th iteration. Specially,

$$
\psi(t) := \max_{\tau \in \mathbb{S}_t} \tau, \quad \text{where } \mathbb{S}_t := \{\tau = 2, 3, \cdots, t | \text{sampling randomly at iteration } \tau\}.
$$

Then, with the fact that the $p_2 = 1$, it holds for $\tau = 2, \cdots, t$ that $\Pr(\psi(t) = \tau) = \begin{cases} (1-p)^{t-2}, & \text{if } \tau = 2 \\ p(1-p)^{t-\tau}, & \text{else.} \end{cases}$

For $e = 1, 2, \cdots, E-1$ and $t = 2, 3, \cdots, T+1$, the error between the evaluated gradient and the true gradient satisfies:

$$
\tilde{u}_e^{(t)} - \nabla_e\ell(\boldsymbol{w}^{(t)})
$$

$$
= \sum_{\tau=\psi(t)}^{t} m(1-m)^{t-\tau}\nabla_2 a_e(\tilde{y}_{e-1}^{(\tau)}, w_e^{(\tau)})^\top \tilde{v}_e^{(\tau)} + (1-m)^{t+1-\psi(t)}\tilde{u}_e^{(\psi(t)-1)} - \nabla_e\ell(\boldsymbol{w}^{(t)})
$$

$$
= \underbrace{\sum_{\tau=\psi(t)}^{t} m(1-m)^{t-\tau}\left(\nabla_2 a_e(\tilde{y}_{e-1}^{(\tau)}, w_e^{(\tau)})^\top \tilde{v}_e^{(\tau)} - \nabla_2 a_e(\hat{y}_{e-1}^{(\tau)}, w_e^{(\tau)})^\top \hat{v}_e^{(\tau)}\right)}_{:=\Xi_{e,1}}
$$

$$
+ \sum_{\tau=\psi(t)}^{t} m(1-m)^{t-\tau}\left(\nabla_2 a_e(\hat{y}_{e-1}^{(\tau)}, w_e^{(\tau)})^\top \hat{v}_e^{(\tau)} - \nabla_e\ell(\boldsymbol{w}^\tau)\right)
$$

$$
+ \underbrace{\sum_{\tau=\psi(t)}^{t} m(1-m)^{t-\tau}\left(\nabla_e\ell(\boldsymbol{w}^{(\tau)}) - \nabla_e\ell(\boldsymbol{w}^{(t)})\right)}_{:=\Xi_{e,2}} + \underbrace{(1-m)^{t+1-\psi(t)}\left(\nabla_e\ell(\boldsymbol{w}^{(\psi(t)-1)}) - \nabla_e\ell(\boldsymbol{w}^{(t)})\right)}_{:=\Xi_{e,3}}
$$

$$
+ (1-m)^{t+1-\psi(t)}\left(\tilde{u}_e^{(\psi(t)-1)} - \nabla_e\ell(\boldsymbol{w}^{(\psi(t)-1)})\right),
$$

(14)

where the first equation is from the momentum update rule. Moreover, we use $\Xi_{e,1}, \Xi_{e,2}, \Xi_{e,3}$ to denote some complex terms, which have been shown in Eq. (14).

Additionally, we denote $\mathcal{F}^{(t)}$ as the filtration before the $t$-th iteration. Thus, $\tilde{u}_e^{(\psi(t)-1)} - \nabla_e\ell(\boldsymbol{w}^{(\psi(t)-1)})$ is measureable with respect to $\mathcal{F}^{(\psi(t))}$ for any $e = 1, 2, \cdots, E$. Moreover, the sampling process at the iteration $\psi(t)$ is **independent** with respect to $\mathcal{F}^{(\psi(t))}$. Thus, $\nabla_2 a_e(\hat{y}_{e-1}^{(\tau)}, w_e^{(\tau)})\hat{v}_e^{(\tau)}$ is an unbiased estimation of the gradient $\nabla_e\ell(\boldsymbol{w}^{(\tau)})$ with bounded variance according to Assumption 2. Thus, taking the $\ell_2$-norm and conditional expectation with respect to

$\mathcal{F}^{(\psi(t))}$ on both sides of Eq. (14), we can obtain:

$$\mathbb{E}\left[\left\|\tilde{u}_e^{(t)} - \nabla_e\ell(\boldsymbol{w}^{(t)})\right\|^2\Big|\mathcal{F}^{(\psi(t))}\right]$$

$$=\mathbb{E}\left[\left\|\sum_{\tau=\psi(t)}^{t} m(1-m)^{t-\tau}\left(\nabla_2 a_e(\hat{y}_{e-1}^{(\tau)}, w_e^{(\tau)})^\top \hat{v}_e^{(\tau)} - \nabla_e\ell(\boldsymbol{w}^\tau)\right)\right\|^2\Bigg|\mathcal{F}^{(\psi(t))}\right]$$

$$+ \mathbb{E}\left[\left\|(1-m)^{t+1-\psi(t)}\left(\tilde{u}_e^{(\psi(t)-1)} - \nabla_e\ell(\boldsymbol{w}^{(\psi(t)-1)})\right) + \Xi_{e,1} + \Xi_{e,2} + \Xi_{e,3}\right\|^2\Big|\mathcal{F}^{(\psi(t))}\right]$$

$$+ 2\mathbb{E}\Bigg[\Bigg\langle \sum_{\tau=\psi(t)}^{t} m(1-m)^{t-\tau}\left(\nabla_2 a_e(\hat{y}_{e-1}^{(\tau)}, w_e^{(\tau)})^\top \hat{v}_e^{(\tau)} - \nabla_e\ell(\boldsymbol{w}^\tau)\right),$$

$$\qquad (1-m)^{t+1-\psi(t)}\left(\tilde{u}_e^{(\psi(t)-1)} - \nabla_e\ell(\boldsymbol{w}^{(\psi(t)-1)})\right) + \Xi_{e,1} + \Xi_{e,2} + \Xi_{e,3}\Bigg\rangle\Bigg|\mathcal{F}^{(\psi(t))}\Bigg] \tag{15}$$

$$\leq 2\mathbb{E}\left[\left\|\sum_{\tau=\psi(t)}^{t} m(1-m)^{t-\tau}\left(\nabla_2 a_e(\hat{y}_{e-1}^{(\tau)}, w_e^{(\tau)})^\top \hat{v}_e^{(\tau)} - \nabla_e\ell(\boldsymbol{w}^\tau)\right)\right\|^2\Bigg|\mathcal{F}^{(\psi(t))}\right]$$

$$+ \mathbb{E}\left[\left\|\Xi_{e,1} + \Xi_{e,2} + \Xi_{e,3}\right\|^2\Big|\mathcal{F}^{(\psi(t))}\right]$$

$$+ \mathbb{E}\left[\left\|(1-m)^{t+1-\psi(t)}\left(\tilde{u}_e^{(\psi(t)-1)} - \nabla_e\ell(\boldsymbol{w}^{(\psi(t)-1)})\right) + \Xi_{e,1} + \Xi_{e,2} + \Xi_{e,3}\right\|^2\Big|\mathcal{F}^{(\psi(t))}\right],$$

where the inequality is due to Cauchy-Schwarz inequality and Assumption 2.

For the second term of the right-hand-side of Eq. (15), it holds that:

$$\mathbb{E}\left[\left\|\Xi_{e,1} + \Xi_{e,2} + \Xi_{e,3}\right\|^2\Big|\mathcal{F}^{(\psi(t))}\right]$$

$$\leq 2\sum_{\tau=\psi(t)}^{t} m(1-m)^{t-\tau}\mathbb{E}\left[\left\|\nabla_2 a_e(\tilde{y}_{e-1}^{(\tau)}, w_e^{(\tau)})^\top \tilde{v}_e^{(\tau)} - \nabla_2 a_e(\hat{y}_{e-1}^{(\tau)}, w_e^{(\tau)})^\top \hat{v}_e^{(\tau)}\right\|^2\Big|\mathcal{F}^{(\psi(t))}\right]$$

$$+ 2\sum_{\tau=\psi(t)}^{t} m(1-m)^{t-\tau}\mathbb{E}\left[\left\|\nabla_e\ell(\boldsymbol{w}^{(\tau)}) - \nabla_e\ell(\boldsymbol{w}^{(t)})\right\|^2\Big|\mathcal{F}^{(\psi(t))}\right] \tag{16}$$

$$+ (1-m)^{t+1-\psi(t)}\mathbb{E}\left[\left\|\nabla_e\ell(\boldsymbol{w}^{(\psi(t)-1)}) - \nabla_e\ell(\boldsymbol{w}^{(t)})\right\|^2\Big|\mathcal{F}^{(\psi(t))}\right],$$

where the inequality holds is due to the convexity of the $\ell_2$-norm.

Moreover, for the last term, it also holds that:

$$\mathbb{E}\left[\left\|(1-m)^{t+1-\psi(t)}\left(\tilde{u}_e^{(\psi(t)-1)} - \nabla_e\ell(\boldsymbol{w}^{(\psi(t)-1)})\right) + \Xi_{e,1} + \Xi_{e,2} + \Xi_{e,3}\right\|^2\Big|\mathcal{F}^{(\psi(t))}\right]$$

$$\leq 2\sum_{\tau=\psi(t)}^{t} m(1-m)^{t-\tau}\mathbb{E}\left[\left\|\nabla_2 a_e(\tilde{y}_{e-1}^{(\tau)}, w_e^{(\tau)})^\top \tilde{v}_e^{(\tau)} - \nabla_2 a_e(\hat{y}_{e-1}^{(\tau)}, w_e^{(\tau)})^\top \hat{v}_e^{(\tau)}\right\|^2\Big|\mathcal{F}^{(\psi(t))}\right]$$

$$+ 2\sum_{\tau=\psi(t)}^{t} m(1-m)^{t-\tau}\mathbb{E}\left[\left\|\nabla_e\ell(\boldsymbol{w}^{(\tau)}) - \nabla_e\ell(\boldsymbol{w}^{(t)})\right\|^2\Big|\mathcal{F}^{(\psi(t))}\right]$$

$$+ (1-m)^{t+1-\psi(t)}\mathbb{E}\left[\left\|\left(\tilde{u}_e^{(\psi(t)-1)} - \nabla_e\ell(\boldsymbol{w}^{(\psi(t)-1)})\right) + \left(\nabla_e\ell(\boldsymbol{w}^{(\psi(t)-1)}) - \nabla_e\ell(\boldsymbol{w}^{(t)})\right)\right\|^2\Big|\mathcal{F}^{(\psi(t))}\right]. \tag{17}$$

Form Young's inequality, for any $u > 0$ the last term of (17) holds that:

$$\mathbb{E}\left[\left\|\left(\tilde{u}_e^{(\psi(t)-1)} - \nabla_e\ell(\boldsymbol{w}^{(\psi(t)-1)})\right) + \left(\nabla_e\ell(\boldsymbol{w}^{(\psi(t)-1)}) - \nabla_e\ell(\boldsymbol{w}^{(t)})\right)\right\|^2\Big|\mathcal{F}^{(\psi(t))}\right]$$

$$\leq (1+u)\mathbb{E}\left[\left\|\tilde{u}_e^{(\psi(t)-1)} - \nabla_e\ell(\boldsymbol{w}^{(\psi(t)-1)})\right\|^2\Big|\mathcal{F}^{(\psi(t))}\right] \tag{18}$$

$$+ \left(1 + \frac{1}{u}\right)\mathbb{E}\left[\left\|\nabla_e\ell(\boldsymbol{w}^{(\psi(t)-1)}) - \nabla_e\ell(\boldsymbol{w}^{(t)})\right\|^2\Big|\mathcal{F}^{(\psi(t))}\right].$$

If we take $u = \dfrac{\left(1 - \frac{m}{2}\right)^{t+1-\psi(t)} - (1 - m)^{t+1-\psi(t)}}{(1 - m)^{t+1-\psi(t)}}$, then $1 + u = \dfrac{\left(1 - \frac{m}{2}\right)^{t+1-\psi(t)}}{(1 - m)^{t+1-\psi(t)}}$ and

$$
\begin{aligned}
1 + \frac{1}{u} &= 1 + \frac{(1 - m)^{t+1-\psi(t)}}{\left(1 - \frac{m}{2}\right)^{t+1-\psi(t)} - (1 - m)^{t+1-\psi(t)}} = 1 + \frac{(1 - m)^{t+1-\psi(t)}}{\frac{m}{2}\left(\sum_{j=0}^{t-\psi(t)}\left(1 - \frac{m}{2}\right)^j (1 - m)^{t-\psi(t)-j}\right)} \\
&\leq 1 + \frac{2}{m(t + 1 - \psi(t))}.
\end{aligned}
$$

Substituting (18) into Eq. (17) and taking the value of $u$, we can obtain that:

$$
\begin{aligned}
&\mathbb{E}\left[\left\|(1 - m)^{t+1-\psi(t)}\left(\tilde{u}_e^{(\psi(t)-1)} - \nabla_e \ell(\boldsymbol{w}^{(\psi(t)-1)})\right) + \Xi_{e,1} + \Xi_{e,2} + \Xi_{e,3}\right\|^2 \middle| \mathcal{F}^{(\psi(t))}\right] \\
&\leq 2\sum_{\tau=\psi(t)}^{t} m(1 - m)^{t-\tau}\mathbb{E}\left[\left\|\nabla_2 a_e(\tilde{y}_{e-1}^{(\tau)}, w_e^{(\tau)})^\top \tilde{v}_e^{(\tau)} - \nabla_2 a_e(\hat{y}_{e-1}^{(\tau)}, w_e^{(\tau)})^\top \hat{v}_e^{(\tau)}\right\|^2 \middle| \mathcal{F}^{(\psi(t))}\right] \\
&\quad + 2\sum_{\tau=\psi(t)}^{t} m(1 - m)^{t-\tau}\mathbb{E}\left[\left\|\nabla_e \ell(\boldsymbol{w}^{(\tau)}) - \nabla_e \ell(\boldsymbol{w}^{(t)})\right\|^2 \middle| \mathcal{F}^{(\psi(t))}\right] \\
&\quad + \left(1 - \frac{m}{2}\right)^{t+1-\psi(t)}\mathbb{E}\left[\left\|\tilde{u}_e^{(\psi(t)-1)} - \nabla_e \ell(\boldsymbol{w}^{(\psi(t)-1)})\right\|^2 \middle| \mathcal{F}^{(\psi(t))}\right] \\
&\quad + (1 - m)^{t+1-\psi(t)}\left(1 + \frac{2}{m(t + 1 - \psi(t))}\right)\mathbb{E}\left[\left\|\nabla_e \ell(\boldsymbol{w}^{(\psi(t)-1)}) - \nabla_e \ell(\boldsymbol{w}^{(t)})\right\|^2 \middle| \mathcal{F}^{(\psi(t))}\right].
\end{aligned}
\tag{19}
$$

Substituting Eq. (16) and Eq. (19) into Eq.(15), we can obtain that:

$$
\begin{aligned}
&\mathbb{E}\left[\left\|\tilde{u}_e^{(t)} - \nabla_e \ell(\boldsymbol{w}^{(t)})\right\|^2 \middle| \mathcal{F}^{(\psi(t))}\right] \\
&\leq 2\mathbb{E}\left[\left\|\sum_{\tau=\psi(t)}^{t} m(1 - m)^{t-\tau}\left(\nabla_2 a_e(\hat{y}_{e-1}^{(\tau)}, w_e^{(\tau)})^\top \hat{v}_e^{(\tau)} - \nabla_e \ell(\boldsymbol{w}^\tau)\right)\right\|^2 \middle| \mathcal{F}^{(\psi(t))}\right] \\
&\quad + 4\sum_{\tau=\psi(t)}^{t} m(1 - m)^{t-\tau}\mathbb{E}\left[\left\|\nabla_2 a_e(\tilde{y}_{e-1}^{(\tau)}, w_e^{(\tau)})^\top \tilde{v}_e^{(\tau)} - \nabla_2 a_e(\hat{y}_{e-1}^{(\tau)}, w_e^{(\tau)})^\top \hat{v}_e^{(\tau)}\right\|^2 \middle| \mathcal{F}^{(\psi(t))}\right] \\
&\quad + 4\sum_{\tau=\psi(t)}^{t} m(1 - m)^{t-\tau}\mathbb{E}\left[\left\|\nabla_e \ell(\boldsymbol{w}^{(\tau)}) - \nabla_e \ell(\boldsymbol{w}^{(t)})\right\|^2 \middle| \mathcal{F}^{(\psi(t))}\right] \\
&\quad + \left(1 - \frac{m}{2}\right)^{t+1-\psi(t)}\mathbb{E}\left[\left\|\tilde{u}_e^{(\psi(t)-1)} - \nabla_e \ell(\boldsymbol{w}^{(\psi(t)-1)})\right\|^2 \middle| \mathcal{F}^{(\psi(t))}\right] \\
&\quad + 2(1 - m)^{t+1-\psi(t)}\left(1 + \frac{1}{m(t + 1 - \psi(t))}\right)\mathbb{E}\left[\left\|\nabla_e \ell(\boldsymbol{w}^{(\psi(t)-1)}) - \nabla_e \ell(\boldsymbol{w}^{(t)})\right\|^2 \middle| \mathcal{F}^{(\psi(t))}\right].
\end{aligned}
\tag{20}
$$

With Assumption 2, we can get:

$$
\mathbb{E}\left[\sum_{e=1}^{E}\left\|\sum_{\tau=\psi(t)}^{t} m(1 - m)^{t-\tau}\left(\nabla_2 a_e(\hat{y}_{e-1}^{(\tau)}, w_e^{(\tau)})^\top \hat{v}_e^{(\tau)} - \nabla_e \ell(\boldsymbol{w}^\tau)\right)\right\|^2 \middle| \mathcal{F}^{(\psi(t))}\right] \leq \left(\sum_{\tau=\psi(t)}^{t} m(1 - m)^{t-\tau}\right)^2 \sigma^2.
\tag{21}
$$

We denote $\Lambda^{(\tau)}$ as the vector resulting from connecting $\nabla_2 a_e(\tilde{y}_{e-1}^{(\tau)}, w_e^{(\tau)})^\top \tilde{v}_e^{(\tau)} - \nabla_2 a_e(\hat{y}_{e-1}^{(\tau)}, w_e^{(\tau)})^\top \hat{v}_e^{(\tau)}$ for all $e = 1, 2, \cdots, E$ end-to-end. Then, taking the summation on

both sides of Eq. (20), it holds that:

$$
\mathbb{E}\left[\sum_{e=1}^{E}\left\|\tilde{u}_e^{(t)} - \nabla_e \ell(\boldsymbol{w}^{(t)})\right\|^2 \middle| \mathcal{F}^{(\psi(t))}\right]
$$

$$
\leq 2\left(\sum_{\tau=\psi(t)}^{t} m(1-m)^{t-\tau}\right)^2 \sigma^2 + 4\sum_{\tau=\psi(t)}^{t} m(1-m)^{t-\tau}\mathbb{E}\left[\left\|\Lambda^{(\tau)}\right\|^2 \middle| \mathcal{F}^{(\psi(t))}\right]
$$

$$
+ 4\sum_{\tau=\psi(t)}^{t} m(1-m)^{t-\tau}\mathbb{E}\left[\left\|\nabla \ell(\boldsymbol{w}^{(\tau)}) - \nabla \ell(\boldsymbol{w}^{(t)})\right\|^2 \middle| \mathcal{F}^{(\psi(t))}\right] \tag{22}
$$

$$
+ \left(1-\frac{m}{2}\right)^{t+1-\psi(t)}\mathbb{E}\left[\sum_{e=1}^{E}\left\|\tilde{u}_e^{(\psi(t)-1)} - \nabla_e \ell(\boldsymbol{w}^{(\psi(t)-1)})\right\|^2 \middle| \mathcal{F}^{(\psi(t))}\right]
$$

$$
+ 2(1-m)^{t+1-\psi(t)}\left(1 + \frac{1}{m(t+1-\psi(t))}\right)\mathbb{E}\left[\left\|\nabla \ell(\boldsymbol{w}^{(\psi(t)-1)}) - \nabla \ell(\boldsymbol{w}^{(t)})\right\|^2 \middle| \mathcal{F}^{(\psi(t))}\right].
$$

Then, taking the conditional expectation with respect to $\psi(t)$ on both sides of (22), it holds that:

$$
\mathbb{E}\left[\sum_{e=1}^{E}\left\|\tilde{u}_e^{(t)} - \nabla_e \ell(\boldsymbol{w}^{(t)})\right\|^2 \middle| \psi(t)\right]
$$

$$
\leq 2\mathbb{E}\left[\left(\sum_{\tau=\psi(t)}^{t} m(1-m)^{t-\tau}\right)^2 \sigma^2 \middle| \psi(t)\right] + 4\sum_{\tau=\psi(t)}^{t} m(1-m)^{t-\tau}\mathbb{E}\left[\left\|\Lambda^{(\tau)}\right\|^2 \middle| \psi(t)\right]
$$

$$
+ 4\sum_{\tau=\psi(t)}^{t} m(1-m)^{t-\tau}\mathbb{E}\left[\left\|\nabla \ell(\boldsymbol{w}^{(\tau)}) - \nabla \ell(\boldsymbol{w}^{(t)})\right\|^2 \middle| \psi(t)\right] \tag{23}
$$

$$
+ \left(1-\frac{m}{2}\right)^{t+1-\psi(t)}\mathbb{E}\left[\sum_{e=1}^{E}\left\|\tilde{u}_e^{(\psi(t)-1)} - \nabla_e \ell(\boldsymbol{w}^{(\psi(t)-1)})\right\|^2 \middle| \psi(t)\right]
$$

$$
+ 2(1-m)^{t+1-\psi(t)}\left(1 + \frac{1}{m(t+1-\psi(t))}\right)\mathbb{E}\left[\left\|\nabla \ell(\boldsymbol{w}^{(\psi(t)-1)}) - \nabla \ell(\boldsymbol{w}^{(t)})\right\|^2 \middle| \psi(t)\right].
$$

Furthermore, taking the expectation over $\psi(t)$, it holds that:

$$
\mathbb{E}\left[\sum_{e=1}^{E}\left\|\tilde{u}_e^{(t)} - \nabla_e \ell(\boldsymbol{w}^{(t)})\right\|^2\right]
$$

$$
\leq \sum_{\kappa=2}^{t} \Pr(\psi(t) = \kappa)\left(1-\frac{m}{2}\right)^{t+1-\kappa}\mathbb{E}\left[\sum_{e=1}^{E}\left\|\tilde{u}_e^{(\kappa-1)} - \nabla_e \ell(\boldsymbol{w}^{(\kappa-1)})\right\|^2\right]
$$

$$
+ 4\sum_{\kappa=2}^{t} \Pr(\psi(t) = \kappa)\sum_{\tau=\kappa}^{t} m(1-m)^{t-\tau}\mathbb{E}\left[\left\|\nabla \ell(\boldsymbol{w}^{(\tau)}) - \nabla \ell(\boldsymbol{w}^{(t)})\right\|^2\right]
$$

$$
+ 2\sum_{\kappa=2}^{t} \Pr(\psi(t) = \kappa)\left(1-(1-m)^{t-\kappa+1}\right)^2 \sigma^2 + 4\sum_{\kappa=2}^{t} \Pr(\psi(t) = \kappa)\sum_{\tau=\kappa}^{t} m(1-m)^{t-\tau}\mathbb{E}\left[\left\|\Lambda^{(\tau)}\right\|^2\right]
$$

$$
+ 2\sum_{\kappa=2}^{t} \Pr(\psi(t) = \kappa)(1-m)^{t+1-\kappa}\left(1 + \frac{1}{m(t+1-\kappa)}\right)\mathbb{E}\left[\left\|\nabla \ell(\boldsymbol{w}^{(\kappa-1)}) - \nabla \ell(\boldsymbol{w}^{(t)})\right\|^2\right].
$$

$$
\tag{24}
$$

Thus,

$$\mathbb{E}\left[\sum_{e=1}^{E}\left\|\tilde{u}_e^{(t)} - \nabla_e \ell(\boldsymbol{w}^{(t)})\right\|^2\right]$$

$$\leq \left(1 - \frac{m}{2}\right)\sum_{\kappa=2}^{t}\Pr(\psi(t) = \kappa)\left(1 - \frac{m}{2}\right)^{t-\kappa}\mathbb{E}\left[\sum_{e=1}^{E}\left\|\tilde{u}_e^{(\kappa-1)} - \nabla_e \ell(\boldsymbol{w}^{(\kappa-1)})\right\|^2\right]$$

$$+ 4\sum_{\kappa=2}^{t}\left(\Pr(\psi(t) = \kappa)(1-m)^{t+1-\kappa}\left(1 + \frac{1}{m(t+1-\kappa)}\right)\right.$$

$$\left. + \sum_{\tau=2}^{\kappa-1}\Pr(\psi(t) = \tau)m(1-m)^{t+1-\kappa}\right)\mathbb{E}\left[\left\|\nabla\ell(\boldsymbol{w}^{(\kappa-1)}) - \nabla\ell(\boldsymbol{w}^{(t)})\right\|^2\right]$$

$$+ 2\sum_{\kappa=2}^{t}\Pr(\psi(t) = \kappa)\left(1 - (1-m)^{t-\kappa+1}\right)^2\sigma^2 + 4\sum_{\kappa=2}^{t}\left(\sum_{\tau=2}^{\kappa}\Pr(\psi(t) = \tau)\right)m(1-m)^{t-\kappa}\mathbb{E}\left[\left\|\Lambda^{(\kappa)}\right\|^2\right]. \tag{25}$$

Taking summatation over $t = 2, \cdots, T+1$, we can obtain that:

$$\sum_{t=2}^{T+1}\mathbb{E}\left[\sum_{e=1}^{E}\left\|\tilde{u}_e^{(t)} - \nabla_e \ell(\boldsymbol{w}^{(t)})\right\|^2\right]$$

$$\leq \left(1 - \frac{m}{2}\right)\sum_{t=2}^{T+1}\sum_{\kappa=2}^{t}\Pr(\psi(t) = \kappa)\left(1 - \frac{m}{2}\right)^{t-\kappa}\mathbb{E}\left[\sum_{e=1}^{E}\left\|\tilde{u}_e^{(\kappa-1)} - \nabla_e \ell(\boldsymbol{w}^{(\kappa-1)})\right\|^2\right]$$

$$+ 4\sum_{t=2}^{T+1}\sum_{\kappa=2}^{t}\left(\Pr(\psi(t) = \kappa)(1-m)^{t+1-\kappa}\left(1 + \frac{1}{m(t+1-\kappa)}\right)\right.$$

$$\left. + \sum_{\tau=2}^{\kappa-1}\Pr(\psi(t) = \tau)m(1-m)^{t+1-\kappa}\right)\mathbb{E}\left[\left\|\nabla\ell(\boldsymbol{w}^{(\kappa-1)}) - \nabla\ell(\boldsymbol{w}^{(t)})\right\|^2\right] \tag{26}$$

$$+ 2\sum_{t=2}^{T+1}\sum_{\kappa=2}^{t}\Pr(\psi(t) = \kappa)\left(1 - (1-m)^{t-\kappa+1}\right)^2\sigma^2$$

$$+ 4\sum_{t=2}^{T+1}\sum_{\kappa=2}^{t}\left(\sum_{\tau=2}^{\kappa}\Pr(\psi(t) = \tau)\right)m(1-m)^{t-\kappa}\mathbb{E}\left[\left\|\Lambda^{(\kappa)}\right\|^2\right].$$

Here we consider each term of Eq. (26). Firstly, we can obtain that:

$$\sum_{\kappa=2}^{t}(1-m)^{t+1-\kappa}\left(\Pr(\psi(t) = \kappa)\left(1 + \frac{1}{m(t+1-\kappa)}\right) + m\sum_{\tau=2}^{\kappa-1}\Pr(\psi(t) = \tau)\right)\mathbb{E}\left[\left\|\nabla\ell(\boldsymbol{w}^{(\kappa-1)}) - \nabla\ell(\boldsymbol{w}^{(t)})\right\|^2\right]$$

$$\leq \sum_{\kappa=2}^{t}(t+1-\kappa)(1-m)^{t+1-\kappa}\left(\Pr(\psi(t) = \kappa)\left(1 + \frac{1}{m(t+1-\kappa)}\right) + m\sum_{\tau=2}^{\kappa-1}\Pr(\psi(t) = \tau)\right)$$

$$\cdot\sum_{\tau=\kappa}^{t}\mathbb{E}\left[\left\|\nabla\ell(\boldsymbol{w}^{(\tau-1)}) - \nabla\ell(\boldsymbol{w}^{(\tau)})\right\|^2\right]$$

$$= \sum_{\kappa=1}^{t-1}(1-m)^{t-\kappa}\left(\Pr(\psi(t) = \kappa+1)(t-\kappa+\frac{1}{m}) + m(t-\kappa)\sum_{\tau=2}^{\kappa}\Pr(\psi(t) = \tau)\right)$$

$$\cdot\sum_{\tau=\kappa}^{t-1}\mathbb{E}\left[\left\|\nabla\ell(\boldsymbol{w}^{(\tau)}) - \nabla\ell(\boldsymbol{w}^{(\tau+1)})\right\|^2\right]$$

$$= \sum_{\kappa=1}^{t-1}\left[\sum_{\tau=1}^{\kappa}(1-m)^{t-\tau}\left(\Pr(\psi(t) = \tau+1)(t-\tau+\frac{1}{m}) + m(t-\tau)\sum_{\iota=2}^{\tau}\Pr(\psi(t) = \iota)\right)\right]$$

$$\cdot\mathbb{E}\left[\left\|\nabla\ell(\boldsymbol{w}^{(\kappa)}) - \nabla\ell(\boldsymbol{w}^{(\kappa+1)})\right\|^2\right]. \tag{27}$$

let $s := t - \tau$, then the coefficient of the term $\mathbb{E}\left[\left\|\nabla\ell(\boldsymbol{w}^{(\kappa)}) - \nabla\ell(\boldsymbol{w}^{(\kappa+1)})\right\|^2\right]$ in Eq. (27) holds that:

$$
\sum_{\tau=1}^{\kappa}(1-m)^{t-\tau}\left(\Pr(\psi(t)=\tau+1)(t-\tau+\frac{1}{m})+m(t-\tau)\sum_{\iota=2}^{\tau}\Pr(\psi(t)=\iota)\right)
$$

$$
=\sum_{\tau=2}^{\kappa}(1-m)^{t-\tau}\left(\Pr(\psi(t)=\tau+1)(t-\tau+\frac{1}{m})+m(t-\tau)\sum_{\iota=2}^{\tau}\Pr(\psi(t)=\iota)\right)+(1-m)^{t-1}(1-p)^{t-2}(t-1+\frac{1}{m})
$$

$$
=\sum_{\tau=2}^{\kappa}(1-m)^{t-\tau}\left(p(1-p)^{t-\tau-1}(t-\tau+\frac{1}{m})+m(t-\tau)(1-p)^{t-\tau}\right)+(1-m)^{t-1}(1-p)^{t-2}(t-1+\frac{1}{m})
$$

$$
=\sum_{s=t-\kappa}^{t-2}(1-m)^{s}\left(p(1-p)^{s-1}(s+\frac{1}{m})+ms(1-p)^{s}\right)+(1-m)^{t-1}(1-p)^{t-2}(t-1+\frac{1}{m})
$$

$$
=\sum_{s=t-\kappa}^{t-2}\left[(1-m)^{s}(1-p)^{s-1}s(p+m(1-p))+\frac{p}{m}(1-m)^{s}(1-p)^{s-1}\right]+(1-m)^{t-1}(1-p)^{t-2}(t-1+\frac{1}{m}),
$$
$$\tag{28}$$

where the first equation is due to the fact that $\sum_{\iota=2}^{\tau}\Pr(\psi(t)=\iota)=(1-p)^{t-\tau}$ for $\tau=2,\cdots,T$.

Substituting Eq. (28) into Eq. (27), it holds that: Thus, we can obtain that:

$$
\sum_{\kappa=2}^{t}(1-m)^{t+1-\kappa}\left(\Pr(\psi(t)=\kappa)\left(1+\frac{1}{m(t+1-\kappa)}\right)+m\sum_{\tau=2}^{\kappa-1}\Pr(\psi(t)=\tau)\right)\mathbb{E}\left[\left\|\nabla\ell(\boldsymbol{w}^{(\kappa-1)})-\nabla\ell(\boldsymbol{w}^{(t)})\right\|^2\right]
$$

$$
\leq\sum_{\kappa=1}^{t-1}\left[\sum_{s=t-\kappa}^{t-2}\left[(1-m)^{s}(1-p)^{s-1}s(p+m(1-p))+\frac{p}{m}(1-m)^{s}(1-p)^{s-1}\right]+(1-m)^{t-1}(1-p)^{t-2}(t-1+\frac{1}{m})\right]
$$

$$
\cdot\mathbb{E}\left[\left\|\nabla\ell(\boldsymbol{w}^{(\kappa)})-\nabla\ell(\boldsymbol{w}^{(\kappa+1)})\right\|^2\right]
$$

$$
\leq(p+m(1-p))\sum_{\kappa=1}^{t-1}\left[\left(\sum_{\tau=t-\kappa}^{t-2}(1-m)^{\tau}(1-p)^{\tau-1}\tau\right)\mathbb{E}\left[\left\|\nabla\ell(\boldsymbol{w}^{(\kappa)})-\nabla\ell(\boldsymbol{w}^{(\kappa+1)})\right\|^2\right]\right]
$$

$$
+\frac{p}{m}\sum_{\kappa=1}^{t-1}\left[\left(\sum_{\tau=t-\kappa}^{t-2}(1-m)^{\tau}(1-p)^{\tau-1}\right)\mathbb{E}\left[\left\|\nabla\ell(\boldsymbol{w}^{(\kappa)})-\nabla\ell(\boldsymbol{w}^{(\kappa+1)})\right\|^2\right]\right]
$$

$$
+(1-m)^{t-1}(1-p)^{t-2}(t-1+\frac{1}{m})\sum_{\kappa=1}^{t-1}\mathbb{E}\left[\left\|\nabla\ell(\boldsymbol{w}^{(\kappa)})-\nabla\ell(\boldsymbol{w}^{(\kappa+1)})\right\|^2\right].
$$
$$\tag{29}$$

Taking summation on both sides over $t=1,2,\cdots,T$ and use the fact that $p\leq m\leq1$, it holds that:

$$
\sum_{t=1}^{T}\left[\sum_{\kappa=2}^{t}(1-m)^{t+1-\kappa}\left(\Pr(\psi(t)=\kappa)\left(1+\frac{1}{m(t+1-\kappa)}\right)+m\sum_{\tau=2}^{\kappa-1}\Pr(\psi(t)=\tau)\right)\right.
$$

$$
\left.\cdot\mathbb{E}\left[\left\|\nabla\ell(\boldsymbol{w}^{(\kappa-1)})-\nabla\ell(\boldsymbol{w}^{(t)})\right\|^2\right]\right]
$$

$$
\leq\sum_{t=1}^{T}\left((p+m(1-p))\sum_{\tau=1}^{+\infty}(1-m)^{\tau}(1-p)^{\tau-1}\tau^2+\frac{p+m}{m}\sum_{\tau=1}^{+\infty}(1-m)^{\tau}(1-p)^{\tau-1}\tau+\frac{1}{m}\sum_{\tau=1}^{+\infty}(1-m)^{\tau}(1-p)^{\tau-1}\right)
$$

$$
\cdot\mathbb{E}\left[\left\|\nabla\ell(\boldsymbol{w}^{(t)})-\nabla\ell(\boldsymbol{w}^{(t+1)})\right\|^2\right]
$$

$$
\leq\sum_{t=1}^{T}\left(\frac{(1-m)(1+(1-p)(1-m))}{(1-(1-p)(1-m))^2}+\frac{p+m}{m}\cdot\frac{1-m}{(1-(1-p)(1-m))^2}+\frac{1}{m}\cdot\frac{1-m}{1-(1-p)(1-m)}\right)
$$

$$
\cdot\mathbb{E}\left[\left\|\nabla\ell(\boldsymbol{w}^{(t)})-\nabla\ell(\boldsymbol{w}^{(t+1)})\right\|^2\right]
$$

$$
\leq\left(\frac{4(p+m)}{m(1-(1-p)(1-m))^2}+\frac{4}{m(1-(1-p)(1-m))}\right)\sum_{t=1}^{T}\mathbb{E}\left[\left\|\nabla\ell(\boldsymbol{w}^{(t)})-\nabla\ell(\boldsymbol{w}^{(t+1)})\right\|^2\right].
$$
$$\tag{30}$$

Moreover, it also holds that:

$$\sum_{\kappa=2}^{t} \Pr(\psi(t) = \kappa) \left(1 - (1-m)^{t-\kappa+1}\right)^2$$

$$= (1-p)^{t-2} \left(1 - (1-m)^{t-1}\right)^2 + \sum_{\kappa=3}^{t} p(1-p)^{t-\kappa} \left(1 - (1-m)^{t-\kappa+1}\right)^2$$

$$\leq \sum_{s=t-2}^{+\infty} p(1-p)^s \left(1 - (1-m)^{s+1}\right)^2 + \sum_{\kappa=3}^{t} p(1-p)^{t-\kappa} \left(1 - (1-m)^{t-\kappa+1}\right)^2$$

$$= \sum_{s=0}^{+\infty} p(1-p)^s \left(1 - (1-m)^{s+1}\right)^2 = \frac{(2-p)m^2 - (1-p)m^3}{(1 - (1-p)(1-m))(1 - (1-p)(1-m)^2)}.$$

(31)

Finally, we can obtain that:

$$(1 - \frac{m}{2}) \sum_{t=2}^{T+1} \left( \sum_{\kappa=2}^{t} \Pr(\psi(t) = \kappa)(1 - \frac{m}{2})^{t-\kappa} \mathbb{E}\left[ \sum_{e=1}^{E} \left\| \tilde{u}_e^{(\kappa-1)} - \nabla_e \ell(\boldsymbol{w}^{(\kappa-1)}) \right\|^2 \right] \right)$$

$$= (1 - \frac{m}{2}) \sum_{t=2}^{T+1} \left( \sum_{\kappa=t}^{T+1} \Pr(\psi(\kappa) = t)(1 - \frac{m}{2})^{\kappa-t} \mathbb{E}\left[ \sum_{e=1}^{E} \left\| \tilde{u}_e^{(t-1)} - \nabla_e \ell(\boldsymbol{w}^{(t-1)}) \right\|^2 \right] \right).$$

(32)

We note that

$$\sum_{\kappa=t}^{T+1} (1-m)^{\kappa-t}(1-p)^{\kappa-t} \leq \frac{1}{1 - (1-m)(1-p)}.$$

For $t \geq 3$, it holds that

$$\sum_{\kappa=t}^{T+1} \Pr(\psi(\kappa) = t)(1 - \frac{m}{2})^{\kappa-t} = p \sum_{\kappa=t}^{T+1} (1-p)^{\kappa-t}(1 - \frac{m}{2})^{\kappa-t} \leq \frac{p}{1 - (1-p)(1 - \frac{m}{2})}.$$

For $t = 2$, it holds that

$$\sum_{\kappa=2}^{T+1} \Pr(\psi(\kappa) = 2)(1 - \frac{m}{2})^{\kappa-2} = \sum_{\kappa=2}^{T+1} (1-p)^{\kappa-2}(1 - \frac{m}{2})^{\kappa-2} \leq \frac{1}{1 - (1-p)(1 - \frac{m}{2})}.$$

Combining Eq. (26), (30), (31), (32) together, we can obtain that:

$$\sum_{t=2}^{T+1} \mathbb{E}\left[ \sum_{e=1}^{E} \left\| \tilde{u}_e^{(t)} - \nabla_e \ell(\boldsymbol{w}^{(t)}) \right\|^2 \right]$$

$$\leq 2T\sigma^2 \frac{(2-p)m^2 - (1-p)m^3}{(1 - (1-p)(1-m))(1 - (1-p)(1-m)^2)} + \frac{4m}{1 - (1-p)(1 - \frac{m}{2})} \sum_{t=2}^{T+1} \mathbb{E}\left[ \left\| \Lambda^{(t)} \right\|^2 \right]$$

$$+ \frac{(1 - \frac{m}{2})p}{1 - (1-p)(1 - \frac{m}{2})} \sum_{t=2}^{T} \mathbb{E}\left[ \sum_{e=1}^{E} \left\| \tilde{u}_e^{(t)} - \nabla_e \ell(\boldsymbol{w}^{(t)}) \right\|^2 \right] + \frac{1}{1 - (1-p)(1 - \frac{m}{2})} \mathbb{E}\left[ \sum_{e=1}^{E} \left\| \tilde{u}_e^{(1)} - \nabla_e \ell(\boldsymbol{w}^{(1)}) \right\|^2 \right]$$

$$+ 16 \left( \frac{1}{m(1 - (1-p)(1 - \frac{m}{2}))} + \frac{p+m}{m(1 - (1-p)(1 - \frac{m}{2}))^2} \right) \sum_{t=1}^{T} \mathbb{E}\left[ \left\| \nabla \ell(\boldsymbol{w}^{(t)}) - \nabla \ell(\boldsymbol{w}^{(t+1)}) \right\|^2 \right].$$

(33)

Thus, we can obtain that:

$$\frac{\frac{m}{2}}{1 - (1-p)(1 - \frac{m}{2})} \sum_{t=1}^{T+1} \mathbb{E}\left[ \sum_{e=1}^{E} \left\| \tilde{u}_e^{(t)} - \nabla_e \ell(\boldsymbol{w}^{(t)}) \right\|^2 \right]$$

$$\leq 2T\sigma^2 \frac{(2-p)m^2 - (1-p)m^3}{(1 - (1-p)(1 - \frac{m}{2}))(1 - (1-p)(1-m)^2)} + \frac{4m}{1 - (1-p)(1 - \frac{m}{2})} \sum_{t=2}^{T+1} \mathbb{E}\left[ \left\| \Lambda^{(t)} \right\|^2 \right]$$

$$+ \frac{\frac{m}{2} + 1}{1 - (1-p)(1 - \frac{m}{2})} \mathbb{E}\left[ \sum_{e=1}^{E} \left\| \tilde{u}_e^{(1)} - \nabla_e \ell(\boldsymbol{w}^{(1)}) \right\|^2 \right]$$

$$+ 16 \left( \frac{1}{m(1 - (1-p)(1 - \frac{m}{2}))} + \frac{p+m}{m(1 - (1-p)(1 - \frac{m}{2}))^2} \right) \sum_{t=1}^{T} \mathbb{E}\left[ \left\| \nabla \ell(\boldsymbol{w}^{(t)}) - \nabla \ell(\boldsymbol{w}^{(t+1)}) \right\|^2 \right].$$

(34)

Then, we can get that:

$$\sum_{t=1}^{T+1} \mathbb{E}\left[\sum_{e=1}^{E}\left\|\tilde{u}_e^{(t)} - \nabla_e \ell(\boldsymbol{w}^{(t)})\right\|^2\right]$$

$$\leq 4T\sigma^2\frac{(2-p)m-(1-p)m^2}{1-(1-p)(1-m)^2} + \frac{3}{m}\mathbb{E}\left[\sum_{e=1}^{E}\left\|\tilde{u}_e^{(1)} - \nabla_e \ell(\boldsymbol{w}^{(1)})\right\|^2\right] + 8\sum_{t=2}^{T+1}\mathbb{E}\left[\left\|\Lambda^{(t)}\right\|^2\right] \quad (35)$$

$$+ 32\left(\frac{p+m}{m^2(1-(1-p)(1-\frac{m}{2}))} + \frac{1}{m^2}\right)\sum_{t=1}^{T}\mathbb{E}\left[\left\|\nabla\ell(\boldsymbol{w}^{(t)}) - \nabla\ell(\boldsymbol{w}^{(t+1)})\right\|^2\right].$$

Finally, with Eq. (12), it holds that

$$\mathbb{E}\left[\left\|\Lambda^{(t)}\right\|^2\right] = \mathbb{E}\left[\sum_{e=1}^{E}\left\|\nabla_2 a_e(\tilde{y}_{e-1}^{(t)}, w_e^{(t)})^\top \tilde{v}_e^{(t)} - \nabla_2 a_e(\hat{y}_{e-1}^{(t)}, w_e^{(t)})^\top \hat{v}_e^{(t)}\right\|^2\right]$$

$$\leq (L_{\nabla a}^\circ)^2\sum_{e=1}^{E}\mathbb{E}\left[\left\|\tilde{v}_e^{(t)} - \hat{v}_e^{(t)}\right\|^2\right] + (L_{\nabla a}')^2\sum_{e=1}^{E}\mathbb{E}\left[\left\|\tilde{y}_{e-1}^{(t)} - \hat{y}_{e-1}^{(t)}\right\|^2\right] \quad (36)$$

$$= (L_{\nabla a}^\circ)^2\sum_{e=1}^{E-1}\mathbb{E}\left[\left\|\tilde{v}_e^{(t)} - \hat{v}_e^{(t)}\right\|^2\right] + (L_{\nabla a}')^2\sum_{e=1}^{E-1}\mathbb{E}\left[\left\|\tilde{y}_e^{(t)} - \hat{y}_e^{(t)}\right\|^2\right],$$

where the last equation is from the fact that $\hat{y}_0 = \tilde{y}_0 = x_0$ and $\hat{v}_E = \tilde{v}_E = 1$.

Thus, with Assumption 1, it holds that:

$$\sum_{e=1}^{E}\sum_{t=1}^{T+1}\mathbb{E}\left[\left\|\tilde{u}_e^{(t)} - \nabla_e \ell(\boldsymbol{w}^{(t)})\right\|^2\right]$$

$$\leq 32L_{\nabla\ell}^2\left(\frac{p+m}{m^2(1-(1-p)(1-\frac{m}{2}))} + \frac{1}{m^2}\right)\sum_{e=1}^{E}\sum_{t=1}^{T}\mathbb{E}\left[\left\|w_e^{(t+1)} - w_e^{(t)}\right\|^2\right]$$

$$+ 8(L_{\nabla a}^\circ)^2\sum_{e=1}^{E-1}\sum_{t=2}^{T+1}\mathbb{E}\left[\left\|\tilde{v}_e^{(t)} - \hat{v}_e^{(t)}\right\|^2\right] + 8(L_{\nabla a}')^2\sum_{e=1}^{E-1}\sum_{t=2}^{T+1}\mathbb{E}\left[\left\|\tilde{y}_e^{(t)} - \hat{y}_e^{(t)}\right\|^2\right]$$

$$+ 4T\sigma^2\frac{(2-p)m-(1-p)m^2}{1-(1-p)(1-m)^2} + \frac{3}{m}\sum_{e=1}^{E}\mathbb{E}\left[\left\|\tilde{u}_e^{(1)} - \nabla_e \ell(\boldsymbol{w}^{(1)})\right\|^2\right].$$

Thus, we finish the proof of this lemma. $\qquad\square$

### C.2 COMPRESS ERROR ANALYSIS

Here we consider the compress error of forward and backward propagation as the following lemma:

**Lemma 4** (Compress error of forward and backward propagation)**.** *Suppose Assumption 3 holds, then for $e = 1, 2, \cdots, E-1$ and $T_1 > T_0 \geq 1$ we have:*

$$\sum_{t=T_0+1}^{T_1+1}\mathbb{E}\left[\left\|\tilde{v}_e^{(t)} - v_e^{(t)}\right\|^2\right] \leq \frac{\omega_B^2}{(1-\omega_B)^2}\sum_{t=T_0}^{T_1}\mathbb{E}\left[\left\|v_e^{(t+1)} - v_e^{(t)}\right\|^2\right] + \frac{\omega_B}{1-\omega_B}\mathbb{E}\left[\left\|\tilde{v}_e^{(T_0)} - v_e^{(T_0)}\right\|^2\right], \tag{37a}$$

$$\sum_{t=T_0+1}^{T_1+1}\mathbb{E}\left[\left\|\tilde{y}_e^{(t)} - y_e^{(t)}\right\|^2\right] \leq \frac{\omega_F^2}{(1-\omega_F)^2}\sum_{t=T_0}^{T_1}\mathbb{E}\left[\left\|y_e^{(t+1)} - y_e^{(t)}\right\|^2\right] + \frac{\omega_F}{1-\omega_F}\mathbb{E}\left[\left\|\tilde{y}_e^{(T_0)} - y_e^{(T_0)}\right\|^2\right]. \tag{37b}$$

*Proof.* Firstly we consider the term $\left\|\tilde{v}_e^{(t)} - v_e^{(t)}\right\|^2$ for $e = 1, 2, \cdots, E-1$. According to Algorithm 3, we have:

$$\mathbb{E}\left[\left\|\tilde{v}_e^{(t)} - v_e^{(t)}\right\|^2\Big|\mathcal{G}_e^{(t)}\right] = \mathbb{E}\left[\left\|\tilde{v}_e^{(t-1)} + \mathcal{C}(v_e^{(t)} - \tilde{v}_e^{(t-1)}) - v_e^{(t)}\right\|^2\Big|\mathcal{G}_e^{(t)}\right] \leq \omega_B^2\left\|\tilde{v}_e^{(t-1)} - v_e^{(t)}\right\|^2$$

$$\leq \omega_B\left\|\tilde{v}_e^{(t-1)} - v_e^{(t-1)}\right\|^2 + \frac{\omega_B^2}{1-\omega_B}\left\|v_e^{(t)} - v_e^{(t-1)}\right\|^2, \tag{38}$$

where the first inequality is due to Assumption 3 and the second inequality uses Young's inequality. Then, taking expectation on both sides and then taking summation over $t = T_0 + 1, \cdots, T_1 + 1$, we have:

$$\sum_{t=T_0+1}^{T_1+1} \mathbb{E}\left[\left\|\tilde{v}_e^{(t)} - v_e^{(t)}\right\|^2\right] \le \omega_B \sum_{t=T_0}^{T_1} \mathbb{E}\left[\left\|\tilde{v}_e^{(t)} - v_e^{(t)}\right\|^2\right] + \frac{\omega_B^2}{1-\omega_B} \sum_{t=T_0}^{T_1} \mathbb{E}\left[\left\|v_e^{(t+1)} - v_e^{(t)}\right\|^2\right]. \quad (39)$$

Then, we can get:

$$\sum_{t=T_0+1}^{T_1+1} \mathbb{E}\left[\left\|\tilde{v}_e^{(t)} - v_e^{(t)}\right\|^2\right] \le \frac{\omega_B^2}{(1-\omega_B)^2} \sum_{t=T_0}^{T_1} \mathbb{E}\left[\left\|v_e^{(t+1)} - v_e^{(t)}\right\|^2\right] + \frac{\omega_B}{1-\omega_B} \mathbb{E}\left[\left\|\tilde{v}_e^{(T_0)} - v_e^{(T_0)}\right\|^2\right].$$

Thus, Eq. (37a) holds.

Next, we consider the term $\left\|\tilde{y}_e^{(t)} - y_e^{(t)}\right\|^2$ for $e = 1, 2, \cdots, E-1$. According to Algorithm 2, we have:

$$\mathbb{E}\left[\left\|\tilde{y}_e^{(t)} - y_e^{(t)}\right\|^2 \bigg| \mathcal{F}_e^{(t)}\right] = \mathbb{E}\left[\left\|\tilde{y}_e^{(t-1)} + \mathcal{C}\big(y_e^{(t)} - \tilde{y}_e^{(t-1)}\big) - y_e^{(t)}\right\|^2 \bigg| \mathcal{F}_e^{(t)}\right] \le \omega_F^2 \left\|\tilde{y}_e^{(t-1)} - y_e^{(t)}\right\|^2$$

$$\le \omega_F \left\|\tilde{y}_e^{(t-1)} - y_e^{(t-1)}\right\|^2 + \frac{\omega_F^2}{1-\omega_F} \left\|y_e^{(t)} - y_e^{(t-1)}\right\|^2, \quad (40)$$

where the first inequality is due to Assumption 3 and the second inequality uses Young's inequality. Then, taking expectation on both sides and then taking summation over $t = T_0 + 1, \cdots, T_1 + 1$, we have:

$$\sum_{t=T_0+1}^{T_1+1} \mathbb{E}\left[\left\|\tilde{y}_e^{(t)} - y_e^{(t)}\right\|^2\right] \le \omega_F \sum_{t=T_0}^{T_1} \mathbb{E}\left[\left\|\tilde{y}_e^{(t)} - y_e^{(t)}\right\|^2\right] + \frac{\omega_F^2}{1-\omega_F} \sum_{t=T_0}^{T_1} \mathbb{E}\left[\left\|y_e^{(t+1)} - y_e^{(t)}\right\|^2\right]. \quad (41)$$

Then, we can get:

$$\sum_{t=T_0+1}^{T_1+1} \mathbb{E}\left[\left\|\tilde{y}_e^{(t)} - y_e^{(t)}\right\|^2\right] \le \frac{\omega_F^2}{(1-\omega_F)^2} \sum_{t=T_0}^{T_1} \mathbb{E}\left[\left\|y_e^{(t+1)} - y_e^{(t)}\right\|^2\right] + \frac{\omega_F}{1-\omega_F} \mathbb{E}\left[\left\|\tilde{y}_e^{(T_0)} - y_e^{(T_0)}\right\|^2\right].$$

Thus, Eq. (37b) holds. $\qquad\square$

### C.3 ERROR ACCUMULATION IN FORWARD PROPAGATION

With Lemma 4, we can present the following lemma to show the analysis of the error term $\left\|\tilde{y}_e^{(t)} - \hat{y}_e^{(t)}\right\|^2$. Then we can obtain the error accumulation in forward propagation.

**Lemma 5.** *Suppose Assumption 1 and 3 holds, then for any $T_1 > T_0 \ge 1$ we have:*

$$\sum_{e=1}^{E-1} \sum_{t=T_0+1}^{T_1+1} \mathbb{E}\left[\left\|\tilde{y}_e^{(t)} - \hat{y}_e^{(t)}\right\|^2\right] \le \sum_{e=1}^{E-1} \sum_{\iota=e}^{E-1} 2(2L_a^2)^{\iota-e} \frac{\omega_F^2}{(1-\omega_F)^2} \sum_{t=T_0}^{T_1} \mathbb{E}\left[\left\|y_e^{(t+1)} - y_e^{(t)}\right\|^2\right]$$

$$+ \sum_{e=1}^{E-1} \sum_{\iota=e}^{E-1} 2(2L_a^2)^{\iota-e} \frac{\omega_F}{1-\omega_F} \mathbb{E}\left[\left\|\tilde{y}_e^{(T_0)} - y_e^{(T_0)}\right\|^2\right]. \quad (42)$$

*Proof.* For $1 \le e \le E-1$, we have:

$$\left\|\tilde{y}_e^{(t)} - \hat{y}_e^{(t)}\right\|^2 \le 2\left\|\tilde{y}_e^{(t)} - y_e^{(t)}\right\|^2 + 2\left\|y_e^{(t)} - \hat{y}_e^{(t)}\right\|^2$$

$$= 2\left\|\tilde{y}_e^{(t)} - y_e^{(t)}\right\|^2 + 2\left\|a_e(\tilde{y}_{e-1}^{(t)}, w_e^{(t)}) - a_e(\hat{y}_{e-1}^{(t)}, w_e^{(t)})\right\|^2 \quad (43)$$

$$\le 2\left\|\tilde{y}_e^{(t)} - y_e^{(t)}\right\|^2 + 2L_a^2 \left\|\tilde{y}_{e-1}^{(t)} - \hat{y}_{e-1}^{(t)}\right\|^2 \le \cdots \le \sum_{\iota=1}^{e} 2(2L_a^2)^{e-\iota} \left\|\tilde{y}_\iota^{(t)} - y_\iota^{(t)}\right\|^2,$$

where the second inequality is due to Assumption 1.

Taking expectation and then taking summation on both sides of (43) over $t = T_0 + 1, \cdots, T_1 + 1$, then we have:

$$\sum_{t=T_0+1}^{T_1+1} \mathbb{E}\left[\left\|\tilde{y}_e^{(t)} - \hat{y}_e^{(t)}\right\|^2\right] \leq \sum_{\iota=1}^{e} 2(2L_a^2)^{e-\iota} \sum_{t=T_0+1}^{T_1+1} \mathbb{E}\left[\left\|\tilde{y}_\iota^{(t)} - y_\iota^{(t)}\right\|^2\right]$$

$$\leq \sum_{\iota=1}^{e} 2(2L_a^2)^{e-\iota} \frac{\omega_F^2}{(1-\omega_F)^2} \sum_{t=T_0}^{T_1} \mathbb{E}\left[\left\|y_\iota^{(t+1)} - y_\iota^{(t)}\right\|^2\right] + \sum_{\iota=1}^{e} 2(2L_a^2)^{e-\iota} \frac{\omega_F}{1-\omega_F} \mathbb{E}\left[\left\|\tilde{y}_\iota^{(T_0)} - y_\iota^{(T_0)}\right\|^2\right],$$

$$\tag{44}$$

where the last inequality is due to the result of (37b).

Taking summation on both sides of (44) over $e$, we can get:

$$\sum_{e=1}^{E-1} \sum_{t=T_0+1}^{T_1+1} \mathbb{E}\left[\left\|\tilde{y}_e^{(t)} - \hat{y}_e^{(t)}\right\|^2\right] \leq \sum_{e=1}^{E-1} \sum_{\iota=e}^{E-1} 2(2L_a^2)^{\iota-e} \frac{\omega_F^2}{(1-\omega_F)^2} \sum_{t=T_0}^{T_1} \mathbb{E}\left[\left\|y_e^{(t+1)} - y_e^{(t)}\right\|^2\right]$$

$$+ \sum_{e=1}^{E-1} \sum_{\iota=e}^{E-1} 2(2L_a^2)^{\iota-e} \frac{\omega_F}{1-\omega_F} \mathbb{E}\left[\left\|\tilde{y}_e^{(T_0)} - y_e^{(T_0)}\right\|^2\right].$$

$\square$

Then, the following lemma shows the analysis of the term $\left\|\tilde{y}_e^{(t)} - \tilde{y}_e^{(t-1)}\right\|^2$.

**Lemma 6.** *Suppose Assumption 3 and Eq. (37b) holds, then for any $T_1 > T_0 \geq 1$ we have:*

$$\sum_{t=T_0}^{T_1} \mathbb{E}\left[\left\|\tilde{y}_e^{(t+1)} - \tilde{y}_e^{(t)}\right\|^2\right] \leq \frac{8}{(1-\omega_F)^2} \sum_{t=T_0}^{T_1} \mathbb{E}\left[\left\|y_e^{(t+1)} - y_e^{(t)}\right\|^2\right] + \frac{8}{1-\omega_F} \mathbb{E}\left[\left\|\tilde{y}_e^{(T_0)} - y_e^{(T_0)}\right\|^2\right].$$

$$\tag{45}$$

*Proof.* For $t = T_0 + 1, \cdots, T_1 + 1$, we have:

$$\mathbb{E}\left[\left\|\tilde{y}_e^{(t)} - \tilde{y}_e^{(t-1)}\right\|^2 \Big| \mathcal{F}^{(t)}\right] = \mathbb{E}\left[\left\|\mathcal{C}\left(y_e^{(t)} - \tilde{y}_e^{(t-1)}\right)\right\|^2 \Big| \mathcal{F}^{(t)}\right]$$

$$\leq \left(1 + \frac{1}{\omega_F}\right) \mathbb{E}\left[\left\|\mathcal{C}\left(y_e^{(t)} - \tilde{y}_e^{(t-1)}\right) - \left(y_e^{(t)} - \tilde{y}_e^{(t-1)}\right)\right\|^2 \Big| \mathcal{F}^{(t)}\right] + (1 + \omega_F)\left\|y_e^{(t)} - \tilde{y}_e^{(t-1)}\right\|^2 \tag{46}$$

$$\leq (1 + \omega_F)^2 \left\|y_e^{(t)} - \tilde{y}_e^{(t-1)}\right\|^2 \leq 8\left\|y_e^{(t)} - y_e^{(t-1)}\right\|^2 + 8\left\|y_e^{(t-1)} - \tilde{y}_e^{(t-1)}\right\|^2,$$

where the first inequality is due to Young's inequality, the second inequality is due to Assumption 3, and the last inequality is due to $\omega_F \in [0, 1)$ in Assumption 3.

Then, taking expectation and taking summation on both sides of (46) over $t = T_0 + 1, \cdots, T_1 + 1$, we have:

$$\sum_{t=T_0}^{T_1} \mathbb{E}\left[\left\|\tilde{y}_e^{(t+1)} - \tilde{y}_e^{(t)}\right\|^2\right] \leq 8 \sum_{t=T_0}^{T_1} \mathbb{E}\left[\left\|y_e^{(t+1)} - y_e^{(t)}\right\|^2\right] + 8 \sum_{t=T_0}^{T_1} \mathbb{E}\left[\left\|y_e^{(t)} - \tilde{y}_e^{(t)}\right\|^2\right]$$

$$\leq \frac{8}{(1-\omega_F)^2} \sum_{t=T_0}^{T_1} \mathbb{E}\left[\left\|y_e^{(t+1)} - y_e^{(t)}\right\|^2\right] + \frac{8}{1-\omega_F} \mathbb{E}\left[\left\|\tilde{y}_e^{(T_0)} - y_e^{(T_0)}\right\|^2\right],$$

where the last inequality is due to Eq. (37b) and the fact that $1 + \frac{\omega_F^2}{(1-\omega_F)^2} \leq \frac{1}{(1-\omega_F)^2}$ as $\omega_F \in [0, 1)$. $\square$

## C.4 Convergence rate of the case $E = 2$

We firstly consider the proof of the case $E = 2$ for Clapping-**FU** . In that case, there is once communication in both forward and backward propagation. Nevertheless, the analysis of the error accumulation and propagation in backward is not complex in this case. Thus, the proof of that case is more simple than the general case but can show how error feedback and lazy sampling benefit the convergence.

**Lemma 7** (Convergence rate of Clapping-**FU** in the case $E = 2$). *Suppose Assumption 1, 2, and 3 hold. Then for* Clapping-**FU** *, there exist $\gamma, m > 0$ such that:*

$$\frac{1}{T} \sum_{t=1}^{T} \mathbb{E}\left[\left\|\nabla \ell(\boldsymbol{w}^{(t)})\right\|^2\right] \lesssim \frac{\sigma}{\sqrt{T}} + \frac{1}{T(1-\omega_B)(1-\omega_F)}. \tag{47}$$

*Proof.* Substituting $E = 2$ into (13), we have:

$$\sum_{t=1}^{T+1} \sum_{e=1}^{2} \mathbb{E}\left[\left\|\tilde{u}_e^{(t)} - \nabla_e \ell(\boldsymbol{w}^{(t)})\right\|^2\right]$$

$$\leq 32 L_{\nabla\ell}^2 \left(\frac{p+m}{m^2(1-(1-p)(1-\frac{m}{2}))} + \frac{1}{m^2}\right) \sum_{t=1}^{T} \sum_{e=1}^{2} \mathbb{E}\left[\left\|w_e^{(t+1)} - w_e^{(t)}\right\|^2\right]$$

$$+ 8(L_{\nabla a}^{\circ})^2 \sum_{t=2}^{T+1} \mathbb{E}\left[\left\|\tilde{v}_1^{(t)} - \hat{v}_1^{(t)}\right\|^2\right] + 8(L_{\nabla a}')^2 \sum_{t=2}^{T+1} \mathbb{E}\left[\left\|\tilde{y}_1^{(t)} - \hat{y}_1^{(t)}\right\|^2\right] \tag{48}$$

$$+ 4T\sigma^2 \frac{(2-p)m - (1-p)m^2}{1-(1-p)(1-m)^2} + \frac{3}{m} \sum_{e=1}^{2} \mathbb{E}\left[\left\|\tilde{u}_e^{(1)} - \nabla_e \ell(\boldsymbol{w}^{(1)})\right\|^2\right].$$

For the first $T - 1$ iterations, suppose $1 = Q_1 < Q_2 < \cdots < Q_{r_0} \leq T$ are the all moments at which the sample is randomly obtained from $\mathcal{D}$. We also denote $Q_{r_0+1} = T + 1$.

For any $1 \leq r \leq r_0$, we note that $\tilde{y}_1^{(Q_r)} = y_1^{(Q_r)} = \hat{y}_1^{(Q_r)}$ as we do not compress the activation and gradients in Clapping-**FU** . Thus, substituting $E = 2$ into (42), we can get:

$$\sum_{t=Q_r+1}^{Q_{r+1}} \mathbb{E}\left[\left\|\tilde{y}_1^{(t)} - \hat{y}_1^{(t)}\right\|^2\right] = \sum_{t=Q_r+1}^{Q_{r+1}-1} \mathbb{E}\left[\left\|\tilde{y}_1^{(t)} - \hat{y}_1^{(t)}\right\|^2\right] \leq C_y^2 \frac{\omega_F^2}{(1-\omega_F)^2} \sum_{t=Q_r}^{Q_{r+1}-2} \mathbb{E}\left[\left\|y_1^{(t+1)} - y_1^{(t)}\right\|^2\right]. \tag{49}$$

Taking summation over $r = 1, 2, \cdots, r_0$, it holds that:

$$\sum_{t=2}^{T+1} \mathbb{E}\left[\left\|\tilde{y}_1^{(t)} - \hat{y}_1^{(t)}\right\|^2\right] = \sum_{r=1}^{r_0} \sum_{t=Q_r+1}^{Q_{r+1}} \mathbb{E}\left[\left\|\tilde{y}_1^{(t)} - \hat{y}_1^{(t)}\right\|^2\right]$$

$$\leq C_y^2 \frac{\omega_F^2}{(1-\omega_F)^2} \sum_{r=1}^{r_0} \sum_{t=Q_r}^{Q_{r+1}-2} \mathbb{E}\left[\left\|y_1^{(t+1)} - y_1^{(t)}\right\|^2\right]. \tag{50}$$

Then, use the fact that $\nabla a_2$ is Lipschitz continous, we have:

$$\left\|v_1^{(t)} - \hat{v}_1^{(t)}\right\|^2 = \left\|\nabla a_2(\tilde{y}_1^{(t)}, w_2^{(t)}) - \nabla a_2(\hat{y}_1^{(t)}, w_2^{(t)})\right\|^2 \leq L_{\nabla a}^2 \left\|\tilde{y}_1^{(t)} - \hat{y}_1^{(t)}\right\|^2. \tag{51}$$

From $\tilde{v}_1^{(Q_r)} = v_1^{(Q_r)} = \hat{v}_1^{(Q_r)}$, we can obtain that:

$$\sum_{t=Q_r+1}^{Q_{r+1}} \mathbb{E}\left[\left\|\tilde{v}_1^{(t)} - \hat{v}_1^{(t)}\right\|^2\right] \leq 2 \sum_{t=Q_r+1}^{Q_{r+1}} \mathbb{E}\left[\left\|\tilde{v}_1^{(t)} - v_1^{(t)}\right\|^2\right] + 2 \sum_{t=Q_r+1}^{Q_{r+1}} \mathbb{E}\left[\left\|v_1^{(t)} - \hat{v}_1^{(t)}\right\|^2\right]$$

$$= 2 \sum_{t=Q_r+1}^{Q_{r+1}-1} \mathbb{E}\left[\left\|\tilde{v}_1^{(t)} - v_1^{(t)}\right\|^2\right] + 2 \sum_{t=Q_r+1}^{Q_{r+1}-1} \mathbb{E}\left[\left\|v_1^{(t)} - \hat{v}_1^{(t)}\right\|^2\right] \tag{52}$$

$$\leq 2 \frac{\omega_B^2}{(1-\omega_B)^2} \sum_{t=Q_r}^{Q_{r+1}-2} \mathbb{E}\left[\left\|v_1^{(t+1)} - v_1^{(t)}\right\|^2\right] + 2L_{\nabla a}^2 \sum_{t=Q_r+1}^{Q_{r+1}-1} \mathbb{E}\left[\left\|\tilde{y}_1^{(t)} - \hat{y}_1^{(t)}\right\|^2\right],$$

where the second inequality is due to (37a) and (51). Taking summation over $r$, it holds that:

$$\sum_{t=2}^{T+1} \mathbb{E}\left[\left\|\tilde{v}_1^{(t)} - \hat{v}_1^{(t)}\right\|^2\right]$$

$$\leq 2 \frac{\omega_B^2}{(1-\omega_B)^2} \sum_{r=1}^{r_0} \sum_{t=Q_r}^{Q_{r+1}-2} \mathbb{E}\left[\left\|v_1^{(t+1)} - v_1^{(t)}\right\|^2\right] + 2L_{\nabla a}^2 \sum_{t=2}^{T+1} \mathbb{E}\left[\left\|\tilde{y}_1^{(t)} - \hat{y}_1^{(t)}\right\|^2\right], \tag{53}$$

Plugging Eq. (50) and Eq. (53) into Eq. (48), it holds that:

$$\sum_{t=1}^{T+1} \sum_{e=1}^{2} \mathbb{E}\left[\left\|\tilde{u}_e^{(t)} - \nabla_e \ell(\boldsymbol{w}^{(t)})\right\|^2\right]$$

$$\leq 32 L_{\nabla \ell}^2 \left(\frac{p+m}{m^2(1-(1-p)(1-\frac{m}{2}))} + \frac{1}{m^2}\right) \sum_{t=1}^{T} \sum_{e=1}^{2} \mathbb{E}\left[\left\|w_e^{(t+1)} - w_e^{(t)}\right\|^2\right]$$

$$+ 16(L_{\nabla a}^\circ)^2 \frac{\omega_B^2}{(1-\omega_B)^2} \sum_{r=1}^{r_0} \sum_{t=Q_r}^{Q_{r+1}-2} \mathbb{E}\left[\left\|v_1^{(t+1)} - v_1^{(t)}\right\|^2\right] \qquad (54)$$

$$+ \left(8(L_{\nabla a}')^2 + 16(L_{\nabla a}^\circ)^2 L_{\nabla a}^2\right) C_y^2 \frac{\omega_F^2}{(1-\omega_F)^2} \sum_{r=1}^{r_0} \sum_{t=Q_r}^{Q_{r+1}-2} \mathbb{E}\left[\left\|y_1^{(t+1)} - y_1^{(t)}\right\|^2\right]$$

$$+ 4T\sigma^2 \frac{(2-p)m - (1-p)m^2}{1-(1-p)(1-m)^2} + \frac{3}{m} \sum_{e=1}^{2} \mathbb{E}\left[\left\|\tilde{u}_e^{(1)} - \nabla_e \ell(\boldsymbol{w}^{(1)})\right\|^2\right].$$

Note that $E = 2$, thus for any $1 \leq r \leq r_0$ we have:

$$\sum_{t=Q_r}^{Q_{r+1}-2} \mathbb{E}\left[\left\|v_1^{(t+1)} - v_1^{(t)}\right\|^2\right] = \sum_{t=Q_r}^{Q_{r+1}-2} \mathbb{E}\left[\left\|\nabla_1 a_2(\tilde{y}_1^{(t+1)}, w_2^{(t+1)}) - \nabla_1 a_2(\tilde{y}_1^{(t)}, w_2^{(t)})\right\|^2\right]$$

$$\leq L_{\nabla a}^2 \sum_{t=Q_r}^{Q_{r+1}-2} \mathbb{E}\left[\left\|\tilde{y}_1^{(t+1)} - \tilde{y}_1^{(t)}\right\|^2 + \left\|w_2^{(t+1)} - w_2^{(t)}\right\|^2\right] \qquad (55)$$

$$\leq L_{\nabla a}^2 \frac{8}{(1-\omega_F)^2} \sum_{t=Q_r}^{Q_{r+1}-2} \mathbb{E}\left[\left\|y_1^{(t+1)} - y_1^{(t)}\right\|^2\right] + L_{\nabla a}^2 \sum_{t=Q_r}^{Q_{r+1}-2} \mathbb{E}\left[\left\|w_2^{(t+1)} - w_2^{(t)}\right\|^2\right],$$

where the first inequality is due to Assumption 1 and the second inequality is due to (45).

Taking summation over $r$, it holds that:

$$\sum_{r=1}^{r_0} \sum_{t=Q_r}^{Q_{r+1}-2} \mathbb{E}\left[\left\|v_1^{(t+1)} - v_1^{(t)}\right\|^2\right]$$

$$\leq L_{\nabla a}^2 \frac{8}{(1-\omega_F)^2} \sum_{r=1}^{r_0} \sum_{t=Q_r}^{Q_{r+1}-2} \mathbb{E}\left[\left\|y_1^{(t+1)} - y_1^{(t)}\right\|^2\right] + L_{\nabla a}^2 \sum_{t=1}^{T} \mathbb{E}\left[\left\|w_2^{(t+1)} - w_2^{(t)}\right\|^2\right], \qquad (56)$$

Then as $x^{(t+1)} = x^{(t)}$ for any $t = Q_r, \cdots, Q_{r+1} - 2$, we also have

$$\sum_{t=Q_r}^{Q_{r+1}-2} \mathbb{E}\left[\left\|y_1^{(t+1)} - y_1^{(t)}\right\|^2\right] = \sum_{t=Q_r}^{Q_{r+1}-2} \mathbb{E}\left[\left\|\hat{y}_1^{(t+1)} - \hat{y}_1^{(t)}\right\|^2\right]$$

$$= \sum_{t=Q_r}^{Q_{r+1}-2} \mathbb{E}\left[\left\|a_1(x^{(t+1)}, w_1^{(t+1)}) - a_1(x^{(t)}, w_1^{(t)})\right\|^2\right] \leq L_a^2 \sum_{t=Q_r}^{Q_{r+1}-2} \mathbb{E}\left[\left\|w_1^{(t+1)} - w_1^{(t)}\right\|^2\right], \qquad (57)$$

where the inequality is due to Assumption 1.

Taking summation over $r$, it holds that:

$$\sum_{r=1}^{r_0} \sum_{t=Q_r}^{Q_{r+1}-2} \mathbb{E}\left[\left\|y_1^{(t+1)} - y_1^{(t)}\right\|^2\right] \leq L_a^2 \sum_{t=1}^{T} \mathbb{E}\left[\left\|w^{(t+1)} - w^{(t)}\right\|^2\right], \qquad (58)$$

Plugging (56) and (58) into (54), we can get:

$$\sum_{t=1}^{T+1}\sum_{e=1}^{2}\mathbb{E}\left[\left\|\tilde{u}_e^{(t)}-\nabla_e\ell(\boldsymbol{w}^{(t)})\right\|^2\right]$$

$$\leq 32L_{\nabla\ell}^2\left(\frac{p+m}{m^2(1-(1-p)(1-\frac{m}{2}))}+\frac{1}{m^2}\right)\sum_{t=1}^{T}\sum_{e=1}^{2}\mathbb{E}\left[\left\|w_e^{(t+1)}-w_e^{(t)}\right\|^2\right]$$

$$+16L_{\nabla_a}^2(L_{\nabla a}^{\circ})^2\frac{\omega_B^2}{(1-\omega_B)^2}\sum_{t=1}^{T}\mathbb{E}\left[\left\|w_2^{(t+1)}-w_2^{(t)}\right\|^2\right]$$

$$+\left(\left(8(L_{\nabla a}')^2+16(L_{\nabla a}^{\circ})^2L_{\nabla a}^2\right)C_y^2\frac{\omega_F^2}{(1-\omega_F)^2}+128L_{\nabla_a}^2(L_{\nabla a}^{\circ})^2\frac{\omega_B^2}{(1-\omega_B)^2(1-\omega_F)^2}\right)$$

$$\cdot L_a^2\sum_{t=1}^{T}\mathbb{E}\left[\left\|w^{(t+1)}-w^{(t)}\right\|^2\right]$$

$$+4T\sigma^2\frac{(2-p)m-(1-p)m^2}{1-(1-p)(1-m)^2}+\frac{3}{m}\sum_{e=1}^{2}\mathbb{E}\left[\left\|\tilde{u}_e^{(1)}-\nabla_e\ell(\boldsymbol{w}^{(1)})\right\|^2\right]$$

$$\leq C_w\sum_{t=1}^{T}\mathbb{E}\left[\left\|\boldsymbol{w}^{(t+1)}-\boldsymbol{w}^{(t)}\right\|^2\right]+4T\sigma^2\frac{(2-p)m-(1-p)m^2}{1-(1-p)(1-m)^2}+\frac{3}{m}\sum_{e=1}^{2}\mathbb{E}\left[\left\|\tilde{u}_e^{(1)}-\nabla_e\ell(\boldsymbol{w}^{(1)})\right\|^2\right].$$

$$(59)$$

where:

$$C_w=32L_{\nabla\ell}^2\left(\frac{p+m}{m^2(1-(1-p)(1-\frac{m}{2}))}+\frac{1}{m^2}\right)+16L_{\nabla_a}^2(L_{\nabla a}^{\circ})^2\frac{\omega_B^2}{(1-\omega_B)^2}$$

$$+\left(8(L_{\nabla a}')^2+16(L_{\nabla a}^{\circ})^2L_{\nabla a}^2\right)L_a^2C_y^2\frac{\omega_F^2}{(1-\omega_F)^2}+128L_{\nabla_a}^2(L_{\nabla a}^{\circ})^2L_a^2\frac{\omega_B^2}{(1-\omega_B)^2(1-\omega_F)^2}.$$

Plugging $E=2$ into (8), combining it with (59), and then taken $p=m$, we can get

$$\frac{1}{T}\sum_{t=1}^{T}\mathbb{E}\left[\left\|\nabla\ell(\boldsymbol{w}^{(t)})\right\|^2\right]$$

$$\leq\frac{2}{\gamma T}\mathbb{E}\left[\ell(\boldsymbol{w}^{(1)})-\inf_{\boldsymbol{w}}\ell(\boldsymbol{w})\right]+\frac{1}{T}\left(C_w-\frac{1}{2\gamma^2}\right)\sum_{t=1}^{T}\mathbb{E}\left[\left\|\boldsymbol{w}^{(t+1)}-\boldsymbol{w}^{(t)}\right\|^2\right] \qquad (60)$$

$$+4\sigma^2\frac{(2-p)m-(1-p)m^2}{1-(1-p)(1-m)^2}+\frac{3}{mT}\sum_{e=1}^{2}\mathbb{E}\left[\left\|\tilde{u}_e^{(1)}-\nabla_e\ell(\boldsymbol{w}^{(1)})\right\|^2\right].$$

Let $p=p_0$ as a constant with the order of $\mathcal{O}(1)$, and let:

$$m\sim\left(\frac{1}{(1-\omega_B)(1-\omega_F)}+\sigma\sqrt{T}\right)^{-1},\quad \gamma\sim\left(\frac{1}{(1-\omega_B)(1-\omega_F)}+\sigma\sqrt{T}\right)^{-1}\text{ and }m,\gamma\leq 1.$$

Then, as $\gamma$ has the same order as $m$ with respect to $\omega_B,\omega_F,\sigma,T$, it holds that $C_w-\frac{1}{2\gamma^2}\leq 0$ if $\gamma/m$ is sufficiently small.

Thus, we have:

$$\frac{1}{T}\sum_{t=1}^{T}\mathbb{E}\left[\left\|\nabla\ell(\boldsymbol{w}^{(t)})\right\|^2\right]\lesssim\frac{\sigma}{\sqrt{T}}+\frac{1}{T(1-\omega_B)(1-\omega_F)}.$$

$\square$

## C.5 ERROR ACCUMULATION IN BACKWARD PROPAGATION

Here, we analysis the error accumulation and propagation in backward propagation. In the beginning, we present a lemma that shows the error analysis of $\left\|\tilde{v}_e^{(t)}-\hat{v}_e^{(t)}\right\|^2$.

**Lemma 8.** *Suppose Assumption 1 and 3 holds, then for any $T_1 > T_0 \geq 1$ and $e = 1, 2, \cdots, E - 1$ we have:*

$$\sum_{t=T_0+1}^{T_1+1} \mathbb{E}\left[\left\|\tilde{v}_e^{(t)} - \hat{v}_e^{(t)}\right\|^2\right] \leq 2\frac{\omega_B^2}{(1-\omega_B)^2} \sum_{\iota=e}^{E-1} (2(L_{\nabla a}^\circ)^2)^{\iota-e} \sum_{t=T_0}^{T_1} \mathbb{E}\left[\left\|v_\iota^{(t+1)} - v_\iota^{(t)}\right\|^2\right]$$

$$+ (2(L_{\nabla a}')^2) \sum_{\iota=e}^{E-1} (2(L_{\nabla a}^\circ)^2)^{\iota-e} \sum_{t=T_0+1}^{T_1+1} \mathbb{E}\left[\left\|\tilde{y}_\iota^{(t)} - \hat{y}_\iota^{(t)}\right\|^2\right] + 2\frac{\omega_B}{1-\omega_B} \sum_{\iota=e}^{E-1} (2(L_{\nabla a}^\circ)^2)^{\iota-e} \mathbb{E}\left[\left\|\tilde{v}_\iota^{(T_0)} - v_\iota^{(T_0)}\right\|^2\right]. \tag{61}$$

*Proof.* For $1 \leq e \leq E - 1$, we have:

$$\left\|\tilde{v}_e^{(t)} - \hat{v}_e^{(t)}\right\|^2 \leq 2\left\|\tilde{v}_e^{(t)} - v_e^{(t)}\right\|^2 + 2\left\|v_e^{(t)} - \hat{v}_e^{(t)}\right\|^2$$

$$= 2\left\|\tilde{v}_e^{(t)} - v_e^{(t)}\right\|^2 + 2\left\|\nabla_1 a_{e+1}(\tilde{y}_e^{(t)}, w_{e+1}^{(t)})^\top \tilde{v}_{e+1}^{(t)} - \nabla_1 a_{e+1}(\hat{y}_e^{(t)}, w_{e+1}^{(t)})^\top \hat{v}_{e+1}^{(t)}\right\|^2 \tag{62}$$

$$\leq 2\left\|\tilde{v}_e^{(t)} - v_e^{(t)}\right\|^2 + 2(L_{\nabla a}')^2\left\|\tilde{y}_e^{(t)} - \hat{y}_e^{(t)}\right\|^2 + 2(L_{\nabla a}^\circ)^2\left\|\tilde{v}_{e+1}^{(t)} - \hat{v}_{e+1}^{(t)}\right\|^2,$$

where the second inequality is due to (12). Then, we can get:

$$\left\|\tilde{v}_e^{(t)} - \hat{v}_e^{(t)}\right\|^2 \leq \sum_{\iota=e}^{E-1} 2(2(L_{\nabla a}^\circ)^2)^{\iota-e}\left[\left\|\tilde{v}_\iota^{(t)} - v_\iota^{(t)}\right\|^2 + (L_{\nabla a}')^2\left\|\tilde{y}_\iota^{(t)} - \hat{y}_\iota^{(t)}\right\|^2\right]$$

$$\leq 2\sum_{\iota=e}^{E-1} (2(L_{\nabla a}^\circ)^2)^{\iota-e}\left\|\tilde{v}_\iota^{(t)} - v_\iota^{(t)}\right\|^2 + (2(L_{\nabla a}')^2) \sum_{\iota=e}^{E-1} (2(L_{\nabla a}^\circ)^2)^{\iota-e}\left\|\tilde{y}_\iota^{(t)} - \hat{y}_\iota^{(t)}\right\|^2. \tag{63}$$

Taking expectation and then taking summation on both sides of (63) over $t = T_0 + 1, \cdots, T_1 + 1$, then we gan get

$$\sum_{t=T_0+1}^{T_1+1} \mathbb{E}\left[\left\|\tilde{v}_e^{(t)} - \hat{v}_e^{(t)}\right\|^2\right]$$

$$\leq 2\sum_{\iota=e}^{E-1} (2(L_{\nabla a}^\circ)^2)^{\iota-e} \sum_{t=T_0+1}^{T_1+1} \mathbb{E}\left[\left\|\tilde{v}_\iota^{(t)} - v_\iota^{(t)}\right\|^2\right] + (2(L_{\nabla a}')^2) \sum_{\iota=e}^{E-1} (2(L_{\nabla a}^\circ)^2)^{\iota-e} \sum_{t=T_0+1}^{T_1+1} \mathbb{E}\left[\left\|\tilde{y}_\iota^{(t)} - \hat{y}_\iota^{(t)}\right\|^2\right]$$

$$\leq 2\frac{\omega_B^2}{(1-\omega_B)^2} \sum_{\iota=e}^{E-1} (2(L_{\nabla a}^\circ)^2)^{\iota-e} \sum_{t=T_0}^{T_1} \mathbb{E}\left[\left\|v_\iota^{(t+1)} - v_\iota^{(t)}\right\|^2\right] + 2\frac{\omega_B}{1-\omega_B} \sum_{\iota=e}^{E-1} (2(L_{\nabla a}^\circ)^2)^{\iota-e} \mathbb{E}\left[\left\|\tilde{v}_\iota^{(T_0)} - v_\iota^{(T_0)}\right\|^2\right]$$

$$+ (2(L_{\nabla a}')^2) \sum_{\iota=e}^{E-1} (2(L_{\nabla a}^\circ)^2)^{\iota-e} \sum_{t=T_0+1}^{T_1+1} \mathbb{E}\left[\left\|\tilde{y}_\iota^{(t)} - \hat{y}_\iota^{(t)}\right\|^2\right], \tag{64}$$

where the last inequality is due to Eq. (37a). $\square$

Eq. (61) suggest that we can analysis the error term $\left\|v_e^{(t+1)} - v_e^{(t)}\right\|^2$. Thus, we consider the following lemma:

**Lemma 9.** *Suppose Assumption 1 and (45) hold, then for any $T_1 > T_0 \geq 1$ and $e = 1, 2, \cdots, E - 1$ we have:*

$$\sum_{t=T_0}^{T_1} \mathbb{E}\left[\left\|v_e^{(t+1)} - v_e^{(t)}\right\|^2\right] \leq 5(L_{\nabla a}^\circ)^2 \sum_{t=T_0+1}^{T_1+1} \mathbb{E}\left[\left\|\tilde{v}_{e+1}^{(t)} - \hat{v}_{e+1}^{(t)}\right\|^2\right] + 5(L_{\nabla a}^\circ)^2 \sum_{t=T_0}^{T_1} \mathbb{E}\left[\left\|\hat{v}_{e+1}^{(t+1)} - \hat{v}_{e+1}^{(t)}\right\|^2\right]$$

$$+ 5(L_{\nabla a}')^2 \sum_{t=T_0}^{T_1} \mathbb{E}\left[\left\|w_{e+1}^{(t+1)} - w_{e+1}^{(t)}\right\|^2\right] + \frac{40(L_{\nabla a}')^2}{(1-\omega_F)^2} \sum_{t=T_0}^{T_1} \mathbb{E}\left[\left\|y_e^{(t+1)} - y_e^{(t)}\right\|^2\right] \tag{65}$$

$$+ \frac{40(L_{\nabla a}')^2}{1-\omega_F} \mathbb{E}\left[\left\|\tilde{y}_e^{(T_0)} - y_e^{(T_0)}\right\|^2\right] + \frac{5}{2}(L_{\nabla a}^\circ)^2 \mathbb{E}\left[\left\|\tilde{v}_{e+1}^{(T_0)} - \hat{v}_{e+1}^{(T_0)}\right\|^2\right].$$

*Proof.* Firstly, we consider the case of $e = 1, \cdots, E - 1$, we have:

$$\sum_{t=T_0}^{T_1} \mathbb{E}\left[\left\|v_e^{(t+1)} - v_e^{(t)}\right\|^2\right]$$

$$= \sum_{t=T_0}^{T_1} \mathbb{E}\left[\left\|\nabla_1 a_{e+1}(\tilde{y}_e^{(t+1)}, w_{e+1}^{(t+1)})^\top \tilde{v}_{e+1}^{(t+1)} - \nabla_1 a_{e+1}(\tilde{y}_e^{(t)}, w_{e+1}^{(t)})^\top \tilde{v}_{e+1}^{(t)}\right\|^2\right]$$

$$\leq \frac{5}{2} \sum_{t=T_0}^{T_1} \mathbb{E}\left[\left\|\nabla_1 a_{e+1}(\tilde{y}_e^{(t+1)}, w_{e+1}^{(t+1)})^\top \tilde{v}_{e+1}^{(t+1)} - \nabla_1 a_{e+1}(\tilde{y}_e^{(t+1)}, w_{e+1}^{(t+1)})^\top \hat{v}_{e+1}^{(t+1)}\right\|^2\right] \qquad (66)$$

$$+ \frac{5}{2} \sum_{t=T_0}^{T_1} \mathbb{E}\left[\left\|\nabla_1 a_{e+1}(\tilde{y}_e^{(t)}, w_{e+1}^{(t)})^\top \tilde{v}_{e+1}^{(t)} - \nabla_1 a_{e+1}(\tilde{y}_e^{(t)}, w_{e+1}^{(t)})^\top \hat{v}_{e+1}^{(t)}\right\|^2\right]$$

$$+ 5 \sum_{t=T_0}^{T_1} \mathbb{E}\left[\left\|\nabla_1 a_{e+1}(\tilde{y}_e^{(t+1)}, w_{e+1}^{(t+1)})^\top \hat{v}_{e+1}^{(t+1)} - \nabla_1 a_{e+1}(\tilde{y}_e^{(t)}, w_{e+1}^{(t)})^\top \hat{v}_{e+1}^{(t)}\right\|^2\right].$$

Then, we have:

$$\sum_{t=T_0}^{T_1} \mathbb{E}\left[\left\|v_e^{(t+1)} - v_e^{(t)}\right\|^2\right]$$

$$\leq \frac{5}{2}(L_{\nabla a}^\circ)^2 \sum_{t=T_0}^{T_1} \mathbb{E}\left[\left\|\tilde{v}_{e+1}^{(t+1)} - \hat{v}_{e+1}^{(t+1)}\right\|^2\right] + \frac{5}{2}(L_{\nabla a}^\circ)^2 \sum_{t=T_0}^{T_1} \mathbb{E}\left[\left\|\tilde{v}_{e+1}^{(t)} - \hat{v}_{e+1}^{(t)}\right\|^2\right]$$

$$+ 5(L_{\nabla a}^\circ)^2 \sum_{t=T_0}^{T_1} \mathbb{E}\left[\left\|\hat{v}_{e+1}^{(t+1)} - \hat{v}_{e+1}^{(t)}\right\|^2\right] + 5(L_{\nabla a}')^2 \sum_{t=T_0}^{T_1} \left(\mathbb{E}\left[\left\|\tilde{y}_e^{(t+1)} - \tilde{y}_e^{(t)}\right\|^2\right] + \mathbb{E}\left[\left\|w_{e+1}^{(t+1)} - w_{e+1}^{(t)}\right\|^2\right]\right)$$

$$\leq 5(L_{\nabla a}^\circ)^2 \sum_{t=T_0+1}^{T_1+1} \mathbb{E}\left[\left\|\tilde{v}_{e+1}^{(t)} - \hat{v}_{e+1}^{(t)}\right\|^2\right] + 5(L_{\nabla a}^\circ)^2 \sum_{t=T_0}^{T_1} \mathbb{E}\left[\left\|\hat{v}_{e+1}^{(t+1)} - \hat{v}_{e+1}^{(t)}\right\|^2\right]$$

$$+ 5(L_{\nabla a}')^2 \sum_{t=T_0}^{T_1} \mathbb{E}\left[\left\|w_{e+1}^{(t+1)} - w_{e+1}^{(t)}\right\|^2\right] + \frac{40(L_{\nabla a}')^2}{(1 - \omega_F)^2} \sum_{t=T_0}^{T_1} \mathbb{E}\left[\left\|y_e^{(t+1)} - y_e^{(t)}\right\|^2\right]$$

$$+ \frac{40(L_{\nabla a}')^2}{1 - \omega_F} \mathbb{E}\left[\left\|\tilde{y}_e^{(T_0)} - y_e^{(T_0)}\right\|^2\right] + \frac{5}{2}(L_{\nabla a}^\circ)^2 \mathbb{E}\left[\left\|\tilde{v}_{e+1}^{(T_0)} - \hat{v}_{e+1}^{(T_0)}\right\|^2\right],$$

$$(67)$$

where the first inequality is due to Eq. (12) and the last inequality is due to Eq. (45). □

The following lemma combines the result of Lemma 8 and Lemma 9 together and give a further analysis of $\left\|\tilde{v}_e^{(t)} - \hat{v}_e^{(t)}\right\|^2$.

**Lemma 10.** *Suppose Lemma 8 and Lemma 9 are all satisfied, then there exist coefficients* $C_{y,e}^\circ, C_{W,e}^\circ, C_{v,e}^\circ, C_{v,e}^1, C_{v,e}^2, C_{\theta,e}^\circ, C_{\theta,e}^1 \geq 0$ *(which has been defined by Eq. (72)) for each* $e = 1, 2, \cdots, E - 1$ *and* $T_1 > T_0 \geq 1$ *such that:*

$$\sum_{e=1}^{E-1} \sum_{t=T_0+1}^{T_1+1} \mathbb{E}\left[\left\|\tilde{v}_e^{(t)} - \hat{v}_e^{(t)}\right\|^2\right]$$

$$\leq \sum_{e=1}^{E-1} C_{y,e}^\circ \sum_{t=T_0+1}^{T_1+1} \mathbb{E}\left[\left\|\tilde{y}_e^{(t)} - \hat{y}_e^{(t)}\right\|^2\right] + \sum_{e=1}^{E} C_{W,e}^\circ \sum_{t=T_0}^{T_1} \mathbb{E}\left[\left\|w_e^{(t+1)} - w_e^{(t)}\right\|^2\right]$$

$$+ \sum_{e=1}^{E-1} C_{v,e}^\circ \sum_{t=T_0}^{T_1} \mathbb{E}\left[\left\|\hat{v}_e^{(t+1)} - \hat{v}_e^{(t)}\right\|^2\right] + \sum_{e=1}^{E-1} C_{\theta,e}^\circ \sum_{t=T_0}^{T_1} \mathbb{E}\left[\left\|y_e^{(t+1)} - y_e^{(t)}\right\|^2\right] + \sum_{e=1}^{E-1} C_{\theta,e}^1 \mathbb{E}\left[\left\|\tilde{y}_e^{(T_0)} - y_e^{(T_0)}\right\|^2\right]$$

$$+ \sum_{e=1}^{E-1} C_{v,e}^1 \mathbb{E}\left[\left\|\tilde{v}_e^{(T_0)} - \hat{v}_e^{(T_0)}\right\|^2\right] + \sum_{e=1}^{E-1} C_{v,e}^2 \mathbb{E}\left[\left\|\tilde{v}_e^{(T_0)} - v_e^{(T_0)}\right\|^2\right].$$

$$(68)$$

*Proof.* Combing (61) and (65) together, we have:

$$\sum_{t=T_0+1}^{T_1+1} \mathbb{E}\left[\left\|\tilde{v}_e^{(t)} - \hat{v}_e^{(t)}\right\|^2\right]$$

$$\leq \frac{5(L_{\nabla a}^*)^2 \omega_B^2}{(1-\omega_B)^2} \sum_{\iota=e}^{E-1} \sum_{t=T_0+1}^{T_1+1} \mathbb{E}\left[\left\|\tilde{v}_{\iota+1}^{(t)} - \hat{v}_{\iota+1}^{(t)}\right\|^2\right] + \frac{5(L_{\nabla a}^*)^2 \omega_B^2}{(1-\omega_B)^2} \sum_{\iota=e}^{E-1} \sum_{t=T_0}^{T_1} \mathbb{E}\left[\left\|\hat{v}_{\iota+1}^{(t+1)} - \hat{v}_{\iota+1}^{(t)}\right\|^2\right]$$

$$+ (L_{\nabla a}^*)^2 (2(L_{\nabla a}')^2) \sum_{\iota=e}^{E-1} \sum_{t=T_0+1}^{T_1+1} \mathbb{E}\left[\left\|\tilde{y}_\iota^{(t)} - \hat{y}_\iota^{(t)}\right\|^2\right] + \frac{20(L_{\nabla a}^*)^2 (L_{\nabla a}')^2 \omega_B^2}{(1-\omega_B)^2} \sum_{\iota=e}^{E-1} \sum_{t=T_0}^{T_1} \mathbb{E}\left[\left\|w_{\iota+1}^{(t+1)} - w_{\iota+1}^{(t)}\right\|^2\right]$$

$$+ \frac{80(L_{\nabla a}^*)^2 (L_{\nabla a}')^2 \omega_B^2}{(1-\omega_B)^2(1-\omega_F)^2} \sum_{\iota=e}^{E-1} \sum_{t=T_0}^{T_1} \mathbb{E}\left[\left\|y_\iota^{(t+1)} - y_\iota^{(t)}\right\|^2\right] + \frac{80(L_{\nabla a}^*)^2 (L_{\nabla a}')^2 \omega_B^2}{(1-\omega_B)^2(1-\omega_F)} \sum_{\iota=e}^{E-1} \mathbb{E}\left[\left\|\tilde{y}_\iota^{(T_0)} - y_\iota^{(T_0)}\right\|^2\right]$$

$$+ \frac{5(L_{\nabla a}^*)^2 \omega_B^2}{(1-\omega_B)^2} \sum_{\iota=e}^{E-2} \mathbb{E}\left[\left\|\tilde{v}_{\iota+1}^{(T_0)} - \hat{v}_{\iota+1}^{(T_0)}\right\|^2\right] + 2(L_{\nabla a}^*)^2 \frac{\omega_B}{1-\omega_B} \sum_{\iota=e}^{E-1} \mathbb{E}\left[\left\|\tilde{v}_\iota^{(T_0)} - v_\iota^{(T_0)}\right\|^2\right]$$

$$\tag{69}$$

holds for $e = 1, 2, \cdots, E - 1$, where $(L_{\nabla a}^*)^2 := \max\{1, (2(L_{\nabla a}^\circ)^2)^E\} \geq 0$.

For $e = 1, 2, \cdots, E - 1$, define $\{C_{v,e}\}$ as:

$$C_{v,1} = 1, \quad C_{v,e} = 1 + \sum_{\iota=1}^{e-1} C_{v,\iota} \frac{5(L_{\nabla a}^*)^2 \omega_B^2}{(1-\omega_B)^2} \quad (e = 2, \cdots, E - 1). \tag{70}$$

Then taking summation on both sides of (69) over $e = 1, 2, \cdots, E - 1$, we have:

$$\sum_{e=1}^{E-1} C_{v,e} \sum_{t=T_0+1}^{T_1+1} \mathbb{E}\left[\left\|\tilde{v}_e^{(t)} - \hat{v}_e^{(t)}\right\|^2\right] \leq \sum_{e=1}^{E-1} \left(\sum_{\iota=1}^{e-1} C_{v,\iota} \frac{5(L_{\nabla a}^*)^2 \omega_B^2}{(1-\omega_B)^2}\right) \sum_{t=T_0+1}^{T_1+1} \mathbb{E}\left[\left\|\tilde{v}_e^{(t)} - \hat{v}_e^{(t)}\right\|^2\right]$$

$$+ \sum_{e=1}^{E-1} \left(\sum_{\iota=1}^{e-1} C_{v,\iota} \frac{5(L_{\nabla a}^*)^2 \omega_B^2}{(1-\omega_B)^2}\right) \sum_{t=T_0}^{T_1} \mathbb{E}\left[\left\|\hat{v}_e^{(t+1)} - \hat{v}_e^{(t)}\right\|^2\right]$$

$$+ \sum_{e=1}^{E-1} \left(\sum_{\iota=1}^{e} C_{v,\iota} (L_{\nabla a}^*)^2 (2(L_{\nabla a}')^2)\right) \sum_{t=T_0+1}^{T_1+1} \mathbb{E}\left[\left\|\tilde{y}_e^{(t)} - \hat{y}_e^{(t)}\right\|^2\right]$$

$$+ \sum_{e=1}^{E} \left(\sum_{\iota=1}^{e-1} C_{v,\iota} \frac{20(L_{\nabla a}^*)^2 (L_{\nabla a}')^2 \omega_B^2}{(1-\omega_B)^2}\right) \sum_{t=T_0}^{T_1} \mathbb{E}\left[\left\|w_e^{(t+1)} - w_e^{(t)}\right\|^2\right]$$

$$+ \sum_{e=1}^{E-1} \left(\sum_{\iota=1}^{e} C_{v,\iota} \frac{80(L_{\nabla a}^*)^2 (L_{\nabla a}')^2 \omega_B^2}{(1-\omega_B)^2(1-\omega_F)^2}\right) \sum_{t=T_0}^{T_1} \mathbb{E}\left[\left\|y_e^{(t+1)} - y_e^{(t)}\right\|^2\right]$$

$$+ \sum_{e=1}^{E-1} \left(\sum_{\iota=1}^{e-1} C_{v,\iota} \frac{5(L_{\nabla a}^*)^2 \omega_B^2}{(1-\omega_B)^2}\right) \mathbb{E}\left[\left\|\tilde{v}_e^{(T_0)} - \hat{v}_e^{(T_0)}\right\|^2\right] + \sum_{e=1}^{E-1} \left(\sum_{\iota=1}^{e} C_{v,\iota} 2(L_{\nabla a}^*)^2 \frac{\omega_B}{1-\omega_B}\right) \mathbb{E}\left[\left\|\tilde{v}_e^{(T_0)} - v_e^{(T_0)}\right\|^2\right]$$

$$+ \sum_{e=1}^{E-1} \left(\sum_{\iota=1}^{e} C_{v,\iota} \frac{80(L_{\nabla a}^*)^2 (L_{\nabla a}')^2 \omega_B^2}{(1-\omega_B)^2(1-\omega_F)}\right) \mathbb{E}\left[\left\|\tilde{y}_e^{(T_0)} - y_e^{(T_0)}\right\|^2\right]. \tag{71}$$

Then, with the definition of $C_{v,e}$ in Eq. (70), we have:

$$\sum_{e=1}^{E-1} \sum_{t=T_0+1}^{T_1+1} \mathbb{E}\left[\left\|\tilde{v}_e^{(t)} - \hat{v}_e^{(t)}\right\|^2\right]$$

$$\leq \sum_{e=1}^{E-1} C_{y,e}^\circ \sum_{t=T_0+1}^{T_1+1} \mathbb{E}\left[\left\|\tilde{y}_e^{(t)} - \hat{y}_e^{(t)}\right\|^2\right] + \sum_{e=1}^{E} C_{W,e}^\circ \sum_{t=T_0}^{T_1} \mathbb{E}\left[\left\|w_e^{(t+1)} - w_e^{(t)}\right\|^2\right]$$

$$+ \sum_{e=1}^{E-1} C_{v,e}^\circ \sum_{t=T_0}^{T_1} \mathbb{E}\left[\left\|\hat{v}_e^{(t+1)} - \hat{v}_e^{(t)}\right\|^2\right] + \sum_{e=1}^{E-1} C_{\theta,e}^\circ \sum_{t=T_0}^{T_1} \mathbb{E}\left[\left\|y_e^{(t+1)} - y_e^{(t)}\right\|^2\right] + \sum_{e=1}^{E-1} C_{\theta,e}^1 \mathbb{E}\left[\left\|\tilde{y}_e^{(T_0)} - y_e^{(T_0)}\right\|^2\right]$$

$$+ \sum_{e=1}^{E-1} C_{v,e}^1 \mathbb{E}\left[\left\|\tilde{v}_e^{(T_0)} - \hat{v}_e^{(T_0)}\right\|^2\right] + \sum_{e=1}^{E-1} C_{v,e}^2 \mathbb{E}\left[\left\|\tilde{v}_e^{(T_0)} - v_e^{(T_0)}\right\|^2\right],$$

where the coefficients are defined as follows:

$$C_{v,e}^\circ = \sum_{\iota=1}^{e-1} C_{v,\iota} \frac{5(L_{\nabla a}^*)^2 \omega_B^2}{(1-\omega_B)^2} \leq \mathcal{O}\left(\frac{\omega_B^2}{(1-\omega_B)^{2(e-1)}}\right), \tag{72a}$$

$$C_{v,e}^1 = \sum_{\iota=1}^{e-1} C_{v,\iota} \frac{5(L_{\nabla a}^*)^2 \omega_B^2}{(1-\omega_B)^2} \le \mathcal{O}\left(\frac{\omega_B^2}{(1-\omega_B)^{2(e-1)}}\right), \tag{72b}$$

$$C_{v,e}^2 = \sum_{\iota=1}^{e} C_{v,\iota} \frac{2(L_{\nabla a}^*)^2 \omega_B}{1-\omega_B} \le \mathcal{O}\left(\frac{\omega_B}{(1-\omega_B)^{2e-1}}\right), \tag{72c}$$

$$C_{W,e}^\circ = \sum_{\iota=1}^{e-1} C_{v,\iota} \frac{20(L_{\nabla a}^*)^2 (L_{\nabla a}')^2 \omega_B^2}{(1-\omega_B)^2} \le \mathcal{O}\left(\frac{\omega_B^2}{(1-\omega_B)^{2(e-1)}}\right), \tag{72d}$$

$$C_{y,e}^\circ = \sum_{\iota=1}^{e} C_{v,\iota} (L_{\nabla a}^*)^2 (2(L_{\nabla a}')^2) \le \mathcal{O}\left(\frac{1}{(1-\omega_B)^{2(e-1)}}\right), \tag{72e}$$

$$C_{\theta,e}^\circ = \sum_{\iota=1}^{e} C_{v,\iota} \frac{80(L_{\nabla a}^*)^2 (L_{\nabla a}')^2 \omega_B^2}{(1-\omega_B)^2(1-\omega_F)^2} \le \mathcal{O}\left(\frac{\omega_B^2}{(1-\omega_B)^{2e}(1-\omega_F)^2}\right), \tag{72f}$$

$$C_{\theta,e}^1 = \sum_{\iota=1}^{e} C_{v,\iota} \frac{80(L_{\nabla a}^*)^2 (L_{\nabla a}')^2 \omega_B^2}{(1-\omega_B)^2(1-\omega_F)} \le \mathcal{O}\left(\frac{\omega_B^2}{(1-\omega_B)^{2e}(1-\omega_F)}\right). \tag{72g}$$

$$\square$$

Now, we try to combine the result of Eq. (68) with Eq. (13). For `Clapping-`**FU** , it holds for any $r = 1, 2, \cdots, r_0$ that:

$$\sum_{e=1}^{E-1} \sum_{t=Q_r+1}^{Q_{r+1}} \mathbb{E}\left[\left\|\tilde{v}_e^{(t)} - \hat{v}_e^{(t)}\right\|^2\right] = \sum_{e=1}^{E-1} \sum_{t=Q_r+1}^{Q_{r+1}-1} \mathbb{E}\left[\left\|\tilde{v}_e^{(t)} - \hat{v}_e^{(t)}\right\|^2\right]$$

$$\le \sum_{e=1}^{E-1} C_{y,e}^\circ \sum_{t=Q_r+1}^{Q_{r+1}-1} \mathbb{E}\left[\left\|\tilde{y}_e^{(t)} - \hat{y}_e^{(t)}\right\|^2\right] + \sum_{e=1}^{E} C_{W,e}^\circ \sum_{t=Q_r}^{Q_{r+1}-2} \mathbb{E}\left[\left\|w_e^{(t+1)} - w_e^{(t)}\right\|^2\right] \tag{73}$$

$$+ \sum_{e=1}^{E-1} C_{v,e}^\circ \sum_{t=Q_r}^{Q_{r+1}-2} \mathbb{E}\left[\left\|\hat{v}_e^{(t+1)} - \hat{v}_e^{(t)}\right\|^2\right] + \sum_{e=1}^{E-1} C_{\theta,e}^\circ \sum_{t=Q_r}^{Q_{r+1}-2} \mathbb{E}\left[\left\|y_e^{(t+1)} - y_e^{(t)}\right\|^2\right],$$

where it holds due to the fact that in `Clapping-`**FU** at $Q_r$-iteration it satisfies that

$$\tilde{v}_e^{(Q_r)} = v_e^{(Q_r)} = \hat{v}_e^{(Q_r)}, \quad \tilde{y}_e^{(Q_r)} = y_e^{(Q_r)} = \hat{y}_e^{(Q_r)}.$$

Taking summation over $r$, it holds that:

$$\sum_{e=1}^{E-1} \sum_{t=2}^{T+1} \mathbb{E}\left[\left\|\tilde{v}_e^{(t)} - \hat{v}_e^{(t)}\right\|^2\right]$$

$$\le \sum_{e=1}^{E-1} C_{y,e}^\circ \sum_{t=2}^{T+1} \mathbb{E}\left[\left\|\tilde{y}_e^{(t)} - \hat{y}_e^{(t)}\right\|^2\right] + \sum_{e=1}^{E} C_{W,e}^\circ \sum_{t=1}^{T} \mathbb{E}\left[\left\|w_e^{(t+1)} - w_e^{(t)}\right\|^2\right] \tag{74}$$

$$+ \sum_{e=1}^{E-1} C_{v,e}^\circ \sum_{r=1}^{r_0} \sum_{t=Q_r}^{Q_{r+1}-2} \mathbb{E}\left[\left\|\hat{v}_e^{(t+1)} - \hat{v}_e^{(t)}\right\|^2\right] + \sum_{e=1}^{E-1} C_{\theta,e}^\circ \sum_{r=1}^{r_0} \sum_{t=Q_r}^{Q_{r+1}-2} \mathbb{E}\left[\left\|y_e^{(t+1)} - y_e^{(t)}\right\|^2\right].$$

Plugging Eq. (74) into (13), we can find that:

$$\sum_{e=1}^{E} \sum_{t=1}^{T+1} \mathbb{E}\left[\left\|\tilde{u}_e^{(t)} - \nabla_e \ell(\boldsymbol{w}^{(t)})\right\|^2\right]$$

$$\le \sum_{e=1}^{E} \left(32 L_{\nabla\ell}^2 \left(\frac{p+m}{m^2(1-(1-p)(1-\frac{m}{2}))} + \frac{1}{m^2}\right) + 8(L_{\nabla a}^\circ)^2 C_{W,e}^\circ\right) \sum_{t=1}^{T} \mathbb{E}\left[\left\|w_e^{(t+1)} - w_e^{(t)}\right\|^2\right]$$

$$+ \sum_{e=1}^{E-1} \left(8(L_{\nabla a}')^2 + 8(L_{\nabla a}^\circ)^2 C_{y,e}^\circ\right) \sum_{t=2}^{T+1} \mathbb{E}\left[\left\|\tilde{y}_e^{(t)} - \hat{y}_e^{(t)}\right\|^2\right]$$

$$+ \sum_{e=1}^{E-1} 8(L_{\nabla a}^\circ)^2 C_{v,e}^\circ \sum_{r=1}^{r_0} \sum_{t=Q_r}^{Q_{r+1}-2} \mathbb{E}\left[\left\|\hat{v}_e^{(t+1)} - \hat{v}_e^{(t)}\right\|^2\right] + \sum_{e=1}^{E-1} 8(L_{\nabla a}^\circ)^2 C_{\theta,e}^\circ \sum_{r=1}^{r_0} \sum_{t=Q_r}^{Q_{r+1}-2} \mathbb{E}\left[\left\|y_e^{(t+1)} - y_e^{(t)}\right\|^2\right]$$

$$+ 4T\sigma^2 \frac{(2-p)m - (1-p)m^2}{1-(1-p)(1-m)^2} + \frac{3}{m} \sum_{e=1}^{E} \mathbb{E}\left[\left\|\tilde{u}_e^{(1)} - \nabla_e \ell(\boldsymbol{w}^{(1)})\right\|^2\right]. \tag{75}$$

Moreover for `Clapping`-**FC** , it holds from Eq. (68) that:

$$
\sum_{e=1}^{E-1}\sum_{t=2}^{T+1}\mathbb{E}\left[\left\|\tilde{v}_e^{(t)}-\hat{v}_e^{(t)}\right\|^2\right]
$$

$$
\leq \sum_{e=1}^{E-1} C_{y,e}^{\circ}\sum_{t=2}^{T+1}\mathbb{E}\left[\left\|\tilde{y}_e^{(t)}-\hat{y}_e^{(t)}\right\|^2\right] + \sum_{e=1}^{E} C_{W,e}^{\circ}\sum_{t=1}^{T}\mathbb{E}\left[\left\|w_e^{(t+1)}-w_e^{(t)}\right\|^2\right]
$$

$$
+ \sum_{e=1}^{E-1} C_{v,e}^{\circ}\sum_{t=1}^{T}\mathbb{E}\left[\left\|\hat{v}_e^{(t+1)}-\hat{v}_e^{(t)}\right\|^2\right] + \sum_{e=1}^{E-1} C_{\theta,e}^{\circ}\sum_{t=1}^{T}\mathbb{E}\left[\left\|y_e^{(t+1)}-y_e^{(t)}\right\|^2\right] + \sum_{e=1}^{E-1} C_{\theta,e}^{1}\mathbb{E}\left[\left\|\tilde{y}_e^{(1)}-y_e^{(1)}\right\|^2\right]
$$

$$
+ \sum_{e=1}^{E-1} C_{v,e}^{1}\mathbb{E}\left[\left\|\tilde{v}_e^{(1)}-\hat{v}_e^{(1)}\right\|^2\right] + \sum_{e=1}^{E-1} C_{v,e}^{2}\mathbb{E}\left[\left\|\tilde{v}_e^{(1)}-v_e^{(1)}\right\|^2\right]. \tag{76}
$$

Plugging Eq. (76) into (13), we can find that:

$$
\sum_{e=1}^{E}\sum_{t=2}^{T+1}\mathbb{E}\left[\left\|\tilde{u}_e^{(t)}-\nabla_e\ell(\boldsymbol{w}^{(t)})\right\|^2\right]
$$

$$
\leq \sum_{e=1}^{E}\left(32L_{\nabla\ell}^2\left(\frac{p+m}{m^2(1-(1-p)(1-\frac{m}{2}))}+\frac{1}{m^2}\right)+8(L_{\nabla a}^{\circ})^2 C_{W,e}^{\circ}\right)\sum_{t=1}^{T}\mathbb{E}\left[\left\|w_e^{(t+1)}-w_e^{(t)}\right\|^2\right]
$$

$$
+ \sum_{e=1}^{E-1}\left(8(L_{\nabla a}')^2+8(L_{\nabla a}^{\circ})^2 C_{y,e}^{\circ}\right)\sum_{t=2}^{T+1}\mathbb{E}\left[\left\|\tilde{y}_e^{(t)}-\hat{y}_e^{(t)}\right\|^2\right]
$$

$$
+ \sum_{e=1}^{E-1} 8(L_{\nabla a}^{\circ})^2 C_{v,e}^{\circ}\sum_{t=1}^{T}\mathbb{E}\left[\left\|\hat{v}_e^{(t+1)}-\hat{v}_e^{(t)}\right\|^2\right] + \sum_{e=1}^{E-1} 8(L_{\nabla a}^{\circ})^2 C_{\theta,e}^{\circ}\sum_{t=1}^{T}\mathbb{E}\left[\left\|y_e^{(t+1)}-y_e^{(t)}\right\|^2\right]
$$

$$
+ 4T\sigma^2\frac{(2-p)m-(1-p)m^2}{1-(1-p)(1-m)^2}+\frac{3}{m}\sum_{e=1}^{E}\mathbb{E}\left[\left\|\tilde{u}_e^{(1)}-\nabla_e\ell(\boldsymbol{w}^{(1)})\right\|^2\right] + \sum_{e=1}^{E-1} 8(L_{\nabla a}^{\circ})^2 C_{\theta,e}^{1}\mathbb{E}\left[\left\|\tilde{y}_e^{(1)}-y_e^{(1)}\right\|^2\right]
$$

$$
+ \sum_{e=1}^{E-1} 8(L_{\nabla a}^{\circ})^2 C_{v,e}^{1}\mathbb{E}\left[\left\|\tilde{v}_e^{(1)}-\hat{v}_e^{(1)}\right\|^2\right] + \sum_{e=1}^{E-1} 8(L_{\nabla a}^{\circ})^2 C_{v,e}^{2}\mathbb{E}\left[\left\|\tilde{v}_e^{(1)}-v_e^{(1)}\right\|^2\right]. \tag{77}
$$

Both Eq. (75) and Eq. (77) calls for the further analysis of the term:

$$
\sum_{e=1}^{E-1}\left(8(L_{\nabla a}')^2+8(L_{\nabla a}^{\circ})^2 C_{y,e}^{\circ}\right)\sum_{t=T_0+1}^{T_1+1}\mathbb{E}\left[\left\|\tilde{y}_e^{(t)}-\hat{y}_e^{(t)}\right\|^2\right] + \sum_{e=1}^{E-1} 8(L_{\nabla a}^{\circ})^2 C_{\theta,e}^{\circ}\sum_{t=T_0}^{T_1}\mathbb{E}\left[\left\|y_e^{(t+1)}-y_e^{(t)}\right\|^2\right]
$$

$$
+ \sum_{e=1}^{E-1} 8(L_{\nabla a}^{\circ})^2 C_{v,e}^{\circ}\sum_{t=1}^{T}\mathbb{E}\left[\left\|\hat{v}_e^{(t+1)}-\hat{v}_e^{(t)}\right\|^2\right]
$$

for any $T_1 > T_0 \geq 1$. Thus, we present the analysis as the following lemma:

**Lemma 11.** *Suppose Assumption 1 and 3 holds, then there exist coefficients $C_{y,\theta,e}^{\circ}$ and $C_{y,\theta,e}^{1}$ defined by (85) for each $e = 1, 2, \cdots, E-1$ and $T_1 > T_0 \geq 1$ such that:*

$$
\sum_{e=1}^{E-1}\left(8(L_{\nabla a}')^2+8(L_{\nabla a}^{\circ})^2 C_{y,e}^{\circ}\right)\sum_{t=T_0+1}^{T_1+1}\mathbb{E}\left[\left\|\tilde{y}_e^{(t)}-\hat{y}_e^{(t)}\right\|^2\right] + \sum_{e=1}^{E-1} 8(L_{\nabla a}^{\circ})^2 C_{\theta,e}^{\circ}\sum_{t=T_0}^{T_1}\mathbb{E}\left[\left\|y_e^{(t+1)}-y_e^{(t)}\right\|^2\right]
$$

$$
\leq \sum_{e=1}^{E-1} L_a^2\left(\sum_{\iota=e}^{E-1} C_{y,\theta,\iota}^{\circ}\left(\frac{8L_a^2}{(1-\omega_F)^2}\right)^{\iota-e}\right)\sum_{t=T_0}^{T_1}\mathbb{E}\left[\left\|w_e^{(t+1)}-w_e^{(t)}\right\|^2\right]
$$

$$
+ \left(\sum_{\iota=1}^{E-1} C_{y,\theta,\iota}^{\circ} L_a^2\left(\frac{8L_a^2}{(1-\omega_F)^2}\right)^{\iota-1}\right)\sum_{t=T_0}^{T_1}\mathbb{E}\left[\left\|x^{(t+1)}-x^{(t)}\right\|^2\right]
$$

$$
+ \sum_{e=1}^{E-1}\left(C_{y,\theta,e}^{1}+\sum_{\iota=e+1}^{E-1}\frac{8L_a^2}{1-\omega_F}\left(\frac{8L_a^2}{(1-\omega_F)^2}\right)^{\iota-e-1}\right)\mathbb{E}\left[\left\|\tilde{y}_e^{(T_0)}-y_e^{(T_0)}\right\|^2\right]. \tag{78}
$$

*Proof.* For $e = 2, \cdots, E-1$ and $t = T_0, \cdots, T_1$, we have:

$$
\left\|y_e^{(t+1)}-y_e^{(t)}\right\|^2 = \left\|a_e(\tilde{y}_{e-1}^{(t+1)}, w_e^{(t+1)}) - a_e(\tilde{y}_{e-1}^{(t)}, w_e^{(t)})\right\|^2
$$

$$
\leq L_a^2\left\|\tilde{y}_{e-1}^{(t+1)}-\tilde{y}_{e-1}^{(t)}\right\|^2 + L_a^2\left\|w_e^{(t+1)}-w_e^{(t)}\right\|^2. \tag{79}
$$

Taking expectation and then taking summation on boths over $t$, we can get:

$$\sum_{t=T_0}^{T_1} \mathbb{E}\left[\left\|y_e^{(t+1)} - y_e^{(t)}\right\|^2\right] \le L_a^2 \sum_{t=T_0}^{T_1} \mathbb{E}\left[\left\|\tilde{y}_{e-1}^{(t+1)} - \tilde{y}_{e-1}^{(t)}\right\|^2\right] + L_a^2 \sum_{t=T_0}^{T_1} \mathbb{E}\left[\left\|w_e^{(t+1)} - w_e^{(t)}\right\|^2\right]$$

$$\le \frac{8L_a^2}{(1-\omega_F)^2} \sum_{t=T_0}^{T_1} \mathbb{E}\left[\left\|y_{e-1}^{(t+1)} - y_{e-1}^{(t)}\right\|^2\right] + \frac{8L_a^2}{1-\omega_F}\mathbb{E}\left[\left\|\tilde{y}_{e-1}^{(T_0)} - y_{e-1}^{(T_0)}\right\|^2\right] \tag{80}$$

$$+ L_a^2 \sum_{t=T_0}^{T_1} \mathbb{E}\left[\left\|w_e^{(t+1)} - w_e^{(t)}\right\|^2\right],$$

where the first inequality is due to Assumption 1 and the second inequality is due to (45).

Then, we have:

$$\sum_{t=T_0}^{T_1} \mathbb{E}\left[\left\|y_e^{(t+1)} - y_e^{(t)}\right\|^2\right]$$

$$\le \left(\frac{8L_a^2}{(1-\omega_F)^2}\right)^{e-1} \sum_{t=T_0}^{T_1} \mathbb{E}\left[\left\|y_1^{(t+1)} - y_1^{(t)}\right\|^2\right] + \sum_{\iota=1}^{e-1} \frac{8L_a^2}{1-\omega_F}\left(\frac{8L_a^2}{(1-\omega_F)^2}\right)^{e-1-\iota} \mathbb{E}\left[\left\|\tilde{y}_\iota^{(T_0)} - y_\iota^{(T_0)}\right\|^2\right] \tag{81}$$

$$+ \sum_{\iota=2}^{e} L_a^2 \left(\frac{8L_a^2}{(1-\omega_F)^2}\right)^{e-\iota} \sum_{t=T_0}^{T_1} \mathbb{E}\left[\left\|w_\iota^{(t+1)} - w_\iota^{(t)}\right\|^2\right].$$

Moreover, from Assumption 1 we get:

$$\left\|y_1^{(t+1)} - y_1^{(t)}\right\|^2 = \left\|a_1(x^{(t+1)}, w_1^{(t+1)}) - a_1(x^{(t)}, w_1^{(t)})\right\|^2$$

$$\le L_a^2 \left\|x^{(t+1)} - x^{(t)}\right\|^2 + L_a^2 \left\|w_1^{(t+1)} - w_1^{(t)}\right\|^2. \tag{82}$$

Plugging (82) into (81), then we have:

$$\sum_{t=T_0}^{T_1} \mathbb{E}\left[\left\|y_e^{(t+1)} - y_e^{(t)}\right\|^2\right]$$

$$\le \sum_{\iota=1}^{e} L_a^2 \left(\frac{8L_a^2}{(1-\omega_F)^2}\right)^{e-\iota} \sum_{t=T_0}^{T_1} \mathbb{E}\left[\left\|w_\iota^{(t+1)} - w_\iota^{(t)}\right\|^2\right] + L_a^2 \left(\frac{8L_a^2}{(1-\omega_F)^2}\right)^{e-1} \sum_{t=T_0}^{T_1} \mathbb{E}\left[\left\|x^{(t+1)} - x^{(t)}\right\|^2\right] \tag{83}$$

$$+ \sum_{\iota=1}^{e-1} \frac{8L_a^2}{1-\omega_F} \left(\frac{8L_a^2}{(1-\omega_F)^2}\right)^{e-1-\iota} \mathbb{E}\left[\left\|\tilde{y}_\iota^{(T_0)} - y_\iota^{(T_0)}\right\|^2\right].$$

Then, consider Eq. (44), we have:

$$\sum_{e=1}^{E-1} \left(8(L'_{\nabla a})^2 + 8(L^\circ_{\nabla a})^2 C^\circ_{y,e}\right) \sum_{t=T_0+1}^{T_1+1} \mathbb{E}\left[\left\|\tilde{y}_e^{(t)} - \hat{y}_e^{(t)}\right\|^2\right] + \sum_{e=1}^{E-1} 8(L^\circ_{\nabla a})^2 C^\circ_{\theta,e} \sum_{t=T_0}^{T_1} \mathbb{E}\left[\left\|y_e^{(t+1)} - y_e^{(t)}\right\|^2\right]$$

$$\le \sum_{e=1}^{E-1} \left(8(L'_{\nabla a})^2 + 8(L^\circ_{\nabla a})^2 C^\circ_{y,e}\right) \sum_{\iota=1}^{e} 2(2L_a^2)^{e-\iota} \frac{\omega_F^2}{(1-\omega_F)^2} \sum_{t=T_0}^{T_1} \mathbb{E}\left[\left\|y_\iota^{(t+1)} - y_\iota^{(t)}\right\|^2\right]$$

$$+ \sum_{e=1}^{E-1} \left(8(L'_{\nabla a})^2 + 8(L^\circ_{\nabla a})^2 C^\circ_{y,e}\right) \sum_{\iota=1}^{e} 2(2L_a^2)^{e-\iota} \frac{\omega_F}{1-\omega_F} \mathbb{E}\left[\left\|\tilde{y}_\iota^{(T_0)} - y_\iota^{(T_0)}\right\|^2\right]$$

$$+ \sum_{e=1}^{E-1} 8(L^\circ_{\nabla a})^2 C^\circ_{\theta,e} \sum_{t=T_0}^{T_1} \mathbb{E}\left[\left\|y_e^{(t+1)} - y_e^{(t)}\right\|^2\right]$$

$$\le \sum_{e=1}^{E-1} C^\circ_{y,\theta,e} \sum_{t=T_0}^{T_1} \mathbb{E}\left[\left\|y_e^{(t+1)} - y_e^{(t)}\right\|^2\right] + \sum_{e=1}^{E-1} C^1_{y,\theta,e} \mathbb{E}\left[\left\|\tilde{y}_e^{(T_0)} - y_e^{(T_0)}\right\|^2\right], \tag{84}$$

where

$$C^\circ_{y,\theta,e} = 8(L^\circ_{\nabla a})^2 C^\circ_{\theta,e} + \frac{2\omega_F^2 \sum_{\iota=e}^{E-1} \left(8(L'_{\nabla a})^2 + 8(L^\circ_{\nabla a})^2 C^\circ_{y,\iota}\right)(2L_a^2)^{\iota-e}}{(1-\omega_F)^2}, \tag{85a}$$

$$C^1_{y,\theta,e} = \frac{2\omega_F \sum_{\iota=e}^{E-1} \left(8(L'_{\nabla a})^2 + 8(L^\circ_{\nabla a})^2 C^\circ_{y,\iota}\right)(2L_a^2)^{\iota-e}}{1-\omega_F}. \tag{85b}$$

From (72) we know that:

$$C_{y,e}^{\circ} \le \mathcal{O}\left(\frac{1}{(1-\omega_B)^{2(e-1)}}\right), \quad C_{\theta,e}^{\circ} \le \mathcal{O}\left(\frac{\omega_B^2}{(1-\omega_B)^{2e}(1-\omega_F)^2}\right).$$

$$C_{y,\theta,e}^{\circ} \le \mathcal{O}\left(\frac{\omega_B^2}{(1-\omega_B)^2(1-\omega_F)^2} + \frac{\omega_F^2}{(1-\omega_B)^{2(E-2)}(1-\omega_F)^2}\right),$$

$$C_{y,\theta,e}^{1} \le \mathcal{O}\left(\frac{\omega_F^2}{(1-\omega_B)^{2(E-2)}(1-\omega_F)^2}\right).$$

Then, plugging (83) into (84), we can get:

$$\sum_{e=1}^{E-1}\left(8(L_{\nabla a}')^2 + 8(L_{\nabla a}^{\circ})^2 C_{y,e}^{\circ}\right)\sum_{t=T_0+1}^{T_1+1}\mathbb{E}\left[\left\|\tilde{y}_e^{(t)} - \hat{y}_e^{(t)}\right\|^2\right] + \sum_{e=1}^{E-1}8(L_{\nabla a}^{\circ})^2 C_{\theta,e}^{\circ}\sum_{t=T_0}^{T_1}\mathbb{E}\left[\left\|y_e^{(t+1)} - y_e^{(t)}\right\|^2\right]$$

$$\le \sum_{e=1}^{E-1} L_a^2\left(\sum_{\iota=e}^{E-1} C_{y,\theta,\iota}^{\circ}\left(\frac{8L_a^2}{(1-\omega_F)^2}\right)^{\iota-e}\right)\sum_{t=T_0}^{T_1}\mathbb{E}\left[\left\|w_e^{(t+1)} - w_e^{(t)}\right\|^2\right]$$

$$+ \left(\sum_{\iota=1}^{E-1} C_{y,\theta,\iota}^{\circ} L_a^2\left(\frac{8L_a^2}{(1-\omega_F)^2}\right)^{\iota-1}\right)\sum_{t=T_0}^{T_1}\mathbb{E}\left[\left\|x^{(t+1)} - x^{(t)}\right\|^2\right]$$

$$+ \sum_{e=1}^{E-1}\left(C_{y,\theta,e}^{1} + \sum_{\iota=e+1}^{E-1}\frac{8L_a^2}{1-\omega_F}\left(\frac{8L_a^2}{(1-\omega_F)^2}\right)^{\iota-e-1}\right)\mathbb{E}\left[\left\|\tilde{y}_e^{(T_0)} - y_e^{(T_0)}\right\|^2\right].$$

$\square$

Finally, we consider the term $\left\|\hat{v}_e^{(t+1)} - \hat{v}_e^{(t)}\right\|^2$:

**Lemma 12.** *Suppose Assumption 1 and 3 holds, then there exist coefficients $C_{v,x}^{\circ}$ and $C_{v,w,e}^{\circ}$ defined by (92) for each $e = 1, 2, \cdots, E$ and $T_1 > T_0 \ge 1$ such that:*

$$\sum_{e=1}^{E-1}8(L_{\nabla a}^{\circ})^2 C_{v,e}^{\circ}\sum_{t=T_0}^{T_1}\mathbb{E}\left[\left\|\hat{v}_e^{(t+1)} - \hat{v}_e^{(t)}\right\|^2\right]$$

$$\le C_{v,x}^{\circ}\sum_{t=T_0}^{T_1}\mathbb{E}\left[\left\|x^{(t+1)} - x^{(t)}\right\|^2\right] + \sum_{e=1}^{E} C_{v,w,e}^{\circ}\sum_{t=T_0}^{T_1}\mathbb{E}\left[\left\|w_e^{(t+1)} - w_e^{(t)}\right\|^2\right]. \tag{86}$$

*Proof.* For every $e = 1, 2, \cdots, E-2$ and $t = 1, 2, \cdots, T$ and $T = T_0, \cdots, T_1$, we have:

$$\left\|\hat{v}_e^{(t+1)} - \hat{v}_e^{(t)}\right\|^2 = \left\|\nabla_1 a_{e+1}(\hat{y}_e^{(t+1)}, W_{e+1}^{(t+1)})^{\top}\hat{v}_{e+1}^{(t+1)} - \nabla_1 a_{e+1}(\hat{y}_e^{(t)}, W_{e+1}^{(t)})^{\top}\hat{v}_{e+1}^{(t)}\right\|^2$$

$$\le (L_{\nabla a}')^2\left(\left\|\hat{y}_e^{(t+1)} - \hat{y}_e^{(t)}\right\|^2 + \left\|w_{e+1}^{(t+1)} - w_{e+1}^{(t)}\right\|^2\right) + (L_{\nabla a}^{\circ})^2\left\|\hat{v}_{e+1}^{(t+1)} - \hat{v}_{e+1}^{(t)}\right\|^2, \tag{87}$$

where the first inequality is due to (12).

Then we can get:

$$\left\|\hat{v}_e^{(t+1)} - \hat{v}_e^{(t)}\right\|^2 \le \sum_{\iota=e}^{E-1}(L_{\nabla a}')^2((L_{\nabla a}^{\circ})^2)^{\iota-e}\left\|\hat{y}_\iota^{(t+1)} - \hat{y}_\iota^{(t)}\right\|^2$$

$$+ \sum_{\iota=e+1}^{E}(L_{\nabla a}')^2((L_{\nabla a}^{\circ})^2)^{\iota-e-1}\left\|w_\iota^{(t+1)} - w_\iota^{(t)}\right\|^2, \tag{88}$$

where the second inequality is due to the definition of $\hat{v}_{E-1}^{(t)}$ and Assumption 1. And we can know that (88) also holds in the case of $e = E - 1$.

Then we consider the term $\left\|\hat{y}_e^{(t+1)} - \hat{y}_e^{(t)}\right\|^2$. For $e = 1, 2, \cdots, E-1$, we obtain from Assumption 1 that:

$$\left\|\hat{y}_e^{(t+1)} - \hat{y}_e^{(t)}\right\|^2 = \left\|a_e(\hat{y}_{e-1}^{(t+1)}, w_e^{(t+1)}) - a_e(\hat{y}_{e-1}^{(t)}, w_e^{(t)})\right\|^2$$

$$\le L_a^2\left\|\hat{y}_{e-1}^{(t+1)} - \hat{y}_{e-1}^{(t)}\right\|^2 + L_a^2\left\|w_e^{(t+1)} - w_e^{(t)}\right\|^2. \tag{89}$$

Then we have:

$$\left\| \hat{y}_e^{(t+1)} - \hat{y}_e^{(t)} \right\|^2 \leq \sum_{\iota=1}^{e} (L_a^2)^{e-\iota+1} \left\| w_\iota^{(t+1)} - w_\iota^{(t)} \right\|^2 + (L_a^2)^e \left\| x^{(t+1)} - x^{(t)} \right\|^2. \tag{90}$$

Combining (88) and (90) together, then we can get:

$$\sum_{e=1}^{E-1} 8(L_{\nabla a}^\circ)^2 C_{v,e}^\circ \sum_{t=1}^{T} \mathbb{E}\left[ \left\| \hat{v}_e^{(t+1)} - \hat{v}_e^{(t)} \right\|^2 \right]$$

$$\leq \sum_{e=1}^{E-1} \left( \sum_{\iota=1}^{e} 8C_{v,\iota}^\circ (L_{\nabla a}')^2 ((L_{\nabla a}^\circ)^2)^{e-\iota+1} \right) \sum_{t=1}^{T} \mathbb{E}\left[ \left\| \hat{y}_e^{(t+1)} - \hat{y}_e^{(t)} \right\|^2 \right]$$

$$+ \sum_{e=2}^{E} \left( \sum_{\iota=1}^{e-1} 8C_{v,\iota}^\circ (L_{\nabla a}')^2 ((L_{\nabla a}^\circ)^2)^{e-\iota} \right) \sum_{t=1}^{T} \mathbb{E}\left[ \left\| w_e^{(t+1)} - w_e^{(t)} \right\|^2 \right]$$

$$\leq \sum_{e=1}^{E-1} \left( \sum_{\iota=1}^{e} 8C_{v,\iota}^\circ (L_{\nabla a}')^2 ((L_{\nabla a}^\circ)^2)^{e-\iota+1} \right) \tag{91}$$

$$\cdot \sum_{t=1}^{T} \mathbb{E}\left[ \sum_{\eta=1}^{e} (L_a^2)^{e-\eta+1} \left\| w_\eta^{(t+1)} - w_\eta^{(t)} \right\|^2 + (L_a^2)^e \left\| x^{(t+1)} - x^{(t)} \right\|^2 \right]$$

$$+ \sum_{e=2}^{E} \left( \sum_{\iota=1}^{e-1} 8C_{v,\iota}^\circ (L_{\nabla a}')^2 ((L_{\nabla a}^\circ)^2)^{e-\iota} \right) \sum_{t=1}^{T} \mathbb{E}\left[ \left\| w_e^{(t+1)} - w_e^{(t)} \right\|^2 \right]$$

$$\leq C_{v,x}^\circ \sum_{t=1}^{T} \mathbb{E}\left[ \left\| x^{(t+1)} - x^{(t)} \right\|^2 \right] + \sum_{e=1}^{E} C_{v,w,e}^\circ \sum_{t=1}^{T} \mathbb{E}\left[ \left\| w_e^{(t+1)} - w_e^{(t)} \right\|^2 \right],$$

where

$$C_{v,x}^\circ = \sum_{e=1}^{E-1} \left( \sum_{\iota=1}^{e} 8C_{v,\iota}^\circ (L_{\nabla a}')^2 ((L_{\nabla a}^\circ)^2)^{e-\iota+1} \right) (L_a^2)^e, \tag{92a}$$

$$C_{v,w,e}^\circ = \sum_{\iota=1}^{e-1} 8C_{v,\iota}^\circ (L_{\nabla a}')^2 ((L_{\nabla a}^\circ)^2)^{e-\iota} + \sum_{\eta=e}^{E-1} \left( \sum_{\iota=1}^{\eta} 8C_{v,\iota}^\circ (L_{\nabla a}')^2 ((L_{\nabla a}^\circ)^2)^{\eta-\iota+1} \right) (L_a^2)^{\eta-e+1}. \tag{92b}$$

And from (72) we know that:

$$C_{v,x}^\circ \leq \mathcal{O}\left( \frac{\omega_B^2}{(1-\omega_B)^{2(E-2)}} \right), \quad C_{v,w,e}^\circ \leq \mathcal{O}\left( \frac{\omega_B^2}{(1-\omega_B)^{2(E-2)}} \right).$$

$\square$

## C.6 CONVERGENCE RATE OF GENERAL CASES

### C.6.1 CONVERGENCE RATE OF Clapping-FU

Based on (8), (75), (78), and (86), we can present the final convergence rate of Clapping-FU , which will be shown as the following lemma.

**Lemma 13** (Convergence rate of Clapping-FU ). *Suppose Assumption 1-4 hold. Then for Algorithm 4 there exist $\gamma, m, p > 0$ such that:*

$$\frac{1}{T} \sum_{t=1}^{T} \mathbb{E}\left[ \left\| \nabla \ell(\boldsymbol{w}^{(t)}) \right\|^2 \right] \lesssim \frac{\sigma}{\sqrt{T}} + \frac{1}{T(1-\omega_B)^{E-1}(1-\omega_F)^{E-1}}. \tag{93}$$

*Proof.* Combining (78), and (86) together, it holds that:

$$\sum_{e=1}^{E-1} 8(L_{\nabla a}^\circ)^2 C_{v,e}^\circ \sum_{r=1}^{r_0} \sum_{t=Q_r}^{Q_{r+1}-2} \mathbb{E}\left[ \left\| \hat{v}_e^{(t+1)} - \hat{v}_e^{(t)} \right\|^2 \right] + \sum_{e=1}^{E-1} 8(L_{\nabla a}^\circ)^2 C_{\theta,e}^\circ \sum_{r=1}^{r_0} \sum_{t=Q_r}^{Q_{r+1}-2} \mathbb{E}\left[ \left\| y_e^{(t+1)} - y_e^{(t)} \right\|^2 \right]$$

$$+ \sum_{e=1}^{E-1} \left( 8(L_{\nabla a}')^2 + 8(L_{\nabla a}^\circ)^2 C_{y,e}^\circ \right) \sum_{t=2}^{T+1} \mathbb{E}\left[ \left\| \tilde{y}_e^{(t)} - \hat{y}_e^{(t)} \right\|^2 \right] \tag{94}$$

$$\leq \sum_{e=1}^{E} \left( L_a^2 \left( \sum_{\iota=e}^{E-1} C_{y,\theta,\iota}^\circ \left( \frac{8L_a^2}{(1-\omega_F)^2} \right)^{\iota-e} \right) + C_{v,w,e}^\circ \right) \sum_{t=1}^{T} \mathbb{E}\left[ \left\| w_e^{(t+1)} - w_e^{(t)} \right\|^2 \right].$$

Eq. (94) holds is also due to the fact that $x^{(t+1)} = x^{(t)}$ for any $t = Q_r, \cdots, Q_{r+1} - 2$, $y_e^{(Q_r)} = \hat{y}_e^{(Q_r)} = \tilde{y}_e^{(Q_r)}$, and $v_e^{(Q_r)} = \hat{v}_e^{(Q_r)} = \tilde{v}_e^{(Q_r)}$ for any $r = 1, 2, \cdots, r_0$.

Combining (75), (78), and (86) together, we can obtain:

$$
\begin{aligned}
&\sum_{e=1}^{E} \sum_{t=1}^{T+1} \mathbb{E}\left[\left\|\tilde{u}_e^{(t)} - \nabla_e \ell(\boldsymbol{w}^{(t)})\right\|^2\right] \\
&\leq \sum_{e=1}^{E}\left(32 L_{\nabla\ell}^2\left(\frac{p+m}{m^2(1-(1-p)(1-\frac{m}{2}))} + \frac{1}{m^2}\right) + C_{w,e}\right) \sum_{t=1}^{T} \mathbb{E}\left[\left\|w_e^{(t+1)} - w_e^{(t)}\right\|^2\right] \\
&\quad + 4T\sigma^2 \frac{(2-p)m - (1-p)m^2}{1-(1-p)(1-m)^2} + \frac{3}{m}\sum_{e=1}^{E} \mathbb{E}\left[\left\|\tilde{u}_e^{(1)} - \nabla_e \ell(\boldsymbol{w}^{(1)})\right\|^2\right],
\end{aligned}
\tag{95}
$$

where

$$
\begin{aligned}
C_{w,e} &= 8(L_{\nabla a}^\circ)^2 C_{W,e}^\circ + L_a^2\left(\sum_{\iota=e}^{E-1} C_{y,\theta,\iota}^\circ\left(\frac{8L_a^2}{(1-\omega_F)^2}\right)^{\iota-e}\right) + C_{v,w,e}^\circ \\
&\lesssim \frac{1}{(1-\omega_B)^{2(E-1)}(1-\omega_F)^{2(E-1)}}.
\end{aligned}
\tag{96}
$$

Let $p = p_0 = \mathcal{O}(1)$ as a constant with respect to $\sigma, T, \omega_F, \omega_B$. Then let

$$
m \sim \left(\frac{1}{(1-\omega_B)^{E-1}(1-\omega_F)^{E-1}} + \sigma\sqrt{T}\right)^{-1}, \; m \leq 1,
$$

$$
\gamma \sim \left(\frac{1}{(1-\omega_B)^{E-1}(1-\omega_F)^{E-1}} + \sigma\sqrt{T}\right)^{-1}, \; \gamma \leq 1.
$$

At this time, if $\gamma/m$ is sufficiently small, we can obtain that

$$
32 L_{\nabla\ell}^2\left(\frac{p+m}{m^2(1-(1-p)(1-\frac{m}{2}))} + \frac{1}{m^2}\right) + C_{w,e} - \frac{1}{2\gamma^2} \leq 0.
$$

Then we have from Eq. (8):

$$
\frac{1}{T}\sum_{t=1}^{T} \mathbb{E}\left[\left\|\nabla\ell(\boldsymbol{w}^{(t)})\right\|^2\right] \lesssim \frac{\sigma}{\sqrt{T}} + \frac{1}{T(1-\omega_B)^{E-1}(1-\omega_F)^{E-1}}.
$$

$\square$

### C.6.2 CONVERGENCE RATE OF Clapping-**FC**

Based on (8), (77), (78), and (86), we can present the final convergence lemma of the general cases, which will be shown as the following lemma.

**Lemma 14** (Convergence rate of Clapping-**FC** ). *Suppose Assumption 1-4 hold. Then for Algorithm 4 there exist $\gamma, m > 0$ such that:*

$$
\begin{aligned}
\frac{1}{T}\sum_{t=1}^{T} \mathbb{E}\left[\left\|\nabla\ell(\boldsymbol{w}^{(t)})\right\|^2\right] &\lesssim \frac{\sigma^{\frac{4}{3}}}{T^{\frac{1}{3}}(1-\omega_B)^{\frac{4(E-1)}{3}}(1-\omega_F)^{\frac{4(E-1)}{3}}} \\
&\quad + \frac{1}{T}\left(\frac{1}{(1-\omega_B)^{E-1}(1-\omega_F)^{E-1}} + \frac{\omega_F^2 + \omega_B}{(1-\omega_B)^{2(E-1)}(1-\omega_F)^2} + \frac{1}{(1-\omega_F)^{2(E-2)-1}}\right).
\end{aligned}
\tag{97}
$$

*Proof.* Combining (77), (78), and (86) together, we can obtain:

$$\sum_{e=1}^{E}\sum_{t=2}^{T+1}\mathbb{E}\left[\left\|\tilde{u}_e^{(t)} - \nabla_e\ell(\boldsymbol{w}^{(t)})\right\|^2\right]$$

$$\leq \sum_{e=1}^{E}\left(32L_{\nabla\ell}^2\left(\frac{p+m}{m^2(1-(1-p)(1-\frac{m}{2}))} + \frac{1}{m^2}\right) + C_{w,e}\right)\sum_{t=1}^{T}\mathbb{E}\left[\left\|w_e^{(t+1)} - w_e^{(t)}\right\|^2\right]$$

$$+ C_x\sum_{t=1}^{T}\mathbb{E}\left[\left\|x^{(t+1)} - x^{(t)}\right\|^2\right] + \sum_{e=1}^{E-1}C_{y,\theta,e}\mathbb{E}\left[\left\|\tilde{y}_e^{(1)} - y_e^{(1)}\right\|^2\right] + \sum_{e=1}^{E-1}C_{v,e}\mathbb{E}\left[\left\|\tilde{v}_e^{(1)} - \hat{v}_e^{(1)}\right\|^2\right] \quad (98)$$

$$+ \sum_{e=1}^{E-1}C_{v,\chi,e}\mathbb{E}\left[\left\|\tilde{v}_e^{(1)} - v_e^{(1)}\right\|^2\right] + 4T\sigma^2\frac{(2-p)m - (1-p)m^2}{1-(1-p)(1-m)^2}$$

$$+ \frac{3}{m}\sum_{e=1}^{E}\mathbb{E}\left[\left\|\tilde{u}_e^{(1)} - \nabla_e\ell(\boldsymbol{w}^{(1)})\right\|^2\right],$$

where

$$C_{w,e} = 8(L_{\nabla a}^\circ)^2 C_{W,e}^\circ + L_a^2\left(\sum_{\iota=e}^{E-1}C_{y,\theta,\iota}^\circ\left(\frac{8L_a^2}{(1-\omega_F)^2}\right)^{\iota-e}\right) + C_{v,w,e}^\circ$$

$$\lesssim \frac{1}{(1-\omega_B)^{2(E-1)}(1-\omega_F)^{2(E-1)}}, \quad (99a)$$

$$C_x = C_{v,x}^\circ + \sum_{\iota=1}^{E-1}C_{y,\theta,\iota}^\circ L_a^2\left(\frac{8L_a^2}{(1-\omega_F)^2}\right)^{\iota-1}$$

$$\lesssim \frac{\omega_F^2 + \omega_B^2}{(1-\omega_B)^{2(E-1)}(1-\omega_F)^{2(E-1)}}, \quad (99b)$$

$$C_{y,\theta,e} = 8(L_{\nabla a}^\circ)^2 C_{\theta,e}^1 + C_{y,\theta,e}^1 + \sum_{\iota=e+1}^{E-1}\frac{8L_a^2}{1-\omega_F}\left(\frac{8L_a^2}{(1-\omega_F)^2}\right)^{\iota-e-1}$$

$$\lesssim \frac{\omega_B^2}{(1-\omega_B)^{2e}(1-\omega_F)^2} + \frac{\omega_F^2}{(1-\omega_B)^{2(E-2)}(1-\omega_F)^2} + \frac{1}{(1-\omega_F)^{2(E-e-2)+1}}, \quad (99c)$$

$$C_{v,e} = 8(L_{\nabla a}^\circ)^2 C_{v,e}^1, \quad (99d)$$

$$C_{v,\chi,e} = 8(L_{\nabla a}^\circ)^2 C_{v,e}^2. \quad (99e)$$

Thus, we know that:

$$\sum_{e=1}^{E-1}C_{y,\theta,e}\mathbb{E}\left[\left\|\tilde{y}_e^{(1)} - y_e^{(1)}\right\|^2\right] + \sum_{e=1}^{E-1}C_{v,e}\mathbb{E}\left[\left\|\tilde{v}_e^{(1)} - \hat{v}_e^{(1)}\right\|^2\right] + \sum_{e=1}^{E-1}C_{v,\chi,e}\mathbb{E}\left[\left\|\tilde{v}_e^{(1)} - v_e^{(1)}\right\|^2\right]$$

$$\lesssim \frac{\omega_B}{(1-\omega_B)^{2(E-1)}(1-\omega_F)^2} + \frac{\omega_F^2}{(1-\omega_B)^{2(E-2)}(1-\omega_F)^2} + \frac{1}{(1-\omega_F)^{2(E-2)-1}}. \quad (100)$$

Plugging (95) into (8), and use the fact that $\frac{p+m}{1-(1-p)(1-\frac{m}{2})} \leq 2$ when $p,m \leq 1$, we can get:

$$\frac{1}{T}\sum_{t=1}^{T}\mathbb{E}\left[\left\|\nabla\ell(\boldsymbol{w}^{(t)})\right\|^2\right]$$

$$\leq \frac{2}{\gamma T}\mathbb{E}\left[\ell(\boldsymbol{w}^{(1)}) - \inf_{\boldsymbol{w}}l(\boldsymbol{w})\right] + \frac{C_x}{T}\sum_{t=1}^{T}\mathbb{E}\left[\left\|x^{(t+1)} - x^{(t)}\right\|^2\right]$$

$$+ \frac{1}{T}\sum_{e=1}^{E}\left(96L_{\nabla\ell}^2\frac{1}{m^2} + C_{w,e} - \frac{1}{2\gamma^2}\right)\sum_{t=1}^{T}\mathbb{E}\left[\left\|w_e^{(t+1)} - w_e^{(t)}\right\|^2\right]$$

$$+ \sum_{e=1}^{E-1}\frac{C_{y,\theta,e}}{T}\mathbb{E}\left[\left\|\tilde{y}_e^{(1)} - y_e^{(1)}\right\|^2\right] + \sum_{e=1}^{E-1}\frac{C_{v,e}}{T}\mathbb{E}\left[\left\|\tilde{v}_e^{(1)} - \hat{v}_e^{(1)}\right\|^2\right] + \sum_{e=1}^{E-1}\frac{C_{v,\chi,e}}{T}\mathbb{E}\left[\left\|\tilde{v}_e^{(1)} - v_e^{(1)}\right\|^2\right]$$

$$+ 4\sigma^2\frac{(2-p)m - (1-p)m^2}{1-(1-p)(1-m)^2} + \frac{3}{mT}\sum_{e=1}^{E}\mathbb{E}\left[\left\|\tilde{u}_e^{(1)} - \nabla_e\ell(\boldsymbol{w}^{(1)})\right\|^2\right] + \frac{1}{T}\mathbb{E}\left[\left\|\nabla\ell(\boldsymbol{w}^{(1)})\right\|^2\right].$$

$$(101)$$

Taking $p = \sqrt{m}$ where $h$ is an undetermined function with respectively to $\omega_F, \omega_B$, then the term of stochastic noise satisfies:

$$4\sigma^2 \frac{(2-p)m - (1-p)m^2}{1 - (1-p)(1-m)^2} \le 4\sigma^2 \frac{(2-\sqrt{m})m - (1-\sqrt{m})m^2}{1 - (1-\sqrt{m})(1-m)^2} \le 4\sigma^2 \cdot 2\sqrt{m}. \tag{102}$$

Let

$$m \sim \left( \frac{1}{(1-\omega_B)^{E-1}(1-\omega_F)^{E-1}} + \frac{\sigma^{\frac{4}{3}} T^{\frac{2}{3}}}{(1-\omega_B)^{\frac{4(E-1)}{3}}(1-\omega_F)^{\frac{4(E-1)}{3}}} \right)^{-1}, \; m \le 1,$$

$$\gamma \sim \left( \frac{1}{(1-\omega_B)^{E-1}(1-\omega_F)^{E-1}} + \frac{\sigma^{\frac{4}{3}} T^{\frac{2}{3}}}{(1-\omega_B)^{\frac{4(E-1)}{3}}(1-\omega_F)^{\frac{4(E-1)}{3}}} \right)^{-1}, \; \gamma \le 1.$$

At this time, if $\gamma/m$ is sufficiently small, we can obtain that

$$96 L_{\nabla \ell}^2 \frac{1}{m^2} + C_{w,e} - \frac{1}{2\gamma^2} \le 0.$$

Then with Assumption 4, we have:

$$\frac{1}{T} \sum_{t=1}^{T} \mathbb{E}\left[ \left\| \nabla l(W^{(t)}) \right\|^2 \right]$$

$$\le \frac{2}{\gamma T} \mathbb{E}\left[ \ell(\boldsymbol{w}^{(1)}) - \inf_{\boldsymbol{w}} l(\boldsymbol{w}) \right] + C_x \sqrt{m} \varphi^2 + 8\sigma^2 \sqrt{m} + \frac{3}{mT} \sum_{e=1}^{E} \mathbb{E}\left[ \left\| \tilde{u}_e^{(1)} - \nabla_e \ell(\boldsymbol{w}^{(1)}) \right\|^2 \right]$$

$$+ \sum_{e=1}^{E-1} \frac{C_{y,\theta,e}}{T} \mathbb{E}\left[ \left\| \tilde{y}_e^{(1)} - y_e^{(1)} \right\|^2 \right] + \sum_{e=1}^{E-1} \frac{C_{v,e}}{T} \mathbb{E}\left[ \left\| \tilde{v}_e^{(1)} - \hat{v}_e^{(1)} \right\|^2 \right] + \sum_{e=1}^{E-1} \frac{C_{v,\chi,e}}{T} \mathbb{E}\left[ \left\| \tilde{v}_e^{(1)} - v_e^{(1)} \right\|^2 \right]$$

$$+ \frac{1}{T} \mathbb{E}\left[ \left\| \nabla \ell(\boldsymbol{w}^{(1)}) \right\|^2 \right]$$

$$\lesssim \frac{\sigma^{\frac{4}{3}}}{T^{\frac{1}{3}} (1-\omega_B)^{\frac{4(E-1)}{3}} (1-\omega_F)^{\frac{4(E-1)}{3}}}$$

$$+ \frac{1}{T} \left( \frac{1}{(1-\omega_B)^{E-1}(1-\omega_F)^{E-1}} + \frac{\omega_F^2 + \omega_B}{(1-\omega_B)^{2(E-1)}(1-\omega_F)^2} + \frac{1}{(1-\omega_F)^{2(E-2)-1}} \right).$$

$$\square$$

**Remark 2.** *It can be observed that the conclusion in Lemma 1 can be directly obtained from Eq. (101).*

## D   ERROR PROPAGATION ANALYSIS

The following lemma states how the compressed error propagates during the forward and backward processes.

**Lemma 15.** *Suppose Assumptions 1 and 3 hold. Then, for $e = 1, \ldots, E-1$, the error of the forward activation in Algorithm 4 can be bounded above as follows:*

$$\left\| \tilde{y}_e^{(t)} - \hat{y}_e^{(t)} \right\|^2 \le \sum_{\iota=1}^{e} 2(2L_a^2)^{e-\iota} \left\| \tilde{y}_\iota^{(t)} - y_\iota^{(t)} \right\|^2. \tag{103}$$

*For the error of backward gradients in Algorithm 4, there exist constants $L_{\nabla a}^\circ, L_{\nabla a}' > 0$ such that:*

$$\left\| \tilde{v}_e^{(t)} - \hat{v}_e^{(t)} \right\|^2 \le 2 \sum_{\iota=e}^{E-1} (2(L_{\nabla a}^\circ)^2)^{\iota-e} \left\| \tilde{v}_\iota^{(t)} - v_\iota^{(t)} \right\|^2$$

$$+ 4(L_{\nabla a}')^2 \sum_{\iota=1}^{E-1} \sum_{s=\max\{e,\iota\}}^{E-1} (2(L_{\nabla a}^\circ)^2)^{s-e} (2L_a^2)^{s-\iota} \left\| \tilde{y}_\iota^{(t)} - y_\iota^{(t)} \right\|^2. \tag{104}$$

*Proof.* Eq. (103) is actually the same as Eq. (43). Then from (63) we can get:

$$\left\| \tilde{v}_e^{(t)} - \hat{v}_e^{(t)} \right\|^2$$
$$\leq 2 \sum_{\iota=e}^{E-1} (2(L_{\nabla a}^\circ)^2)^{\iota-e} \left\| \tilde{v}_\iota^{(t)} - v_\iota^{(t)} \right\|^2 + (2(L_{\nabla a}')^2) \sum_{\iota=e}^{E-1} (2(L_{\nabla a}^\circ)^2)^{\iota-e} \left\| \tilde{y}_\iota^{(t)} - \hat{y}_\iota^{(t)} \right\|^2, \tag{105}$$

where the last inequality uses the fact that $L_{\nabla a}' \geq L_{\nabla a}$ in Remark 1.

Then, plugging (103) into (105), we can obtain:

$$\left\| \tilde{v}_e^{(t)} - \hat{v}_e^{(t)} \right\|^2$$
$$\leq 2 \sum_{\iota=e}^{E-1} (2(L_{\nabla a}^\circ)^2)^{\iota-e} \left\| \tilde{v}_\iota^{(t)} - v_\iota^{(t)} \right\|^2 + (2(L_{\nabla a}')^2) \sum_{\iota=e}^{E-1} (2(L_{\nabla a}^\circ)^2)^{\iota-e} \sum_{s=1}^{\iota} 2(2L_a^2)^{\iota-s} \left\| \tilde{y}_s^{(t)} - y_s^{(t)} \right\|^2$$
$$\leq 2 \sum_{\iota=e}^{E-1} (2(L_{\nabla a}^\circ)^2)^{\iota-e} \left\| \tilde{v}_\iota^{(t)} - v_\iota^{(t)} \right\|^2 + 4(L_{\nabla a}')^2 \sum_{\iota=1}^{E-1} \sum_{s=\max\{e,\iota\}}^{E-1} (2(L_{\nabla a}^\circ)^2)^{s-e} (2L_a^2)^{s-\iota} \left\| \tilde{y}_\iota^{(t)} - y_\iota^{(t)} \right\|^2. \tag{106}$$

Then we know that Eq. (104) holds for $e = 1, \cdots, E-1$. $\qquad\square$

As indicated by Eq. (103), the error in forward activation is directly accumulated due to the compression operations of the preceding machines. Meanwhile, according to Eq. (104), the error in backward gradient evaluation, denoted as $\tilde{v}_e^{(t)} = \tilde{v}_{e+1}^{(t)} \nabla_1 a_{e+1}(\tilde{y}_e^{(t)}, W_e^{(t)})$, arises from two aspects. One aspect pertains to the accumulated errors during the backward compression in preceding machines, while the other relates to the error of the forward activation $\tilde{y}_e^{(t)}$. Moreover, the mathematical formulations presented in Eq. (103) and Eq. (104) jointly demonstrate that compression errors exhibit exponential amplification across distributed computing nodes. Furthermore, this analysis reveals a positive correlation between error magnitude and system complexity: as model dimensionality and parallelism scale increase, communication-induced compression errors emerge as a critical bottleneck in distributed training architectures.

## E  Clapping WITH ADAM OPTIMIZER

In this section, we present `Clapping` equipped with Adam Kingma (2014) optimizer, which can especially fit the need of pre-training and fine-tuning tasks for LLMs.

### E.1  ALGORITHM DESIGN

Similar to the standard back-propagation algorithms with Adam optimizer, we introduce two coefficients $\beta_1, \beta_2 \in (0,1)$ and use $\nu_e^{(t)}$ and $\upsilon_e^{(t)}$ to record the first- and second-order optimizer states for the parameter $w_e^{(t)}$ for $e = 1, 2, \cdots, E$, respectively. Then we use the following update rules to optimize $w_e^{(t)}$:

$$\tilde{u}_e^{(t)} = (1 - \beta_1)\tilde{u}_e^{(t-1)} + \beta_1 \nabla_2 a_e(\tilde{y}_{e-1}^{(t)}, w_e^{(t)})^\mathsf{T} \tilde{v}_e^{(t)}, \tag{107a}$$

$$\upsilon_e^{(t)} = (1 - \beta_2)\upsilon_e^{(t-1)} + \beta_2 \left( \nabla_2 a_e(\tilde{y}_{e-1}^{(t)}, w_e^{(t)})^\mathsf{T} \tilde{v}_e^{(t)} \right)^{\odot 2}, \tag{107b}$$

$$\nu_e^{(t)} = \frac{\tilde{u}_e^{(t)}}{\sqrt{\upsilon_e^{(t)} + \varepsilon}}, \tag{107c}$$

$$w_e^{(t+1)} = w_e^{(t)} - \gamma \nu_e^{(t)}, \tag{107d}$$

where $\odot 2$ denotes the second moment and $\varepsilon > 0$ is a fixed constant. The division and addition operate in (107c) are all sample-wised. Then we can present `Clapping` with Adam optimizer as Algorithm 5.

### E.2  CONVERGENCE OF Clapping WITH ADAM OPTIMIZER

In this subsection we present the convergence analysis of `Clapping` with Adam optimizer. Firstly, we need to present an additional assumption for the bounded gradient estimation:

---

**Algorithm 5** Clapping with Adam optimizer

---

**Require:** Initialize $\tilde{y}_e^{(0)} = 0, \tilde{v}_e^{(0)} = 0, v_e^{(0)} = 0, \tilde{u}_e^{(0)} = 0$ for $e = 1, \cdots, E - 1$. Initialize dataset $\mathcal{D}$,
    learning rate $\gamma_t$, compressor $\mathcal{C}$, and lazy sampling rate $\{p_t\}_{t=1}^T$.
    **for** $t = 1, \cdots, T$ **do**
      $x^{(t)}, f_{\text{FU}}^{(t)} = \mathsf{LazySampling}(\mathcal{D}, t, p_t)$, initialize $\tilde{y}_0^{(t)} = x^{(t)}$, and let $\tilde{v}_E^{(t)} = 1$.
      **for** $e = 1, 2, \cdots, E - 1$ **do**
        $\mathsf{Forward}_e(\tilde{y}_e^{(t-1)}, \tilde{y}_{e-1}^{(t)}, w_e^{(t)}, f_{\text{FU}}^{(t)})$,
      **end for**
      **for** $e = E, E - 1, \cdots, 1$ **do**
        Update $v_e^{(0)}, \tilde{u}_e^{(0)}$ and $w_e^{(t+1)}$ by (107), and take $\mathsf{Backward}_e(\tilde{y}_e^{(t-1)}, \tilde{y}_{e-1}^{(t)}, w_e^{(t)}, f_{\text{FU}}^{(t)})$ **if** $e \neq 1$.
      **end for**
    **end for**

---

**Assumption 5.** *There exist $M_1 \geq 0$ such that for each $e = 1, 2, \cdots, E$, the gradient estimation can be bonuded as:*

$$\left\| \nabla_2 a_e(\tilde{y}_{e-1}^{(t)}, w_e^{(t)})^\mathsf{T} \tilde{v}_e^{(t)} \right\| \leq M_1.$$

**Remark 3.** *We assume the gradient of $a_e(y, \boldsymbol{w})$ is bounded according to Assumption 1. Thus, if there exists a constant $\omega_0$ such that the compressor $\mathcal{C}$ satisfies $\|x - \mathcal{C}(x)\|^2 \leq \omega_0 \|x\|^2$ for any input vector $x$, then Assumption 5 can be satisfied. Such a bounded property is common for traditional compressors like TopK and quantization.*

With Assumption 5, one can obtain that the second-order optimizer state $v_e^{(t)}$ satisfy that $\left\| v_e^{(t)} \right\| \leq M_1^2$. Then we can directly obtain the following lemma

**Lemma 16.** *There exists $M > 0$ such that for $t = 1, 2, \cdots, T$ and $e = 1, 2, \cdots, E$ it holds that:*

$$\varepsilon \leq \left\| \sqrt{v_e^{(t)} + \varepsilon} \right\|_1 \leq M,$$

*where $\| \cdot \|_1$ denote the $\ell_1$-norm.*

**Lemma 17.** *Suppose Assumption 2 and Assumption 5 hold. And the step-size $\gamma$ satisfies that $\gamma \leq \min\left\{ \frac{1}{4L_{\nabla \ell}}, \frac{1}{4C_\nu} \right\}$, where $C_\nu^2 \leq \max\left\{ (M-1)^2, (\varepsilon-1)^2 \right\}$ is a constant. Then in Algorithm 5 we have:*

$$\frac{1}{T} \sum_{t=1}^T \mathbb{E}\left[ \left\| \nabla \ell(\boldsymbol{w}^{(t)}) \right\|^2 \right]$$

$$\leq \frac{2}{\gamma T} \mathbb{E}\left[ \ell(\boldsymbol{w}^{(1)}) - \inf_{\boldsymbol{w}} \ell(W) \right] - \frac{1}{2\gamma^2 T} \sum_{t=1}^T \mathbb{E}\left[ \left\| \boldsymbol{w}^{(t+1)} - \boldsymbol{w}^{(t)} \right\|^2 \right] + \frac{2}{T} \sum_{t=1}^T \sum_{e=1}^E \mathbb{E}\left[ \left\| \tilde{u}_e^{(t)} - \nabla_e \ell(\boldsymbol{w}^{(t)}) \right\|^2 \right].$$

$$(108)$$

*Proof.* As $\ell$ is $L_{\nabla\ell}$-smooth, then it holds that:

$$\ell(\boldsymbol{w}^{(t+1)})$$

$$\leq \ell(\boldsymbol{w}^{(t)}) + \langle \nabla\ell(\boldsymbol{w}^{(t)}), \boldsymbol{w}^{(t+1)} - \boldsymbol{w}^{(t)} \rangle + \frac{L_{\nabla\ell}}{2} \left\| \boldsymbol{w}^{(t+1)} - \boldsymbol{w}^{(t)} \right\|^2$$

$$\leq \ell(\boldsymbol{w}^{(t)}) - \frac{\gamma}{2} \left\| \nabla\ell(\boldsymbol{w}^{(t)}) \right\|^2 - \left( \frac{\gamma}{2} - \frac{L_{\nabla\ell}}{2} \right) \left\| \boldsymbol{w}^{(t+1)} - \boldsymbol{w}^{(t)} \right\|^2 + \frac{\gamma}{2} \sum_{e=1}^{E} \left\| \nu_e^{(t)} - \nabla\ell(\boldsymbol{w}^{(t)}) \right\|^2$$

$$\leq \ell(\boldsymbol{w}^{(t)}) - \frac{\gamma}{2} \left\| \nabla\ell(\boldsymbol{w}^{(t)}) \right\|^2 - \left( \frac{\gamma}{2} - \frac{L_{\nabla\ell}}{2} \right) \left\| \boldsymbol{w}^{(t+1)} - \boldsymbol{w}^{(t)} \right\|^2 + \gamma \sum_{e=1}^{E} \left\| \tilde{u}_e^{(t)} - \nabla\ell(\boldsymbol{w}^{(t)}) \right\|^2$$

$$+ \gamma \sum_{e=1}^{E} \left\| \left( \sqrt{v_e^{(t)} + \varepsilon} - 1 \right) \odot \nu_e^{(t)} \right\|^2$$

$$\leq \ell(\boldsymbol{w}^{(t)}) - \frac{\gamma}{2} \left\| \nabla\ell(\boldsymbol{w}^{(t)}) \right\|^2 - \left( \frac{\gamma}{2} - \frac{L_{\nabla\ell}}{2} - \frac{C_\nu^2}{\gamma} \right) \left\| \boldsymbol{w}^{(t+1)} - \boldsymbol{w}^{(t)} \right\|^2 + \gamma \sum_{e=1}^{E} \left\| \tilde{u}_e^{(t)} - \nabla\ell(\boldsymbol{w}^{(t)}) \right\|^2,$$

$$\tag{109}$$

where $C_\nu^2 \leq \max \left\{ (M-1)^2, (\varepsilon - 1)^2 \right\}$.

Then as $\gamma \leq \min \left\{ \dfrac{1}{4L_{\nabla\ell}}, \dfrac{1}{4C_\nu} \right\}$, we obtain that:

$$\ell(\boldsymbol{w}^{(t+1)})$$

$$\leq \ell(\boldsymbol{w}^{(t)}) - \frac{\gamma}{2} \left\| \nabla\ell(\boldsymbol{w}^{(t)}) \right\|^2 - \frac{\gamma}{4} \left\| \boldsymbol{w}^{(t+1)} - \boldsymbol{w}^{(t)} \right\|^2 + \gamma \sum_{e=1}^{E} \left\| \tilde{u}_e^{(t)} - \nabla\ell(\boldsymbol{w}^{(t)}) \right\|^2. \tag{110}$$

Similar to the proof of Lemma 2, we can obtain from Eq. (110) that:

$$\frac{1}{T} \sum_{t=1}^{T} \mathbb{E} \left[ \left\| \nabla\ell(\boldsymbol{w}^{(t)}) \right\|^2 \right]$$

$$\leq \frac{2}{\gamma T} \mathbb{E} \left[ \ell(\boldsymbol{w}^{(1)}) - \inf_{\boldsymbol{w}} \ell(W) \right] - \frac{1}{2\gamma^2 T} \sum_{t=1}^{T} \mathbb{E} \left[ \left\| \boldsymbol{w}^{(t+1)} - \boldsymbol{w}^{(t)} \right\|^2 \right] + \frac{2}{T} \sum_{t=1}^{T} \sum_{e=1}^{E} \mathbb{E} \left[ \left\| \tilde{u}_e^{(t)} - \nabla_e \ell(\boldsymbol{w}^{(t)}) \right\|^2 \right].$$

Then we finish the proof of Eq. (108). $\qquad \square$

As the analysis of compress error and error accumulation process are independent with the optimizer, we can directly use the other lemmas in Appendix C. Thus, (95) and (98) also hold in `Clapping`-**FC** and `Clapping`-**FU**, respectively. Finally, we can present the convergence rate of `Clapping` with Adam optimizer as follows:

**Lemma 18** (Convergence rate of `Clapping`-**FU** with Adam optimizer)**.** *Suppose Assumption 1-5 hold. Then for Algorithm 5 there exist $\gamma, \beta_1, p > 0$ such that:*

$$\frac{1}{T} \sum_{t=1}^{T} \mathbb{E} \left[ \left\| \nabla\ell(\boldsymbol{w}^{(t)}) \right\|^2 \right] \lesssim \frac{\sigma}{\sqrt{T}} + \frac{1}{T(1-\omega_B)^{E-1}(1-\omega_F)^{E-1}}. \tag{111}$$

**Lemma 19** (Convergence rate of `Clapping`-**FC** with Adam optimizer)**.** *Suppose Assumption 1-5 hold. Then for Algorithm 5 there exist $\gamma, \beta_1, p > 0$ such that:*

$$\frac{1}{T} \sum_{t=1}^{T} \mathbb{E} \left[ \left\| \nabla\ell(\boldsymbol{w}^{(t)}) \right\|^2 \right] \lesssim \frac{\sigma^{\frac{4}{3}}}{T^{\frac{1}{3}} (1-\omega_B)^{\frac{4(E-1)}{3}} (1-\omega_F)^{\frac{4(E-1)}{3}}}$$

$$+ \frac{1}{T} \left( \frac{1}{(1-\omega_B)^{E-1}(1-\omega_F)^{E-1}} + \frac{\omega_F^2 + \omega_B}{(1-\omega_B)^{2(E-1)}(1-\omega_F)^2} + \frac{1}{(1-\omega_F)^{2(E-2)-1}} \right). \tag{112}$$

# F    `Clapping` WITH LARGE BATCH

## F.1    ALGORITHM DEVELOPMENT

Here we establish the convergence of `Clapping` with large-batch gradients. Before presenting the theoretical analysis, we firstly present the detailed algorithm of `Clapping` with large-batch gradients.

---

**Algorithm 6** LazySampling_LargeBatch($\mathcal{D}, t, p$) with Sample-wise rule / Batch-wise rule

---

**if** $t = 1$ **then**

    Get the stochastic samples $x_{11}, \cdots, x_{1B}$ / the first batch of samples randomly from $\mathcal{D}$.

**else**

    **for** $b = 1, \cdots, B$ independently **do**

        Get $x_b^{(t)} = x_b^{(t-1)}$ with probability $1 - p$, and let $f_{\text{FU}}^{(t)} = \text{False}$.

        Get $x_b^{(t)}$ randomly from $\mathcal{D}$ with probability $p$, and let $f_{\text{FU}}^{(t)} = \text{True}$.

    **end for**

    Keep the batch of last iteration with probability $1 - p$, and let $f_{\text{FU}}^{(t)} = \text{False}$.

    Use a new batch with probability $p$, and let $f_{\text{FU}}^{(t)} = \text{True}$.

**end if**

**Return:** $\boldsymbol{x}^{(t)} = (x_{t1}, \cdots, x_{tB})^\top, f_{\text{FU}}^{(t)} = \text{True}$.

---

---

**Algorithm 7** ForwardEF_LargeBatch$_e(\tilde{\boldsymbol{y}}_e^{(t-1)}, \tilde{\boldsymbol{y}}_{e-1}^{(t)}, w_e^{(t)}, f_{\text{FU}}^{(t)})$

---

In machine $e$: $\boldsymbol{\theta}_e^{(t)} = a_e(\tilde{\boldsymbol{y}}_{e-1}^{(t)}, w_e^{(t)})$,

**if** Clapping-**FU** and $f_{\text{FU}}^{(t)} = \text{True}$ **then**

    Send $\boldsymbol{y}_e^{(t)}$ from worker $e$ to $e + 1$,

    $\tilde{\boldsymbol{y}}_e^{(t)} = \boldsymbol{y}_e^{(t)}$.

**else**

    Send $\mathcal{C}(\boldsymbol{\theta}_e^{(t)} - \tilde{\boldsymbol{y}}_e^{(t-1)})$ from machine $e$ to $e + 1$,

    In machine $e, e + 1$: $\tilde{\boldsymbol{y}}_e^{(t)} = \tilde{\boldsymbol{y}}_e^{(t-1)} + \mathcal{C}(\boldsymbol{\theta}_e^{(t)} - \tilde{\boldsymbol{y}}_e^{(t-1)})$.

**end if**

---

**Notations.** Suppose the batch size is $B \geq 1$, and we use bold type like $\boldsymbol{y}, \boldsymbol{v}$ to represent a high-dimensional matrix formed by variables computed by the current batch. And we use standard type with subscribe $b = 1, 2, \cdots, B$ denote the activation and gradient of activation in the corresponding batch. For example, we can denote $\boldsymbol{y}_e^{(t)} := (y_{e,1}^{(t)}, y_{e,2}^{(t)}, \cdots, y_{e,B}^{(t)})^\top$ as the activation of the $e$-th layer. The notation $\odot$ here denotes the sample-wise multiplication for a large batch. Specifically, if we denote $\boldsymbol{\alpha} = (\alpha_1, \cdots, \alpha_B)^\top, \boldsymbol{\beta} = (\beta_1, \cdots, \beta_B)^\top$, then $\boldsymbol{\alpha} \odot \boldsymbol{\beta} = (\alpha_1^\top \beta_1, \cdots, \alpha_B^\top \beta_B)^\top$. Similarly, if we denote $\boldsymbol{\alpha} = (\alpha_1, \cdots, \alpha_B)^\top, \boldsymbol{\beta} = (\beta_1, \cdots, \beta_B)^\top$, then $\boldsymbol{\alpha} \otimes \boldsymbol{\beta} = \sum_{b=1}^B \alpha_b^\top \beta_b$.

**Lazy sampling strategies of Clapping with large batch.** The lazy sampling strategies of Clapping with large batch are shown as Algorithm 6. The strategies in the large-batch are including Sample-wise rule and Batch-wise rule. The sample-wise rule means taking lazy sampling process **sample by sample**. Meanwhile, the batch-wise rule means one can keep the whole batch with probability $1 - p$ and use a new batch with probability $p$. We will show that both strategies can achieve the convergence later in Appendix F.2.

**Algorithm formulation of Clapping with large batch.** In large batch scenario, the forward and backward function for large-bach scenario are easy to obtain as they are all sample-wise. The detail is summarized as Algorithm 7 and 8. With the lazy sampling strategy as well as the forward/backward process, the Clapping algorithm in with large batch can be summarized as Algorithm 9.

## F.2 CONVERGENCE OF Clapping WITH LARGE BATCH

In this subsection, we present the convergence analysis of Clapping with large batch. Firstly, it is worth noting that the Descent Lemma (Lemma 2) remains applicable. Moreover, since the forward and backward propagations are, in fact, sample-wise, the propagations of activations and the gradients of activations during the forward and backward passes of Algorithm 9 are identical to those of Algorithm 4. The sole distinction between Algorithm 9 and Algorithm 4 lies in the acquisition of $\tilde{u}$.

---

**Algorithm 8** BackwardEF_LargeBatch$_{e+1}(\tilde{\boldsymbol{v}}_e^{(t-1)}, \tilde{\boldsymbol{v}}_{e+1}^{(t)}, w_{e+1}^{(t)}, f_{\text{FU}}^{(t)})$

---

In machine $e+1$: $\boldsymbol{\chi}_e^{(t)} = \tilde{\boldsymbol{v}}_{e+1}^{(t)} \odot \nabla_1 a_{e+1}(\tilde{\boldsymbol{y}}_e^{(t)}, w_{e+1}^{(t)})$,

**if** Clapping-**FU** and $f_{\text{FU}}^{(t)}$ = True **then**

   Send $\boldsymbol{v}_{e-1}^{(t)}$ from worker $e$ to $e-1$,

   $\tilde{\boldsymbol{v}}_{e-1}^{(t)} = \boldsymbol{v}_{e-1}^{(t)}$.

**else**

   Send $\mathcal{C}(\boldsymbol{\chi}_e^{(t)} - \tilde{\boldsymbol{v}}_e^{(t-1)})$ from machine $e+1$ to $e$,

   $\tilde{\boldsymbol{v}}_{e-1}^{(t)} = \tilde{\boldsymbol{v}}_{e-1}^{(t-1)} + \mathcal{C}(\boldsymbol{v}_{e-1}^{(t)} - \tilde{\boldsymbol{v}}_{e-1}^{(t-1)})$.

**end if**

---

---

**Algorithm 9** Clapping with large-batch gradients.

---

**Require:** Initialize $\{\tilde{\boldsymbol{y}}_e^{(0)} = 0\}_{e=1}^{E-1}$, $\{\tilde{\boldsymbol{v}}_e^{(0)} = 0\}_{e=1}^{E-1}$, and $\{\tilde{\boldsymbol{u}}_e^{(0)} = 0\}_{e=1}^{E}$, dataset $\mathcal{D}$, learning rate $\gamma_t$, compressor $\mathcal{C}$, lazy sampling rate $\{p_t\}_{t=1}^T$.

**for** $t = 1, \cdots, T$ **do**

   $\boldsymbol{x}^{(t)}, f_{\text{FU}}^{(t)}$ = LazySampling_LargeBatch$(\mathcal{D}, t, p_t)$ and let $\tilde{\boldsymbol{y}}_0^{(t)} = \boldsymbol{x}^{(t)}$.

   **for** $e = 1, 2, \cdots, E-1$ **do**

      Forward_LargeBatch$_e(\tilde{\boldsymbol{y}}_e^{(t-1)}, \tilde{\boldsymbol{y}}_{e-1}^{(t)}, w_e^{(t)}, f_{\text{FU}}^{(t)})$,

   **end for**

   Let $\tilde{v}_E^{(t)} = \mathbf{1}_B$.

   **for** $e = E, E-1, \cdots, 1$ **do**

      **In machine** $e$:

      $\tilde{u}_e^{(t)} = \frac{1}{B} \cdot m_t[\tilde{\boldsymbol{v}}_e^{(t)} \otimes \nabla_2 a_e(\tilde{\boldsymbol{y}}_{e-1}^{(t)}, w_e^{(t)})] + (1 - m_t)\tilde{u}_e^{(t-1)}$,

      $w_e^{(t+1)} = w_e^{(t)} - \gamma\tilde{u}_e^{(t)}$.

      **if** $e \neq 1$ **then**

         Backward_LargeBatch$_e(\tilde{\boldsymbol{y}}_e^{(t-1)}, \tilde{\boldsymbol{y}}_{e-1}^{(t)}, w_e^{(t)}, f_{\text{FU}}^{(t)})$,

      **end if**

   **end for**

**end for**

---

Consequently, the lemmas regarding error accumulation, specifically Lemmas 4 - 6 and Lemmas 8 - 12, can be directly employed in the analysis of Algorithm 9.

The following lemma is the large-batch version of Lemma 3:

**Lemma 20** (Large-batch version of Lemma 3). *Suppose Assumption 1 and 2 holds, and let $m_1 = \cdots = m_T = m_{T+1} = m$ as well as $p_3 = \cdots = p_T = p_{T+1} = p$. Moreover, we set $p_2 = 1$. Then, for all $t = 2, \cdots, T+1$ we have:*

$$\sum_{e=1}^E \sum_{t=1}^{T+1} \mathbb{E}\left[\left\|\tilde{u}_e^{(t)} - \nabla_e\ell(\boldsymbol{w}^{(t)})\right\|^2\right]$$

$$\leq 32L_{\nabla\ell}^2 \left(\frac{p+m}{m^2(1-(1-p)(1-\frac{m}{2}))} + \frac{1}{m^2}\right) \sum_{e=1}^E \sum_{t=1}^T \mathbb{E}\left[\left\|w_e^{(t+1)} - w_e^{(t)}\right\|^2\right]$$

$$+ \frac{8(L_{\nabla a}^\circ)^2}{B} \sum_{e=1}^{E-1} \sum_{b=1}^B \sum_{t=2}^{T+1} \mathbb{E}\left[\left\|\tilde{v}_{e,b}^{(t)} - \hat{v}_{e,b}^{(t)}\right\|^2\right] + \frac{8(L_{\nabla a}')^2}{B} \sum_{e=1}^{E-1} \sum_{b=1}^B \sum_{t=2}^{T+1} \mathbb{E}\left[\left\|\tilde{y}_{e,b}^{(t)} - \hat{y}_{e,b}^{(t)}\right\|^2\right]$$

$$+ 4T\frac{\sigma^2}{B} \frac{(2-p)m - (1-p)m^2}{1-(1-p)(1-m)^2} + \frac{3}{m} \sum_{e=1}^E \mathbb{E}\left[\left\|\tilde{u}_e^{(1)} - \nabla_e\ell(\boldsymbol{w}^{(1)})\right\|^2\right]. \tag{113}$$

*Proof.* For $t = 2, 3, \cdots, T$, we denote $\psi(t)$ as the last moment in which the sample is randomly obtained with $\mathcal{D}$ as of the $t$-th iteration. Specially,

$$\psi(t) := \max_{\tau \in \mathbb{S}_t} \tau, \quad \text{where } \mathbb{S}_t := \{\tau = 2, 3, \cdots, t | \text{sampling randomly at iteration } \tau\}.$$

Then, with the fact that the $p_2 = 1$, it holds for $\tau = 2, \cdots, t$ that $\Pr(\psi(t) = \tau) = \begin{cases} (1-p)^{t-2}, & \text{if } \tau = 2 \\ p(1-p)^{t-\tau}, & \text{else.} \end{cases}$

For $e = 1, 2, \cdots, E - 1$ and $t = 2, 3, \cdots, T + 1$, the error between the evaluated gradient and the true gradient satisfies:

$$
\begin{aligned}
& \tilde{u}_e^{(t)} - \nabla_e \ell(\boldsymbol{w}^{(t)}) \\
&= \underbrace{\sum_{\tau=\psi(t)}^{t} \frac{1}{B} \sum_{b=1}^{B} m(1-m)^{t-\tau} \left( \nabla_2 a_e(\tilde{y}_{e-1,b}^{(\tau)}, w_e^{(\tau)})^\top \tilde{v}_{e,b}^{(\tau)} - \nabla_2 a_e(\hat{y}_{e-1,b}^{(\tau)}, w_e^{(\tau)})^\top \hat{v}_{e,b}^{(\tau)} \right)}_{:=\Xi_{e,1}} \\
& + \sum_{\tau=\psi(t)}^{t} \frac{1}{B} \sum_{b=1}^{B} m(1-m)^{t-\tau} \left( \nabla_2 a_e(\hat{y}_{e-1,b}^{(\tau)}, w_e^{(\tau)})^\top \hat{v}_{e,b}^{(\tau)} - \nabla_e \ell(\boldsymbol{w}^\tau) \right) \\
& + \underbrace{\sum_{\tau=\psi(t)}^{t} m(1-m)^{t-\tau} \left( \nabla_e \ell(\boldsymbol{w}^{(\tau)}) - \nabla_e \ell(\boldsymbol{w}^{(t)}) \right)}_{:=\Xi_{e,2}} + \underbrace{(1-m)^{t+1-\psi(t)} \left( \nabla_e \ell(\boldsymbol{w}^{(\psi(t)-1)}) - \nabla_e \ell(\boldsymbol{w}^{(t)}) \right)}_{:=\Xi_{e,3}} \\
& + (1-m)^{t+1-\psi(t)} \left( \tilde{u}_e^{(\psi(t)-1)} - \nabla_e \ell(\boldsymbol{w}^{(\psi(t)-1)}) \right),
\end{aligned}
\tag{114}
$$

where the first equation is from the momentum update rule. Moreover, we use $\Xi_{e,1}, \Xi_{e,2}, \Xi_{e,3}$ to denote some complex terms, which have been shown in Eq. (114).

Taking the $\ell_2$-norm and conditional expectation with respect to $\mathcal{F}^{(\psi(t))}$ on both sides of Eq. (114), we can obtain:

$$
\begin{aligned}
& \mathbb{E}\left[ \left\| \tilde{u}_e^{(t)} - \nabla_e \ell(\boldsymbol{w}^{(t)}) \right\|^2 \Big| \mathcal{F}^{(\psi(t))} \right] \\
&= \mathbb{E}\left[ \left\| \sum_{\tau=\psi(t)}^{t} \frac{1}{B} \sum_{b=1}^{B} m(1-m)^{t-\tau} \left( \nabla_2 a_e(\hat{y}_{e-1,b}^{(\tau)}, w_e^{(\tau)})^\top \hat{v}_{e,b}^{(\tau)} - \nabla_e \ell(\boldsymbol{w}^\tau) \right) \right\|^2 \Big| \mathcal{F}^{(\psi(t))} \right] \\
& + \mathbb{E}\left[ \left\| (1-m)^{t+1-\psi(t)} \left( \tilde{u}_e^{(\psi(t)-1)} - \nabla_e \ell(\boldsymbol{w}^{(\psi(t)-1)}) \right) + \Xi_{e,1} + \Xi_{e,2} + \Xi_{e,3} \right\|^2 \Big| \mathcal{F}^{(\psi(t))} \right] \\
& + 2\mathbb{E}\left[ \Big\langle \sum_{\tau=\psi(t)}^{t} \frac{1}{B} \sum_{b=1}^{B} m(1-m)^{t-\tau} \left( \nabla_2 a_e(\hat{y}_{e-1,b}^{(\tau)}, w_e^{(\tau)})^\top \hat{v}_{e,b}^{(\tau)} - \nabla_e \ell(\boldsymbol{w}^\tau) \right), \\
& \qquad\qquad (1-m)^{t+1-\psi(t)} \left( \tilde{u}_e^{(\psi(t)-1)} - \nabla_e \ell(\boldsymbol{w}^{(\psi(t)-1)}) \right) + \Xi_{e,1} + \Xi_{e,2} + \Xi_{e,3} \Big\rangle \Big| \mathcal{F}^{(\psi(t))} \right].
\end{aligned}
\tag{115}
$$

Thus, we can get:

$$
\begin{aligned}
& \mathbb{E}\left[ \left\| \tilde{u}_e^{(t)} - \nabla_e \ell(\boldsymbol{w}^{(t)}) \right\|^2 \Big| \mathcal{F}^{(\psi(t))} \right] \\
&\leq \frac{2}{B^2} \sum_{b=1}^{B} \mathbb{E}\left[ \left\| \sum_{\tau=\psi(t)}^{t} m(1-m)^{t-\tau} \left( \nabla_2 a_e(\hat{y}_{e-1,b}^{(\tau)}, w_e^{(\tau)})^\top \hat{v}_{e,b}^{(\tau)} - \nabla_e \ell(\boldsymbol{w}^\tau) \right) \right\|^2 \Big| \mathcal{F}^{(\psi(t))} \right] \\
& + \mathbb{E}\left[ \left\| \Xi_{e,1} + \Xi_{e,2} + \Xi_{e,3} \right\|^2 \Big| \mathcal{F}^{(\psi(t))} \right] \\
& + \mathbb{E}\left[ \left\| (1-m)^{t+1-\psi(t)} \left( \tilde{u}_e^{(\psi(t)-1)} - \nabla_e \ell(\boldsymbol{w}^{(\psi(t)-1)}) \right) + \Xi_{e,1} + \Xi_{e,2} + \Xi_{e,3} \right\|^2 \Big| \mathcal{F}^{(\psi(t))} \right],
\end{aligned}
\tag{116}
$$

where the inequality is due to Cauchy-Schwarz inequality, Assumption 2 and the fact that samples in the batch are obtained independently.

For the second term of the right-hand-side of Eq. (116), it holds that:

$$
\mathbb{E}\left[\left\|\Xi_{e,1} + \Xi_{e,2} + \Xi_{e,3}\right\|^2 \Big| \mathcal{F}^{(\psi(t))}\right]
$$

$$
\leq \frac{2}{B}\sum_{\tau=\psi(t)}^{t}\sum_{b=1}^{B} m(1-m)^{t-\tau}\mathbb{E}\left[\left\|\nabla_2 a_e(\tilde{y}_{e-1,b}^{(\tau)}, w_e^{(\tau)})^{\top}\tilde{v}_{e,b}^{(\tau)} - \nabla_2 a_e(\hat{y}_{e-1,b}^{(\tau)}, w_e^{(\tau)})^{\top}\hat{v}_{e,b}^{(\tau)}\right\|^2 \Big| \mathcal{F}^{(\psi(t))}\right]
$$

$$
+ 2\sum_{\tau=\psi(t)}^{t} m(1-m)^{t-\tau}\mathbb{E}\left[\left\|\nabla_e\ell(\boldsymbol{w}^{(\tau)}) - \nabla_e\ell(\boldsymbol{w}^{(t)})\right\|^2 \Big| \mathcal{F}^{(\psi(t))}\right]
$$

$$
+ (1-m)^{t+1-\psi(t)}\mathbb{E}\left[\left\|\nabla_e\ell(\boldsymbol{w}^{(\psi(t)-1)}) - \nabla_e\ell(\boldsymbol{w}^{(t)})\right\|^2 \Big| \mathcal{F}^{(\psi(t))}\right],
$$

(117)

where the inequality holds is due to the convexity of the $\ell_2$-norm.

Moreover, for the last term, it also holds that:

$$
\mathbb{E}\left[\left\|(1-m)^{t+1-\psi(t)}\left(\tilde{u}_e^{(\psi(t)-1)} - \nabla_e\ell(\boldsymbol{w}^{(\psi(t)-1)})\right) + \Xi_{e,1} + \Xi_{e,2} + \Xi_{e,3}\right\|^2 \Big| \mathcal{F}^{(\psi(t))}\right]
$$

$$
\leq \frac{2}{B}\sum_{\tau=\psi(t)}^{t}\sum_{b=1}^{B} m(1-m)^{t-\tau}\mathbb{E}\left[\left\|\nabla_2 a_e(\tilde{y}_{e-1,b}^{(\tau)}, w_e^{(\tau)})^{\top}\tilde{v}_{e,b}^{(\tau)} - \nabla_2 a_e(\hat{y}_{e-1,b}^{(\tau)}, w_e^{(\tau)})^{\top}\hat{v}_{e,b}^{(\tau)}\right\|^2 \Big| \mathcal{F}^{(\psi(t))}\right]
$$

$$
+ 2\sum_{\tau=\psi(t)}^{t} m(1-m)^{t-\tau}\mathbb{E}\left[\left\|\nabla_e\ell(\boldsymbol{w}^{(\tau)}) - \nabla_e\ell(\boldsymbol{w}^{(t)})\right\|^2 \Big| \mathcal{F}^{(\psi(t))}\right]
$$

$$
+ (1-m)^{t+1-\psi(t)}\mathbb{E}\left[\left\|\left(\tilde{u}_e^{(\psi(t)-1)} - \nabla_e\ell(\boldsymbol{w}^{(\psi(t)-1)})\right) + \left(\nabla_e\ell(\boldsymbol{w}^{(\psi(t)-1)}) - \nabla_e\ell(\boldsymbol{w}^{(t)})\right)\right\|^2 \Big| \mathcal{F}^{(\psi(t))}\right].
$$

(118)

With Assumption 2, we can get:

$$
\frac{1}{B^2}\mathbb{E}\left[\sum_{b=1}^{B}\sum_{e=1}^{E}\left\|\sum_{\tau=\psi(t)}^{t} m(1-m)^{t-\tau}\left(\nabla_2 a_e(\hat{y}_{e-1,b}^{(\tau)}, w_e^{(\tau)})^{\top}\hat{v}_{e,b}^{(\tau)} - \nabla_e\ell(\boldsymbol{w}^{\tau})\right)\right\|^2 \Big| \mathcal{F}^{(\psi(t))}\right]
$$

$$
\leq \left(\sum_{\tau=\psi(t)}^{t} m(1-m)^{t-\tau}\right)^2 \frac{\sigma^2}{B}.
$$

Finally, with the same process of Lemma 3, we can present the result that:

$$
\sum_{e=1}^{E}\sum_{t=1}^{T+1}\mathbb{E}\left[\left\|\tilde{u}_e^{(t)} - \nabla_e\ell(\boldsymbol{w}^{(t)})\right\|^2\right]
$$

$$
\leq 32 L_{\nabla\ell}^2\left(\frac{p+m}{m^2(1-(1-p)(1-\frac{m}{2}))} + \frac{1}{m^2}\right)\sum_{e=1}^{E}\sum_{t=1}^{T}\mathbb{E}\left[\left\|w_e^{(t+1)} - w_e^{(t)}\right\|^2\right]
$$

$$
+ \frac{8(L_{\nabla a}^{\circ})^2}{B}\sum_{e=1}^{E-1}\sum_{b=1}^{B}\sum_{t=2}^{T+1}\mathbb{E}\left[\left\|\tilde{v}_{e,b}^{(t)} - \hat{v}_{e,b}^{(t)}\right\|^2\right] + \frac{8(L_{\nabla a}')^2}{B}\sum_{e=1}^{E-1}\sum_{b=1}^{B}\sum_{t=2}^{T+1}\mathbb{E}\left[\left\|\tilde{y}_{e,b}^{(t)} - \hat{y}_{e,b}^{(t)}\right\|^2\right]
$$

$$
+ 4T\frac{\sigma^2}{B}\frac{(2-p)m-(1-p)m^2}{1-(1-p)(1-m)^2} + \frac{3}{m}\sum_{e=1}^{E}\mathbb{E}\left[\left\|\tilde{u}_e^{(1)} - \nabla_e\ell(\boldsymbol{w}^{(1)})\right\|^2\right].
$$

$\square$

Compared with Eq. (13), the most difference of Eq. (113) is that the noise term $\sigma^2$ has been replaced to $\frac{\sigma^2}{B}$. And thus we can simply obtain the convergence of Clapping in the large batch scenario.

**Lemma 21** (Convergence rate of Clapping-**FU** with large batch). *Suppose Assumption 1, 2, and 3 hold. Then for Algorithm 4 there exist $\gamma, m, p > 0$ such that:*

$$
\frac{1}{T}\sum_{t=1}^{T}\mathbb{E}\left[\left\|\nabla\ell(\boldsymbol{w}^{(t)})\right\|^2\right] \lesssim \frac{\sigma}{\sqrt{BT}} + \frac{1}{T(1-\omega_B)^{E-1}(1-\omega_F)^{E-1}}.
$$

(119)

Table 4: Hyperparameter configurations for LLaMA-2 models of different scales. 'Num_workers' denotes the total workers for a training pipeline.

| Parameters | Hidden size ($h$) | Heads ($a$) | Layers ($L$) | Vocabulary size ($V$) | Num_workers |
|---|---|---|---|---|---|
| 7B | 4096 | 32 | 32 | 32000 | 2 |
| 13B | 5120 | 40 | 40 | 32000 | 4 |
| 70B | 8192 | 64 | 80 | 32000 | 16 |

**Lemma 22** (Convergence rate of `Clapping`-**FC** with large batch). *Suppose Assumption 1, 2, and 3 hold. Then for Algorithm 4 there exist $\gamma, m > 0$ such that:*

$$
\begin{aligned}
\frac{1}{T}\sum_{t=1}^{T}\mathbb{E}\left[\left\|\nabla\ell(\boldsymbol{w}^{(t)})\right\|^2\right] &\lesssim \frac{\sigma^{\frac{4}{3}}}{(BT)^{\frac{1}{3}}(1-\omega_B)^{\frac{4(E-1)}{3}}(1-\omega_F)^{\frac{4(E-1)}{3}}} \\
&+ \frac{1}{T}\left(\frac{1}{(1-\omega_B)^{E-1}(1-\omega_F)^{E-1}} + \frac{\omega_F^2+\omega_B}{(1-\omega_B)^{2(E-1)}(1-\omega_F)^2} + \frac{1}{(1-\omega_F)^{2(E-2)-1}}\right).
\end{aligned}
\tag{120}
$$

## G  ADDITIONAL DETAILS ON MULTI-WORKER SCENARIOS

In Section 2.2, we present the forward and backward process of AQ-SGD (Wang et al., 2022b). However, it is noteworthy that the convergence analysis of (Wang et al., 2022b, Appendix A.1.) assume that the machine $e$ computes the gradient of activation by

$$
v_{e-1}^{(t)} = \nabla_1 a_e(\tilde{y}_{x,e-1}^{(t)}, w_e^{(t)})^\mathsf{T}\mathcal{C}\left[\frac{\partial a_{e+1}\circ\cdots\circ a_E}{\partial\tilde{y}_{x,e}^{(t)}}(\tilde{y}_{x,e}^{(t)}, w_{e+1}^{(t)}, \cdots, w_E^{(t)})\right],
\tag{121}
$$

where $a_{e+1}\circ\cdots\circ a_E$ denotes the composition of $a_{e+1}, \cdots, a_E$. However, the parameters $w_{e+1}^{(t)}, \cdots, w_E^{(t)}$ are held in different machines. Consequently, the computation of the partial gradient in the last term of (121) cannot avoids the communication between workers. Thus, it cannot be obtained losslessly through the communication compression during the propagation process. Thus, there causes a mismatch between the analysis in (Wang et al., 2022b) and the reality. In fact, the gradient of activation $v_{e-1}^{(t)}$ can be obtained by the back-propagation as:

$$
v_{e-1}^{(t)} = \nabla_1 a_e(\tilde{y}_{x,e-1}^{(t)}, w_e^{(t)})^\mathsf{T}\mathcal{C}\left[\nabla_1 a_{e+1}(\tilde{y}_{x,e}^{(t)}, w_{e+1}^{(t)})^\mathsf{T}\mathcal{C}\left[\cdots\mathcal{C}\left(\nabla_1 a_E(\tilde{y}_{x,E-1}^{(t)}, w_E^{(t)})\right)^\mathsf{T}\cdots\right]\right],
\tag{122}
$$

which calls for the analysis of error accumulation because of the multiple compression.

## H  MEMORY OVERHEAD ANALYSIS FOR LLMs PRE-TRAINING WITH `Clapping`

In this section, we present the analysis for the memory overhead introduced by `Clapping` in the pre-training tasks for LLMs.

**Basic setup.** We consider a pre-training task on LLaMA-2-based models with SwiGLU activation (Touvron et al., 2023) running on Nvidia A100 80G GPUs. The dataset is C4-en (Raffel et al., 2020), which is primarily intended for pre-training language models and word representations on a large scale. We use the T5-base tokenizer with a sequence length of $s = 4096$, resulting in a total of approximately $|\mathcal{D}| \approx 45.6M$ training samples. The microbatch size is $B = 16$, and the total batch size is set to 256. The optimizer used is Adam (Kingma, 2014), and we employ BF16 precision, where each parameter requires 2 bytes for storage.

If we assume the intermediate size $h_{\text{ff}} = \frac{8h}{3}$, we use vanilla gradient checkpointing (GCP) Chen et al. (2016) to save activations and do not use GQA for 70B models. The basic memory overhead for parameters, gradients, optimizer states, and activations is:

$$
\underbrace{4Vh + 48Lh^2}_{\text{parameters, gradients, optimizer states}} + \underbrace{LBsh}_{\text{Activation memory with GCP}}
$$

Table 5: The memory overhead for pre-training LLaMA-2 models with different model size. 'Basic M.' means the memory overhead for parameters, gradients, optimizer states, and activation memorys. 'Clapping M.' means the additional memory introduced by Clapping . 'AQ-SGD M.' means the additional memory introduced by AQ-SGD. 'Ratio' denotes the ration between the additional memory of Clapping and the basic memory.

| Parameters | Basic M. (GB) | Clapping M. (GB) | Ratio | AQ-SGD M. (GB) |
|------------|---------------|------------------|--------|----------------|
| 7B | 65.0 | 2.0 | 3.07% | 2850.0 |
| 13B | 118.1 | 7.5 | 6.35% | 10687.5 |
| 70B | 562.0 | 60.0 | 10.68% | 85500.0 |

For Clapping , it should cache the term $\tilde{y}_e$ in workers $e$ and $e+1$, as well as the term $\tilde{v}_{e-1}$ in workers $e-1$ and $e$. Since we often split the model at the end of some transformer block, both $\tilde{y}_e$ and $\tilde{v}_{e-1}$ have a size of Bsh. Thus, the memory overhead of Clapping is:

$$4 \times (\text{num\_workers} - 1)Bsh.$$

Similarly, the memory overhead of AQ-SGD can be expressed as:

$$2 \times (\text{num\_workers} - 1)|\mathcal{D}|sh.$$

We present the memory overhead analysis for pre-training LLaMA-2 models with different communication compression algorithms under pipeline parallelism in Table 5. It can be observed that the memory overhead introduced by Clapping is less than 11% of the basic memory even when the model size scales to 70B, which is acceptable for practical pre-training tasks. Meanwhile, AQ-SGD requires thousands of GBs to store the sample-wise cache, making it unsuitable for pre-training tasks.

# I  EXPERIMENTAL DETAILS

In this section, we present the details of our numerical experiments, which were discussed in Section 5. Additionally, we provide additional experimental results that were not included in the main text due to space limitations.

## I.1  SYNTHETIC LOGISTIC REGRESSION

Herein, we consider the following logistic regression task with the objective function given by:

$$f(w,b) = \mathbb{E}_{(\xi,\zeta)\sim\mathcal{D}} \left[ \ln(1 + \exp(-\zeta \cdot (\xi^\top w + b))) + C_r\|w\|^2 + C_r b^2 \right], \quad (123)$$

where $w$ and $b$ denote the regression parameters, and $C_r$ is a regularization parameter. $\mathcal{D}$ represents the finite sample set. Here, we the sample size $|\mathcal{D}|$. For each stochastic sample $(\xi,\zeta)$ in $\mathcal{D}$, $\xi$ is a 200-dimensional vector generated as the standard $\xi^* + \varepsilon$, where each element of $\xi^*$ is independently normal distribution drawn from n $\mathcal{N}(0, 0.5)$ and each element of $\varepsilon$ is independently drawn from $\mathcal{N}(0, 0.3)$. Let $C_r = 0.005$. Then, Equation (123) is strongly convex. We employ gradient descent to find the minimum of Equation (123), denote as $f^*$. Subsequently, we split the model into two parts according to the formulation in (1), where

$$y_1 = -\zeta \cdot (\xi^\top w + b), \quad y_2 = \ln(1 + \exp(y_1)) + C_r\|w\|^2 + C_r b^2.$$

After obtaining $y_1$ during the forward-propagation process, we introduce an error to $y_1$ as $\delta_1$ to simulate activation compression and use $\tilde{y}_1 = y_1 + \delta_1$ to compute $y_2$, where $\delta_1$ follows the uniform distribution $U(-0.2, 0.2)$. We set the batch size to 128. Then, we run the moving-average Stochastic Gradient Descent (SGD) algorithm for 200,000 iterations with different error compensation strategies, including direct compression without error feedback (**Compression**), compression with error feedback but without lazy sampling (**Compression + EF**), Clapping-**FC** , and Clapping-**FU** . We compare these strategies with the case of no compression (**No Compression**). The moving-average term is set to 0.9, and the step-size is initialized to 0.1. When the iteration number , the step-size is multiplied by 0.5, and the historical gradient is cleared. Figure 5 depicts the gap between the current loss and the optimal loss for different algorithms. Both direct compression and compression with error feedback but without lazy sampling fail to converge as the step-size decreases. However, both Clapping-**FC** and Clapping-**FU** can converge as the step-size decreases, and Clapping-**FU** convergence faster than Clapping-**FC** to a smaller minimum. The error introduced by activation compression only slows down the convergence process, which validates our theoretical findings in Section 4.2.

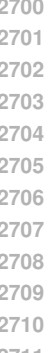

Figure 5: The gap between the current loss and the optimal loss for different algorithms for the logistic regression problem.

Table 6: The best test accuracy in training ResNet-18 on CIFAR-10.

| | Strategy | Direct comp. | Clapping with $p_t$ | | | | |
|---|---|---|---|---|---|---|---|
| | | | 0.2 | 0.4 | 0.6 | 0.8 | 1.0 |
| C. on | f5-b5 | 0.8604 (0.00170) | 0.8719 (0.01041) | 0.8889 (0.00257) | **0.9034** (**0.00117**) | 0.9016 (0.00066) | 0.8963 (0.00209) |
| | f6-b6 | 0.9024 (0.00709) | 0.8985 (0.00997) | 0.9119 (0.00271) | **0.9185** (**0.00466**) | 0.9150 (0.00373) | 0.9133 (0.00397) |
| | f6-b8 | 0.9056 (0.00558) | 0.9014 (0.00199) | 0.9109 (0.00496) | 0.9140 (0.00805) | **0.9184** (**0.00129**) | 0.9160 (0.00535) |
| C. off | f5-b5 | 0.8623 (0.00271) | 0.8733 (0.01010) | 0.8907 (0.00187) | **0.9045** (**0.00104**) | 0.9022 (0.00039) | 0.8982 (0.00197) |
| | f6-b6 | 0.9025 (0.00643) | 0.8986 (0.00978) | 0.9123 (0.00282) | **0.9186** (**0.00438**) | 0.9150 (0.00412) | 0.9137 (0.00373) |
| | f6-b8 | 0.9058 (0.00551) | 0.9015 (0.00179) | 0.9109 (0.00491) | 0.9139 (0.00798) | **0.9184** (**0.00121**) | 0.9161 (0.00533) |

'f[A]-b[B]' means compressing activation to A bits in forward propagation and compressing gradient to B bits in backward propagation. 'C. on' means taking the same compression during inference and 'C. off' means taking no compression.

### I.2 TRAINING RESNET-18 ON CIFAR-10

We trained a ResNet-18 (He et al., 2016) model on CIFAR-10 (Krizhevsky et al., 2009) dataset on a NVIDIA A100 80G GPU with communication compression algorithms for pipeline-parallel learning including directly compression and `Clapping`-**FC**. The basic setting is similar to that of (Rudakov et al., 2023). We split the model into 4 parts and used 3 direct quantization (Alistarh et al., 2017) with different bits to simulate the communication compression. We set the batch size to 128. We trained 100 epochs with directly compression and trained with `Clapping`-**FC** for the same number of iterations. We used the SGD optimizer with momentum 0.9 and weight decay 5e-4. The learning rate is initialized to 0.01 and scheduled by a cosine annealing scheduler with $T_{\max} = 200$. During inference, we obtain the test accuracy with the sample compression as training and without any compression, respectively.

Table 6 shows the average best test accuracy of directly compression and `Clapping` with different lazy sampling coefficient $p_t$ with different compression strategies over 3 independent runs. We can observe that the error feedback technique and lazy sampling strategy can improve the prediction accuracy in the communication compression of pipeline parallelism.

Table 7: The validation perplexity (↓) and accuracy (↑) for the fine-tuning task for LLaMA-2 13B under Wikitext with `Clapping`-**FC** with different number of workers ($E$).

| Number of workers ($E$) | Validation perplexity | Validation accuracy |
|:---:|:---:|:---:|
| 1 | 6.1325 | 0.5823 |
| 2 | 6.7896 | 0.5654 |
| 4 | 7.9840 | 0.5422 |
| 9 | 13.5134 | 0.4606 |

### I.3  FINE-TUNING LLMs

#### I.3.1  FINE-TUNING ON GLUE BENCHMARK

We fine-tune pre-trained RoBERTa-large (Liu et al., 2019) dataset on GLUE benchmark (Wang et al., 2018) with communication compression algorithms including `Clapping` and direct compression with EF21 (Richtárik et al., 2021) and compare then with fine-tuning without compression on two Nvidia A800 GPUs. We use the TopK compression with 30% elements at the middle of the networks. The batch size are all set 64 and the learning rate is 1e-5. For each dataset, we fine-tune the model for 10 epochs using both fine-tuning without compression and the EF21 algorithm. We also tune fine-tune the model for the same number of iterations as in 10 epochs by `Clapping` with lazy sampling coefficient $p = 0.5$. As Table 2 illustrates, `Clapping` outperforms EF21 in majority of tasks and even outperforms fine-tuning without communication compression in tasks including MRPC and RTE. And `Clapping` achieves the highest average score among all the algorithms.

#### I.3.2  FINE-TUNING LLaMA MODELS WITH TOPK COMPRESSOR

We fine-tuned a pretrained LLaMA-2 7B (Touvron et al., 2023) model and a LLaMA-3 8B (Grattafiori et al., 2024) model on Wikitext dataset (wikitext-2-raw-v1 version) (Merity et al., 2016) on two NVIDIA A100 80G GPUs with communication compression algorithms for pipeline-parallel learning including directly compression, AQ-SGD, EF21 and `Clapping`-**FC** . The block size was set to 1024. We used batch size 8, and we fine-tuned the model for 4 epochs for the competitive algorithms and we use the same iterations for `Clapping` , respectively. We use the SGD optimizer and FP16 for fine-tuning. The learning rate is initialized from $2 \times 10^{-5}$ and scheduled by a linear scheduler. We compare different compression algorithms with the compressor of Top-5% (Wangni et al., 2018). For each model, we independently repeat `Clapping` with different lazy sampling coefficient $p_t$ including $\{0.3, 0.4, 0.5\}$.

Table 3 has present the evaluation accuracy of different approaches under Top-5% compressor. It can be observed that `Clapping` outperforms other algorithms, including direct compression, EF21, and AQ-SGD. By tuning the low-rank coefficient $p$, **Clapping-FC achieves 95% communication saving with less than 0.5% error in practical fine-tuning tasks**.

**Ablations on the impact of the number of workers on performance.** In Section 4.2, we demonstrated that the error induced by communication compression accumulates exponentially with the number of workers, denoted as $E$. To validate this finding, we fine-tuned a pre-trained LLaMA-2 13B model (Touvron et al., 2023) on Wikitext dataset for 4 epochs using Top-20% compressor and `Clapping`-**FC** with varying numbers of workers, while maintaining the compression probability at $p = 0.5$. Specifically, we set $E = 1, 2, 4, 9$ workers in separate experiments. All other parameters were consistent with prior configurations, except for an increase in the initial learning rate to $5 \times 10^{-5}$ to amplify the error accumulation effect. Table 7 summarize the validation perplexity and validation accuracy across different worker counts. The results reveal that as the number of workers grows, the compression error accumulates, leading to a corresponding decline in validation performance.

#### I.3.3  FINE-TUNING WITH MULTIPLE COMPRESSION.

Here we present the experimental result in fine-tuning tasks with multiple compression. Different from the fine-tuning tasks introduced in Appendix I.3.1 and I.3.2, the experimental setup we take here is more **STRICT** than the practical fine-tuning tasks to present a comparsion for different algorithms.

Table 8: Evaluation perplexity of `Clapping` in GPT-2 fine-tuning with different lazy coefficient $p_t$.

| Strategy | $p_t$ | | | | | | |
|---|---|---|---|---|---|---|---|
| | 0.1 | 0.2 | 0.3 | 0.4 | 0.6 | 0.8 | 1.0 |
| S1 | 16.118 | 15.597 | 15.523 | **15.482** | 15.675 | 15.999 | 16.582 |
| | (0.095) | (0.070) | (0.013) | **(0.014)** | (0.015) | (0.014) | (0.417) |
| S2 | 13.432 | 12.992 | 12.713 | **12.558** | 12.630 | 12.645 | 12.756 |
| | (0.118) | (0.239) | (0.343) | **(0.352)** | (0.306) | (0.250) | (0.182) |
| S3 | 15.337 | 14.330 | **12.815** | 14.106 | 14.261 | 14.466 | 14.712 |
| | (0.187) | (0.372) | **(0.222)** | (0.208) | (0.218) | (0.231) | (0.246) |

**Fine-tuning GPT-2 on Wikitext.** We fine-tuned a pretrained GPT-2 (Radford et al., 2019) model on Wikitext dataset (wikitext-2-raw-v1 version) (Merity et al., 2016) on a NVIDIA A100 80G GPU with communication compression algorithms for pipeline-parallel learning including directly compression, AQ-SGD, `Clapping` without lazy sampling and `Clapping`-**FC** . As the experiment is taken in a single GPU, we simulate the communication compression by adding the corresponding error to the activation and gradient. The block size was set to 1024. We used batch size 8, and we fine-tuned the model for 8 epochs for the competitive algorithms and we use the same iterations for `Clapping` , respectively. We use the AdamW optimizer (Loshchilov, 2017) and FP16 for fine-tuning. The learning rate is initialized from $2 \times 10^{-5}$ and scheduled by a linear scheduler. We compare different compression algorithms under three compression strategies. These strategies integrated different compressor including TopK (Wangni et al., 2018), direct quantization (Alistarh et al., 2017), and Natural compression (Horvóth et al., 2022) on GPT-2 with different model size, which can be summarized as follow.

**Strategy S1** Basic model: GPT-2 small;
  Compression position: At the end of layer 2, 5, 8;
  Forward compressor: Top 40%;
  Backward compressor: Top 40%.

**Strategy S2** Basic model: GPT-2 medium;
  Compression position: At the end of layer 4, 10, 16;
  Forward compressor: Direct taking Natural Compression;
  Backward compressor: Quantizing to 8 bits, then taking Natural Compression.

**Strategy S3** Basic model: GPT-2 medium;
  Compression position: At the end of layer 4, 10, 16;
  Compressor in layer 4: Direct taking Natural Compression;
  Compressor in layer 10 and 16: Quantizing to 8 bits, then taking Natural Compression.

For each compression strategies, we independently repeat `Clapping` with different lazy sampling coefficient $p_t$ including $\{0.1, 0.2, 0.3, 0.4, 0.6, 0.8\}$ for 3 times and select the $p_t$ with the best average evaluation accuracy.

Figure 6 illustrates the evaluation accuracy and perplexity of `Clapping` with best $p_t$ as well as the other algorithms. It can be observed that `Clapping` outperforms direct compression. Moreover, lazy sampling can improve the evaluation accuracy. In the case, `Clapping` achieves a better accuracy than AQ-SGD because AQ-SGD suffers from a high compression error in the beginning.

And we also present the best evaluation accuracy and perplexity of `Clapping` , which is shown in Table 8. From Table 8, we can observe that a suitable $p_t$ like 0.3 or 0.4 can benefit the convergence and generalization in the fine-tuning tasks. And a large $p_t$ may harm the generalization, which matches our discussion in Section 4.2.

**Fine-tuning GPT-2 on arXiv abstracts.** We also respectively fine-tuned pretrained GPT-2 small models and GPT-2 medium models (Radford et al., 2019) on a dataset with 30K arXiv abstracts (Wang et al., 2022b) on 8 NVIDIA RTX 4090 GPUs with communication compression algorithms for pipeline-parallel learning including directly compression, `Clapping`-**FC** without lazy sampling

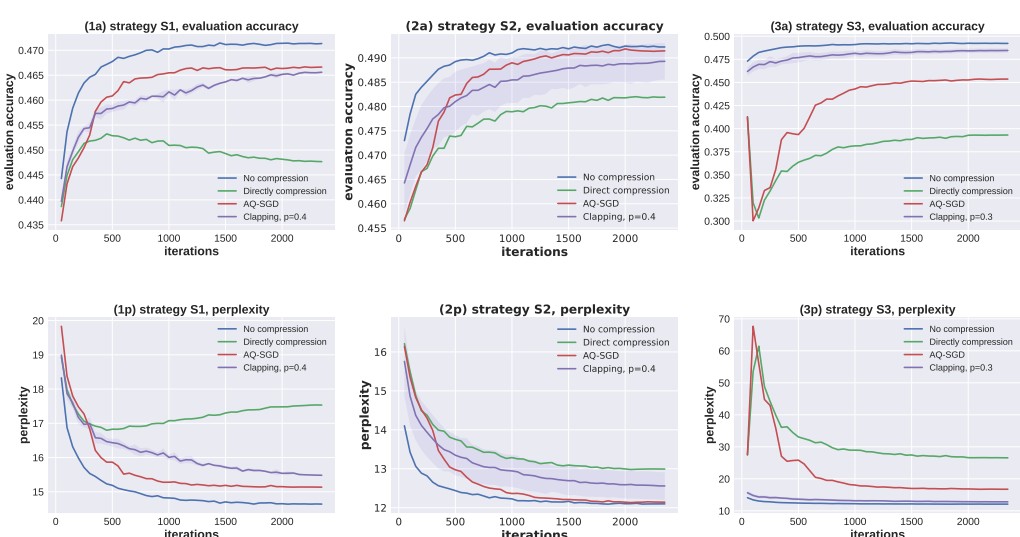

Figure 6: The evaluation accuracy and perplexity of GPT-2 fine-tuning with different compression strategies and difference compression algorithms. (Left: Strategy S1, Middle: Strategy S2, Right: Strategy S3.)

and `Clapping`-**FC** with lazy sampling. We also compared those approaches to the case with no compression. The block size was set to 1024. We set data parallel degree 8 with the macro-batch size 16 and the micro-batch size 2. As each micro-batch is computed in a single GPU, we simulate the communication compression by adding the corresponding error to the activation and gradient. And we fine-tuned the model for 8 epochs with other competitive algorithms and the same iterations with `Clapping`, respectively. We simulated the communication compression for each GPU by adding the compression hook with Top-K (Wangni et al., 2018) compressor that keep 50% of the elements. For GPT-2 small, we add the hook at the end of the 2-nd, 5-th, and 8-th transformer layer. For GPT-2 medium, we add the hook at the end of the 4-th, 10-th, and 16-th transformer layer.

We use the AdamW optimizer (Loshchilov, 2017) and FP16 for fine-tuning. The learning rate is initialized from $5 \times 10^{-5}$ and scheduled by a cosine scheduler. The lazy sampling coefficient $p_t$ for `Clapping` with lazy sampling was set to 0.5. Figure 7 illustrates the evaluation accuracy and perplexity. It can be observed that both `Clapping` with lazy sampling and `Clapping` without lazy sampling outperforms direct compression. Moreover, lazy sampling can make the evaluation accuracy and perplexity comparable to those of non-compressed case. Specifically, we can find that direct compression in GPT-2 medium suffers from the non-convergence but `Clapping` not, which can illustrates the benefit of error feedback technique.

**Fine-tuning LLaMA2-7B on Wikitext.** We fine-tuned a pre-trained LLaMA2-7B model (Touvron et al., 2023) on Wikitext dataset (wikitext-2-raw-v1 version) (Merity et al., 2016) on 4 NVIDIA A100 40G GPUs with communication compression algorithms for pipeline-parallel learning including directly compression, `Clapping`-**FC** without lazy sampling and `Clapping`-**FC** with $p = 0.3, 0.5$. We also compared those approaches to the case with no compression. We set data parallel degree 4 with the macro-batch size 8 and the micro-batch size 2. As each micro-batch is computed in a single GPU, we simulate the communication compression by adding the corresponding error to the activation and gradient. We simulated the communication compression for each GPU by adding the compression hook at the end of 8-th, 16-th, and 24-th transformer layers. The block size was set to 1024. We fine-tuned the model for 6 epoch with the other competitive algorithms and the same iterations with `Clapping`, respectively. We used the SGD optimizer with momentum 0.9 and FP16 for fine-tuning. The learning rate is $5 \times 10^{-5}$. And we used TopK (Wangni et al., 2018) that keeps $50\%$ of elements and natural compression (Horvóth et al., 2022) as the compressor, respectively. According to (Horvóth et al., 2022), FP16 training with natural compression can compression the activations and gradients to 6-bit, thus the communication overhead is 37.5% compared to the non-compressed scenario.

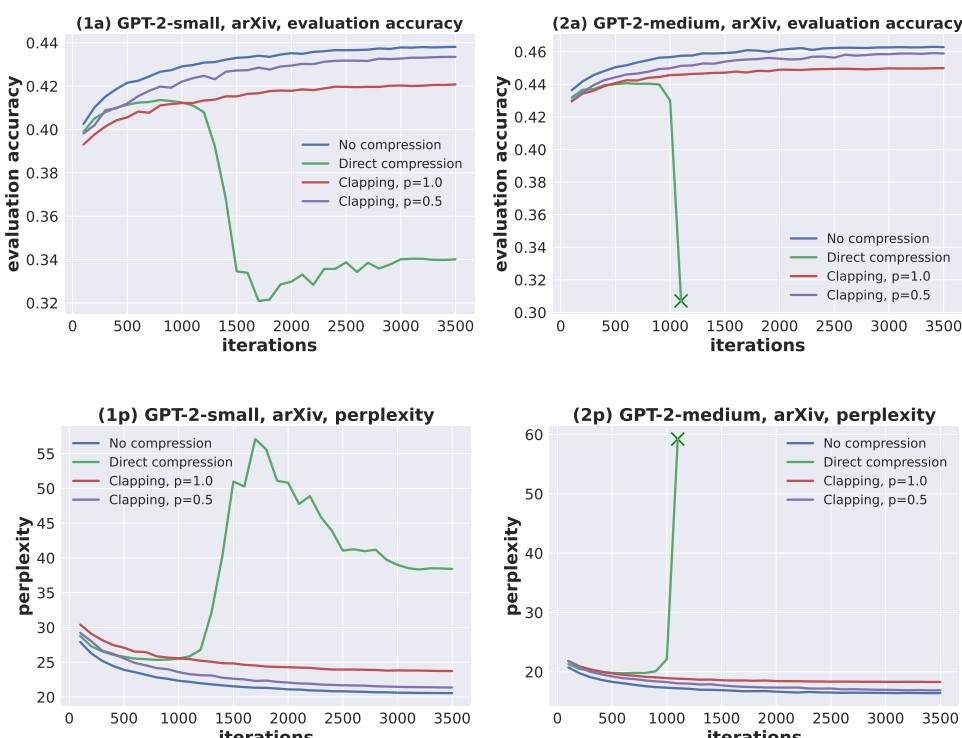

Figure 7: The evaluation accuracy and perplexity of GPT-2 small and GPT-2 medium fine-tuning by arXiv with Top50%. (Left: GPT-2 small, Right: GPT-2 medium.)

Figure 8 shows the evaluation accuracy and perplexity of different algorithms in LLaMA-2 fine-tuning tasks. It can be observed that `Clapping` outperforms directly compression in the evaluation scalability. It is also noteworthy that for both compressor, `Clapping` can achieve nearly the **SAME** evaluation accuracy and perplexity as the non-compressed case. Thus one can tune a suitable $p_t$ to let `Clapping` totally eliminate the negative impact of communication compression with more than 2× communication saving. Figure 9 illustrate the evaluation accuracy and perplexity of different compression algorithms. It can be observed the benefit of error feedback technique and lazy sampling strategy.

**Fine-tuning LLaMA2-7B on arXiv abstracts.** We fine-tuned a pre-trained LLaMA2-7B model (Touvron et al., 2023) on a dataset with 30K arXiv abstracts (Merity et al., 2016) on 4 NVIDIA A100 40G GPUs with communication compression algorithms for pipeline-parallel learning including directly compression, `Clapping`-**FC** without lazy sampling and `Clapping`-**FC** with $p = 0.5$. We also compared those approaches to the case with no compression. We set data parallel degree 4 with the macro-batch size 2 and the micro-batch size 8. We simulated the communication compression for each GPU by adding the corresponding error to the activation and gradient at the end of 8-th, 16-th, and 24-th transformer layers. The block size was set to 1024. We fine-tuned the model for 3 epochs with the other competitive algorithms and the same iterations for `Clapping`, respectively. We used the SGD optimizer with momentum 0.9 and FP16 for fine-tuning. The learning rate is $5 \times 10^{-5}$. And we used TopK (Wangni et al., 2018) that keeps $50\%$ of elements as the compressor.

## I.4 PRE-TRAINING LLMS

### I.4.1 PRE-TRAINING GPT-2 SMALL WITH MULTIPLE COMPRESSION.

We pre-trained a GPT-2 small (Radford et al., 2019) model on openwebtext (Peterson et al., 2019) on 8 NVIDIA RTX 4090 GPUs with 24GB of memory using communication compression algorithms for pipeline-parallel learning including directly compression and `Clapping` with lazy sampling and batch rule. Specifically, we cleared the cache for error feedback unless the former samples are kept.

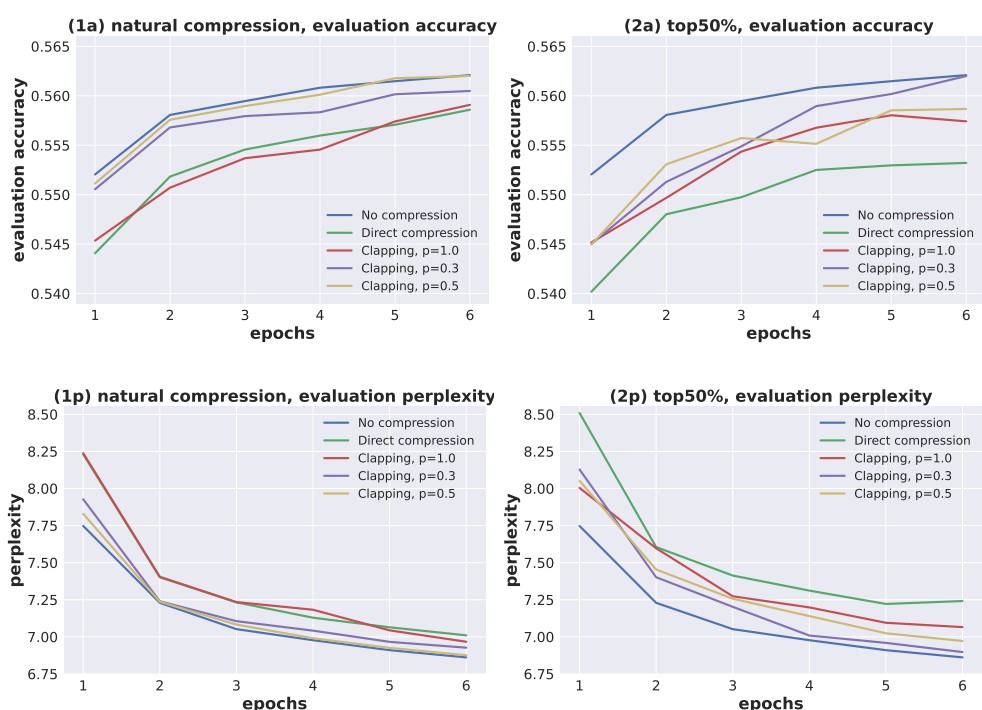

Figure 8: The evaluation accuracy and perplexity of LLaMA-2 fine-tuning with Wikitext-2 with different compressors and different compression algorithms. (Left: natural compression. Right: Top50%)

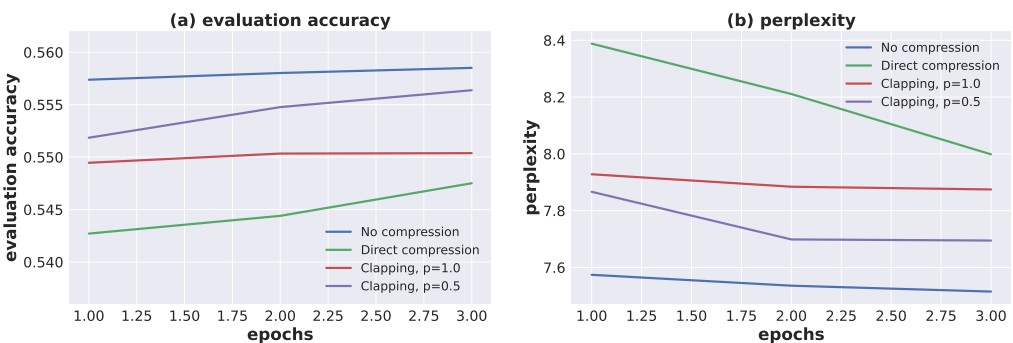

Figure 9: The evaluation accuracy of various approaches in LLaMA-2 fine-tuning task by arXiv abstracts.

The block size was set to 1024. We set data parallel degree 8 with the macro-batch size 64 and the micro-batch size 8. We simulated the communication compression for each GPU by adding the corresponding error to the activation and gradient. We use the AdamW optimizer (Loshchilov, 2017) and FP16 for training. We used natural compression (Horvóth et al., 2022) as compressor and added the compressor at the end of the 4-th and 8-th transformer layer. According to (Horvóth et al., 2022), FP16 training with natural compression can compression the activations and gradients to 6-bit, thus the communication overhead is 37.5% compared to the non-compressed scenario.

We trained the model for 130800 iterations for each algorithms as the total sample complexity is nearly equal to one epoch. The learning is initialized from $6 \times 10^{-4}$ with 2000 warm-up steps and scheduled by a cosine scheduler. For Clapping , we obtain the evaluation perplexity after 5000 steps with different lazy sampling coefficient $p_t \in \{0.2, 0.4, 0.45, 0.5, 0.55, 0.6, 0.8\}$ and finally selected the best one $p_t = 0.55$.

Table 9: Single-iteration time (ms) and speedup of Clapping-FC and non-compression across different network bandwidths

| Method | 100 MB/s | 200 MB/s | 300 MB/s | 400 MB/s | 500 MB/s |
|---|---|---|---|---|---|
| Clapping-FC | 572.68 | 460.18 | 422.68 | 403.93 | 392.68 |
| No Compression | 947.68 | 647.61 | 547.28 | 497.63 | 467.68 |
| Speedup | 1.65× | 1.41× | 1.30× | 1.23× | 1.19 |

Table 10: Comparison of training time ($\times 10^5$ s) to reach target validation perplexity for LLaMA-2 3B pre-training under bandwidth constraint

| Target validation perplexity | 18.0 | 17.5 | 17.0 |
|---|---|---|---|
| No compression | 2.6933 | 2.34036 | 2.69330 |
| Clapping-FC | 2.42791 | 2.08600 | 2.42791 |

Figure 3 has shown the evaluation perplexity, and evaluation accuracy of the pre-training task with different compression algorithms. They illustrates `Clapping` outperforms directly compression in both training and evaluation.

### I.4.2 PRE-TRAINING LLaMA-2 1B

We pre-trained a LLaMA-2 1B model (Touvron et al., 2023) on the C4 dataset (Raffel et al., 2020) using 8 NVIDIA A800 GPUs (80GB memory) with pipeline-parallel learning. Three communication compression strategies were compared: direct compression, `Clapping-`**FU** , `Clapping-`**FC** , and a baseline without compression. The model was split after the 16th transformer layer with a data parallel degree of 4. For communication compression, activations/gradients were quantized to 6 bits followed by Top-30% sparsification (also quantized to 6 bits). Compression was disabled during the initial 15,000 iterations. Network bandwidth was constrained to 100 MB/s throughout training. We employed the AdamW optimizer for 100,000 iterations, with coefficient $p_t$ initialized at 0.5 and progressively increased to 1.0 via cosine scheduling. Other hyperparameters followed (Zhao et al., 2024). The baseline perplexity and total time were derived from (Zhao et al., 2024), where total time was extrapolated from the first 2,000 training steps.

Figure 4 demonstrates the training dynamics. Direct compression failed to converge, while both `Clapping-`**FU** and `Clapping-`**FC** achieved a 2.2× speedup over the uncompressed baseline for the evaluation perplexity. This validates the effectiveness of our compression strategies for large language model pre-training.

**Discussion for the experimental result.** Figure 4 illustrates the training loss and validation perplexity of `Clapping` during pre-training. Notably, `Clapping` achieves validation perplexity comparable to that of the uncompressed scenario. This indicates that even though the lazy sampling technique in `Clapping` reduces the number of samples input to the model, an appropriate choice of the lazy sampling coefficient $p$ can preserve the model's expressive power with negligible degradation. This result demonstrates that `Clapping` , combined with lazy sampling, enables an effective trade-off between communication compression and model convergence and expressive capability.

In this experiment, the compressor achieves approximately 50% reduction in communication overhead. However, existing empirical results illustrate that communication compression can improve evaluation performance Ramasinghe et al. (2025). One possible explanation is that the noise introduced by the compressor implicitly helps avoid overfitting and leads to convergence to a minimum that exhibits better generalization ability. Thus, training with `Clapping` can achieve acceleration beyond mere communication overhead reduction. We plan to investigate how communication compression improves evaluation performance in future work.

### I.4.3 PRE-TRAINING LLaMA-2 3B

To validate the scalability of our proposed algorithm in large-scale pre-training scenarios, we conduct additional experiments using the LLaMA-2 3B model. The experimental setup maintains a bandwidth

Table 11: Comparison of validation loss across different compression methods

| Top-50% | 8-bit Quantization | Natural Compression | Hybrid Compressor |
|---------|--------------------|--------------------|--------------------|
| 5.0260 | 3.1425 | 3.0990 | 2.8944 |

constraint of 50 MB/s and employs identical compressor configurations to those used in the LLaMA-2 1B pre-training task. The compressor remains inactive during the initial 8,000 steps, and due to computational resource limitations, the model is trained for 50,000 steps in total. The microbatch size is set to 8. All other experimental parameters remain consistent with the LLaMA-2 1B baseline.

**Pareto-frontier analysis for a single step.** We firstly consider the overall time for a microbatch with different bandwidth of network. Table 9 compares the total iteration time (encompassing forward pass, backward pass, parameter update, and communication) of our Clapping-FC method against a non-compression baseline for a single step for microbatch, demonstrating the practical speedup gained through communication compression.

**Validation performance.** Table 10 presents a comparison of the total training time required by both the non-compression baseline and `Clapping`-**FC** method to achieve equivalent target validation perplexity levels. The results demonstrate a substantial improvement in training efficiency achieved by our proposed approach.

### I.4.4 ABLATIONS FOR DIFFERENT COMPRESSORS

To comprehensively evaluate the applicability of various compression algorithms in pre-training scenarios, we conducted extensive experiments using the LLaMA-2 250M model. The model was pre-trained with a sequence length of 256 and a batch size of 512, employing a learning rate of 0.001 over 40,000 training steps. The learning rate was scheduled using a cosine annealing strategy, and all computations were performed in BF16 precision. The model was partitioned into two segments with Clapping-FC applied for communication compression. We systematically compared four distinct compression methods: Top-K (50%), Direct 8-bit Quantization, Natural Compression, and a Hybrid Compressor (identical to the configuration described in Appendix I.4.2). While Natural Compression achieves superior compression ratios, the remaining three methods maintain approximately 50% compression rates.

As demonstrated in Table 11, the choice of compression algorithm significantly influences the final validation loss, underscoring the importance of selecting appropriate compression strategies to maximize the performance benefits of Clapping-FC in distributed training environments.

### LLMs USAGE.

In this paper, generative LLMs were used solely for writing polishing, such as grammar and wording improvements. All LLM-edited content was manually verified to ensure compliance with ICLR policies, and authors bear full responsibility for the submission.

