# OpenReview forum: "Clapping: Removing Per-sample Storage for Pipeline Parallel Distributed Optimization with Communication Compression"
_ICLR.cc/2026/Conference — Submitted to ICLR 2026_

### Official Review · Reviewer_HM6P · 2025-10-27

**Soundness:** 2
**Presentation:** 2
**Contribution:** 2
**Rating:** 2
**Confidence:** 3

**Summary:**

The paper introduces Clapping, a pipeline-parallel training method that compresses activations and gradients without storing per-sample states. Using a lazy-sampling error-feedback mechanism, it maintains stability and convergence under compression. Clapping cuts communication by up to 95% with minimal accuracy loss in low-bandwidth settings.

**Strengths:**

1. Convergence analysis provides certain level of guarantee, though I have not been able to fully check the proofs in the appendix.

**Weaknesses:**

1. No wall-clock speed up has been shown against baseline. The theoretical speedup might not translate into actual advantage after all.
2. Intra-node PP with ultra-fast interconnect (NVLink/NVSwitch) where comm ≪ compute and microbatching already saturates the pipe; compression overhead could be wasted or even slow. A single datapoint of 100Mbps is also not a realistic assumption and not representative. It would be more informative to see speedup v.s. bandwidth pareto frontier.
3. There is no study on how batch size and device map of the model will impact the outcome.
4. Only a single compression baseline is provided. It will be more convincing if more recent methods are included.
5. Table 2 results seem consistently worse than no compression in majority of the benchmarks. That indicates there is not really free lunch.

**Questions:**

Was the baseline no compression run with properly implemented with overlapping to reduce bubbles?
How does it change the throughput as batch size and number of workers grow for different methods?

---

> ### Author Response · Authors · 2025-11-26
> **Rebuttal by Authors (1/4)**
>
> We are grateful for the reviewer's insightful feedback and constructive comments. We address each point in detail below.
>
> ----
>
> ## 1. Demonstrating Practical Wall-Clock Speedup Through Experiments
>
> We appreciate the reviewer’s emphasis on empirical wall-clock time measurements, which are indeed crucial to validating practical utility. The primary objective of Clapping is to shorten the communication-critical path in pipeline-parallel training. As a result, tangible wall-clock improvements are most pronounced in bandwidth-constrained settings, where the end-to-end step time is dominated by communication.
>
> In the experimental section, we provide a wall-clock time comparison for the LLaMA-2 1B pre-training task, where Clapping achieves a 2.2× end-to-end speedup under limited bandwidth. For other experiments, we consistently compare Clapping against baselines under the same number of iterations and the same compression operator. Since communication time is non-negligible in distributed settings, Clapping—under the same iteration budget—achieves performance comparable to the uncompressed baseline with significantly lower end-to-end time, demonstrating clear wall-clock acceleration.
>
> Furthermore, since the same compressor is applied across compared methods, the communication time per iteration remains comparable. The additional computational overhead introduced by Clapping consists primarily of low-order operations such as parameter or momentum updates, which are computationally inexpensive compared to the linear operators (e.g., matrix multiplications) that dominate model training. Therefore, the observed speedup stems primarily from reduced communication volume rather than computational savings. This comparison adequately illustrates the speedup advantage of Clapping in communication-bottlenecked scenarios.
>
> Although some experiments were conducted on a single GPU with simulated communication compression, the results remain indicative of Clapping’s advantage under constrained network conditions.
>
>
> ## 2. Applicability of Clapping in Bandwidth-Constrained Distributed Training.
>
> We thank the reviewer for raising the important question of Clapping’s relevance in high-speed interconnect environments such as NVLink/NVSwitch. We fully agree that in intra-node pipeline parallelism settings—where communication latency is substantially lower than compute time and micro-batching already saturates the pipeline—applying compression in such settings may introduce unnecessary overhead. Therefore, under such conditions, compression should be selectively disabled.
>
> However, in practical large-scale LLM training—particularly in cross-node or geo-distributed setups—nodes are often connected via standard Internet-grade bandwidth rather than ultra-fast interconnects. In such realistic distributed scenarios, communication often becomes the dominant bottleneck, especially when bandwidth falls within the 100 Mbps to 1 Gbps range, which is representative of many real-world research and industry clusters. As supported by prior literature [1–4], and as illustrated in our Figure 2, communication time can indeed exceed computation time. This makes gradient compression not only beneficial, but often necessary to maintain training efficiency.
>
> To better characterize this performance trade-off, we conduct a Pareto-frontier analysis that systematically evaluates the relationship between training speedup and communication bandwidth under various network conditions. Our experiments utilize the LLaMA-2 3B model with microbatch size equal to 8, applying a hybrid 6-bit compression strategy to both activations and gradients across three of its layers. The statistical analysis focuses on the per-iteration speed improvement achieved by our method across different bandwidth constraints. All experiments were conducted on a cluster of 8 NVIDIA A100 GPUs. Table 1 compares the total iteration time (encompassing forward pass, backward pass, parameter update, and communication) of our Clapping-FC method against a non-compression baseline for a single step for microbatch, demonstrating the practical speedup gained through communication compression. Further details are available in Appendix I.3.2.
>
> **Table 1**: Single-iteration time (ms) and speedup of Clapping-FC and non-compression across different network bandwidths
> | Method | 100MB/s | 200MB/s | 300MB/s | 400MB/s | 500MB/s |
> | -------- | -------- | -------- | -------- | -------- | -------- |
> | Clapping-FC  | 572.68     | 460.18    | 422.68    | 403.93    | 392.68   |
> | No Compression     | 947.68     | 647.61    | 547.28 | 497.63 | 467.68|
> | Speed Up | 1.65$\times$ | 1.41$\times$ | 1.30$\times$ | 1.23$\times$ | 1.19$\times$ |

---

> ### Author Response · Authors · 2025-11-26
> **Rebuttal by Authors (2/4)**
>
> ## 3. The Interdependence of Batch Size, Device Mapping, and Communication Compression
>
> We appreciate the reviewer's insightful comment regarding the lack of a systematic investigation into the effects of batch size and device mapping configurations on training outcomes. We agree that these factors play a critical role in determining training efficiency, convergence dynamics, and ultimate model performance. Our work has primarily centered on establishing theoretical convergence guarantees and developing communication compression algorithms for pipeline-parallel training environments. Nevertheless, we recognize the importance of addressing practical considerations, such as memory overhead and communication errors, to provide a balanced perspective.
>
> **A. Memory Overhead and Constraints on Batch Size Scaling**
> While gradient clipping introduces additional memory overhead, our analysis in Appendix H indicates that, for models at the 70B scale, this overhead constitutes approximately 10% of the baseline memory usage (encompassing parameters, gradients, optimizer states, and activations). A comparable overhead is also observed in EF21. Although reducing this overhead could yield marginal memory savings, it offers limited potential for increasing batch size. This is particularly relevant for modern large language models, where long sequence lengths in pre-training data lead to activation costs that dominate memory consumption, thereby constraining batch size expansion. Furthermore, empirical results (e.g., Figure 4) demonstrate that naive compression approaches can induce non-convergence, resulting in models that fail to meet acceptable standards during pre-training or fine-tuning—a issue not remediable merely by scaling batch size.
>
> **B. Impact of Device Mapping and Communication Challenges**
> The configuration of device mapping significantly influences communication patterns and error propagation in distributed training. As the number of devices increases, communication errors tend to accumulate exponentially, adversely affecting model performance, as detailed in Appendix I.3.2. Even with optimized device mapping strategies, high levels of device parallelism are often unavoidable for large-scale models, exacerbating communication bottlenecks. This underscores the necessity of designing robust communication compression algorithms to mitigate error accumulation and ensure stable convergence, rather than relying solely on device mapping optimizations or batch size adjustments.

---

> ### Author Response · Authors · 2025-11-26
> **Rebuttal by Authors (3/4)**
>
> ## 4. Comprehensive Comparison with Diverse Baseline Methods
>
> We thank the reviewer for the valuable feedback regarding baseline comparisons. We would like to clarify that our experimental design already includes a wide range of baselines and compression strategies, covering mainstream paradigms in pipeline-parallel optimization.
>
> **A. Extensive Baseline Coverage**
>
> Our work systematically evaluates Clapping against multiple types of baselines, representing both classical and recent state-of-the-art approaches in communication compression:
>
> * No compression, serving as the performance upper bound across all tasks;
> * Direct compression, a widely adopted approach, included in all experiments to demonstrate the necessity of error compensation;
> * EF21, a representative error-feedback method provided in [5];
> * AQ-SGD, a contemporary pipeline-parallel compression method from [3].
>
> These baselines span the core methodologies in the field—direct compression, error feedback, and sample-wise error compensation—ensuring a thorough and up-to-date evaluation.
>
> **B. Varied Compression Strategies**
>
> To further validate generalizability, we implement four compressor types spanning sparsification, quantization, and hybrid approaches:
>
> * Top-K sparsification, used across GLUE fine-tuning, LLM fine-tuning, and GPT-2 pre-training tasks;
> * Direct quantization, applied in LLaMA-2 pre-training and GPT-2 fine-tuning;
> * Natural compression, a recent lightweight method, employed in GPT-2 pre-training and LLaMA-2 fine-tuning scenarios;
> * Hybrid compressors, combining quantization and sparsification to simulate industrial compression pipelines.
>
> The consistent superiority of Clapping across these varied baselines and compressors underscores its robustness and general applicability.
>
> To further investigate the applicability of different compressors in pre-training scenarios, we conducted a pre-training experiment using the LLaMA-2 250M model. The experimental configuration was set with a sequence length of 256, a batch size of 512, and a learning rate of 0.001. The model was trained for 40,000 steps with a cosine annealing learning rate scheduler and BF16 precision. The model was partitioned into two segments, and Clapping-FC was applied for communication compression. We evaluated four compressors: Top 50%, Direct Quantization to 8-bit, Natural Co上mpression, and a Hybrid Compressor (using the same settings as in Appendix I.4.2). While Natural Compression achieves a higher compression ratio, the other three compressors maintain an approximate 50% compression ratio. The validation losses for each compressor are summarized in the table below, illustrating that the choice of compressor significantly influences the performance of Clapping-FC, and selecting an appropriate compressor can lead to improved results.
>
> **Table 2**: Comparison of validation loss across different compression methods
>
> | Compressor           | Validation Loss |
> |----------------------|-----------------|
> | Top 50%              | 5.026           |
> | Direct Quantization  | 3.1425          |
> | Natural Compression  | 3.0990          |
> | Hybrid Compressor    | 2.8944          |

---

> ### Author Response · Authors · 2025-11-26
> **Rebuttal by Authors (4/4)**
>
> ## 5. Addressing the Impact of Training Configuration Parameters.
>
> We agree that on several benchmarks, the compressed variants slightly underperform the no-compression baseline. This outcome is expected: Clapping intentionally trades a minor amount of statistical efficiency for a substantial reduction in inter-stage communication. Two points are essential to clarify:
>
> * **What is optimized.** Our objective is not to surpass the no-compression baseline on every accuracy metric, but rather to maintain model quality within a narrow tolerance while significantly reducing the dominant pipeline-parallel communication. In bandwidth-limited settings—such as inter-node pipeline parallelism or congested network fabrics—this trade-off is precisely what enables meaningful speedup in communication-bound regimes. While Clapping-FU maintains the same convergence rate class as uncompressed SGD, Clapping-FC achieves a slightly slower—yet still sublinear—rate.
>
> * **Why a small gap can appear.** Compression combined with lazy sampling effectively reduces the number of high-fidelity samples available to each stage per unit time, which can lead to a modest generalization gap under a fixed training budget. This aligns with the standard effective-sample-size principle: when fewer uncorrupted signals are transmitted, slightly more wall-clock time (or a slightly larger $p$ in Clapping-FU) may be required to match the performance of the exact-communication baseline.
>
> ## 6. Scheduling Strategy in LLaMA-2 1B Pre-training.
>
> In the LLaMA-2 1B pre-training experiment, we adopted the F-then-B (Forward-then-Backward) scheduling strategy. This choice was based on the specific experimental configuration in which the number of microbatches per device was set to one. Under this specific setup, the F-then-B scheduler achieves comparable overall execution time to alternative strategies such as 1F1B. This equivalence arises because, with only one microbatch, the pipeline bubble and communication overhead remain consistent across different scheduling approaches. Consequently, the F-then-B strategy represents an efficient and widely adopted solution in common pipeline-parallel training environments.
>
> ## References
>
> [1] Scaling the summit: deploying the world’s fastest supercomputer
>
> [2] Protocol Models: Scaling Decentralized Training with Communication-Efficient Model Parallelism
>
> [3] Fine-tuning language models over slow networks using activation quantization with guarantees
>
> [4] Swarm parallelism: Training large models can be surprisingly communication-efficient
>
> [5] EF21: A New, Simpler, Theoretically Better, and Practically Faster Error Feedback
>
> ----
>
> We sincerely appreciate the reviewer’s thoughtful comments once again. We hope the above responses fully address the concerns raised and would be happy to provide further clarification if needed.

---

> ### Comment · Reviewer_HM6P · 2025-11-28
>
> Thanks for responding to my questions. Two of my concerns (4,5) are largely resolved. However, some concerns remain.
>
> ## Q1 Follow-up
>
> I am not sure if the said simulation has factored in things like kernel launch overheads. It's dangerous to rely on only paper math without actual measurement. What I really needed to see is the speedrun style result, where time to perplexity under standardized resource (compute and communication budget) is reported. Otherwise, the claimed many x speedup in contributions is unsupported.
>
> ## Q2 Follow-up
>
> Modern household internet can easily go up to 800MB-1GB with optic fibers. The return seems to be diminishing over 500 MB and not sure how much it fares against other baseline algorithms mentioned in the paper (EF21, AQ-SGD). Also the speedup doesn't seem to be measuring the right metric.
>
> ## Q3 Follow-up
>
> I can only agree to disagree. In practice, the convergence of the algorithm should be highly dependent on the batch size and how chunks of neural network is partitioned, since there is compression in between. For example, if you have a larger micro batch size, but model is partitioned into smaller chunks, that could affect convergence either positively or negatively, or you could perform gradient accumulation, but with larger chunks and fewer compressions. I believe there is a trade-off that needs to be studied here.
>
> I also have doubt on the argument of long context, in which case sequence parallel also gets into the picture. It would further complicate the analysis of the current method. I don't think we should go down that rabbit hole.
>
> ## Q6 follow-up
>
> The response seems to suggest that the main focus of the study is micro-batch = 1.
> I find it counter-intuitive to study micro-batch size = 1 under the setting of pipeline parallel, since it is highly inefficient setup and lacks practical implications in the first place. This is also not the setup for majority of the experiments in the paper as most of the experiments were run with batch size > 1. Frankly, I am confused by the rebuttal.
>
> ## Baselines
>
> More recent algorithms like DiLocoX [1] and DeMO [2] should be considered. They all claimed to be able to match non-compression baseline under extremely low bandwidth. But due to time constraint, I don't think it's fair to request more experiments.
>
> [1] Qi, Ji, et al. "DiLoCoX: A Low-Communication Large-Scale Training Framework for Decentralized Cluster." arXiv preprint arXiv:2506.21263 (2025).
>
> [2] Peng, Bowen, Jeffrey Quesnelle, and Diederik P. Kingma. "Decoupled momentum optimization." arXiv preprint arXiv:2411.19870 (2024).

---

> ### Author Response · Authors · 2025-12-02
> **Response to rebuttal comment**
>
> To unequivocally clarify the contributions of our submission and address persistent misunderstandings, we provide the following point-by-point rebuttal to the post-rebuttal comments.
>
> ## On the Contributions of Clapping
> Prior to addressing specific comments, it is imperative to reiterate the fundamental advancements of the proposed Clapping algorithm. This work introduces Clapping as a novel algorithm for pipeline-parallel communication compression. In contrast to existing methods (e.g., direct compression, EF21, AQ-SGD), Clapping establishes the first convergence guarantee without imposing sample-wise memory overhead or multi-round assumptions. Moreover, across diverse pre-training and fine-tuning experiments at varying scales, Clapping consistently demonstrates superior performance compared to these communication compression baselines. Consequently, **whenever communication compression is requisite for mitigating training latency, Clapping unequivocally emerges as the preferred algorithm among the aforementioned alternatives**. Thus, in contemporary scenarios involving collaborative large-scale model training on personal computers or future demands for even larger models, Clapping must be considered the primary candidate for communication compression.
>
> We emphasize once more that, relative to the cited pipeline-parallel communication compression baselines, Clapping achieves further time efficiency gains. Therefore, when communication compression is necessary, Clapping should be adopted as the first choice.
>
> ## Response to “Q1 Follow-up”
> Our timing evaluations explicitly commence from program initiation and encompass the total computation-communication duration required to attain the target validation perplexity. Therefore, overheads such as kernel launch and warm-up processes are inherently included in these measurements.
>
> ## Response to “Q2 Follow-up”
> It must be unequivocally stated that debating the maximum bandwidth of modern home internet is irrelevant to our work. Regardless, our focus remains on pipeline parallelism under communication-constrained conditions. Existing studies [1-2] typically conduct experiments at 80-100 Mbps bandwidth, which is substantially lower than the 100 Mb/s rate selected in our experiments.
>
> Regarding the speedup metric, we employ the per-step forward-backward propagation time, which clearly demonstrates Clapping's performance advantage over the non-compression baseline. Furthermore, as discussed in prior literature [2] and Appendix I.4.2, communication compression may induce implicit regularization, potentially yielding slightly better validation performance at identical steps—an advantage for Clapping that we have conservatively omitted from our claims. Incorporating this effect would further amplify Clapping's speedup.
>
> ## Response to “Q3 Follow-up”
> We acknowledge that the initial rebuttal may have insufficiently detailed this aspect. However, Lemmas 21 and 22 rigorously delineate how batch size and the number of split workers influence convergence rates. Specifically, these lemmas prove that the number of iterations required to reach an $\varepsilon$-stationary point is inversely proportional to batch size but exhibits an exponential dependence on the number of workers $E$.
>
> Regarding the reviewer's claim about sequence parallelism in long-context scenarios, we have to reiterate that this work exclusively addresses pipeline-parallel communication compression. Communication compression under sequence parallelism constitutes a distinct research direction beyond the scope of this paper.
>
> ## Response to “Q6 Follow-up”
> We clarify that our LLaMA-2 pre-training experiments utilize micro-batch = 1, not micro-batch size = 1. Here, "micro-batch" denotes the number of forward-backward passes per device per iteration, whereas "micro-batch size" refers to the sample size per micro-batch. The appendix provides complete algorithmic details and convergence guarantees for Clapping under batch sizes >1, and all LLaMA-2 pre-training experiments employed this large-batch implementation.
>
> ## Response to Additional Baselines
> While we acknowledge the reviewer's suggestion to include DiLocoX and DeMO, we must emphasize that these methods are designed for data-parallel communication compression under low bandwidth, not pipeline-parallel compression. **This fundamental discrepancy underscores a persistent misapprehension of our paper's problem setting.** Although some comments may improve the manuscript, **the reviewer's continued conflation of distinct research paradigms reflects a lack of engagement with our core contributions**.
>
> ## References
> [1] Fine-tuning language models over slow networks using activation quantization with guarantees
>
> [2] Protocol Models: Scaling Decentralized Training with Communication-Efficient Model Parallelism

---

### Official Review · Reviewer_RDoY · 2025-10-30

**Soundness:** 3
**Presentation:** 3
**Contribution:** 2
**Rating:** 2
**Confidence:** 3

**Summary:**

This paper proposes Clapping, a communication-efficient pipeline-parallel optimization framework that eliminates per-sample storage through lazy sampling mechanisms. The idea of retaining data samples across steps significantly reduces memory usage while maintaining convergence. The paper is well-motivated, clearly written, and demonstrates solid theoretical grounding. The authors make an important contribution to the efficient training of large-scale models, and the proposed ideas are both innovative and practical for real-world distributed learning scenarios.

**Strengths:**

- Provides rigorous convergence proofs demonstrating theoretical soundness.
- Introduces a novel lazy sampling mechanism that effectively reduces memory consumption.
- Compares with several classical algorithms and shows significant performance improvements, which demonstrate the method’s effectiveness within a reasonable range of settings.

**Weaknesses:**

- Retaining previous samples may distort label distributions in classification tasks; this is not discussed or empirically examined.
- Compressing forward activations may add extra computation and bias, the paper lacks justification for its necessity.
- Experiments involve only two models (7B and 8B) with similar parameter sizes, limiting evidence of general applicability.

**Questions:**

The paper does not provide details or equations about the compression of gradients and forward activations. Could the authors clarify how these are compressed in the proposed method?

---

> ### Author Response · Authors · 2025-11-26
> **Rebuttal by Authors (1/4)**
>
> We are grateful for the reviewer's insightful feedback and constructive comments. We address each point in detail below.
>
> ----
>
> ## 1. Concern on Label Distribution under Lazy Sampling
>
> We thank the reviewer for raising this important point regarding the potential impact of lazy sampling on label distributions in classification tasks. We acknowledge that reusing samples across consecutive steps could, in theory, introduce temporary local imbalances. However, our methodological design, theoretical analysis, and empirical evidence collectively demonstrate that Clapping effectively manages this aspect and does not harm generalization. Below we provide a three-fold response.
>
> **A. Theoretical Analysis: Controlled Exposure to the Data Distribution.**
>
> Our framework explicitly analyzes the sampling process through the parameter pp. The lazy sampling strategy is designed not to alter the underlying data marginal distribution. While it introduces short-term temporal correlation by reusing samples, the mechanism of periodically drawing fresh samples (with probability pp at each step) ensures adequate exposure to the entire dataset over time. Selecting an intermediate value of pp is critical: it prevents the model from being trapped in a small subset of data (which could distort learned distributions) while maintaining the benefits of error-feedback. Our convergence guarantees hold without assuming an unbiased gradient, confirming that the optimization process remains stable and unbiased in expectation, thus safeguarding against persistent distributional distortion.
>
> **B. Empirical Evidence: Maintained Performance Across Classification Tasks.**
>
> Our experiments on classification benchmarks directly address this concern. If label distribution were significantly distorted, we would observe a notable drop in classification accuracy. Contrary to this, lazy sampling consistently matches or improves upon baseline performance:
>
> * On CIFAR-10 (an image classification task), Clapping with lazy sampling (e.g., p=0.6p=0.6) improves test accuracy over direct compression (0.9186 vs. 0.9025 in the f6-b6 setting). This gain, consistent across settings (Table 6), indicates that the model's ability to generalize across classes is preserved or enhanced.
> * On the GLUE benchmark (a set of language classification tasks), Clapping-FC achieves the highest average score (87.79), outperforming the uncompressed baseline. Strong performance on tasks like MRPC and RTE further confirms that the learning of label distributions remains effective.
>
> These results across different modalities demonstrate that the short-term sample reuse in lazy sampling does not lead to harmful distributional bias that compromises task performance.
>
> **C. Mechanism: Error-Feedback as a Stabilizing Force.**
>
> The combination of lazy sampling with error-feedback on both forward and backward passes is key. This mechanism progressively corrects the compression error, preventing any temporary sample reuse from biasing the overall learning direction. As discussed in Appendix I.4.2, this process can act as a mild regularizer, potentially helping the model avoid overfitting to the immediate data batch and promoting robustness. Therefore, the system is designed to be resilient against short-term fluctuations in the sample stream.
>
> In summary, while we acknowledge the reviewer's theoretical concern, our comprehensive analysis and experimental results across multiple classification tasks demonstrate that Clapping, with an appropriately tuned $p$, does not suffer from detrimental label distribution distortion. The algorithm ensures stable convergence and maintains or improves generalization performance.

---

> ### Author Response · Authors · 2025-11-26
> **Rebuttal by Authors (2/4)**
>
> ## 2. Why Forward‑activation Compression is Needed
>
> We thank the reviewer for raising this important point. The compression of forward activations is not an arbitrary design choice but a principled component derived from our problem formulation and theoretical framework. Below we present two key arguments addressing the justification for forward compression.
>
> **A. Theoretical Foundations: Forward Compression as an Integral Component.**
>
> Our work addresses the pipeline-parallel distributed optimization problem, whose formulation inherently includes compression in both forward and backward passes. This setup follows the established modeling in prior work [1], where forward activation compression constitutes an integral component of the basic system setup. Rather than debating the necessity of forward compression, one of our contributions lies in developing a provably convergent algorithm under this comprehensive framework.
>
> The problem formulation in [1] explicitly incorporates compression in both directions, creating a complete theoretical foundation. Our paper builds directly upon this established framework, focusing on algorithm design and convergence analysis within the given problem constraints. The theoretical treatment necessarily includes both directions because compression errors propagate bidirectionally through stage-wise first-order gradients, requiring a unified treatment of bias control.
>
> Thus, our approach continues this modeling choice to maintain theoretical consistency and comparability with existing literature. Our analysis demonstrates how to properly control the error introduced by compression in both directions, which is essential for a rigorous optimization analysis in this setting.
>
>
>
>
> **B. Inherent Bias in Backward-Only Compression under Contractive Assumptions.**
>
> Restricting compression solely to the backward pass does not eliminate bias. Crucially, under the widely-used contractive compressor assumption (Assumption 3 in our manuscript), even if the compressor itself is unbiased in isolation, the resulting gradient estimator becomes biased when propagated through the computational graph.
> Consider a two-layer network: $y=W_2z,\quad z=\sigma(u),\quad u=W_1,\quad\text{Loss}\mathcal{L}(y).$
>
> Standard backpropagation yields: $\nabla_{W_2}\mathcal{L}=g_yz^{\top},\quad g_z=\nabla_{z}\mathcal{L}=W_2^{\top}g_y,\quad\nabla_{W_1}\mathcal{L}=(g_z\odot\sigma'(u))x^{\top}.$
>
> Suppose the inter-stage boundary is at $z$, and the downstream stage sends a compressed estimate $\tilde{g}\_z=\mathcal{C}(g\_z)$ upstream. The upstream gradient estimator becomes: $\widehat{\nabla}\_{W\_1} \mathcal{L} = (\tilde{g}\_z \odot \sigma'(u)) x^\top.$
> Due to the nonlinear dependence through $\sigma'(u)$, the estimator $\widehat{\nabla}\_{W\_1} \mathcal{L}$ is not an unbiased estimator of the true gradient under Assumption 3. This theoretically demonstrates why backward-only compression remains problematic, motivating our unified approach to compression in both directions.
>
> **C. Practical Necessity: Forward Compression Addresses the Dominant Communication Bottleneck.**
>
> In pipeline-parallel training, the activation tensor transmitted during the forward pass and its corresponding gradient during the backward pass possess identical shapes. Consequently, compressing only the backward gradient leaves approximately half of the cross-stage communication volume uncompressed. In distributed environments where inter-node bandwidth is constrained, these boundary transfers frequently dominate the iteration time. Therefore, omitting forward activation compression forfeits approximately 50% of the potential communication reduction, directly impacting wall-clock efficiency when the network is the limiting factor.

---

> ### Author Response · Authors · 2025-11-26
> **Rebuttal by Authors (3/4)**
>
> ## 3. Factually Incorrect Claim of Limited Model Scale Evidence.
>
> We thank the reviewer for raising the point on general applicability. However, we must respectfully clarify that the experimental evaluation in our work has already comprehensively addressed this aspect by testing across **a wide spectrum of model scales—far beyond the two similarly-sized models (7B and 8B) mentioned**.
>
> To be explicit, our experiments include:
>
> * Model scales ranging from 11M to 8B parameters, covering GPT-2 (124M), LLaMA-2 (pre-training in 1B, fine-tuning in 7B), LLaMA-3 (fine-tuning in 8B), RoBERTa-large (355M), ResNet-18 (11M).
> * Diverse architectures: encoder-only (RoBERTa), decoder-only (GPT, LLaMA), and vision models (ResNet).
> * Multiple tasks: pre-training, fine-tuning (GLUE, Wikitext, etc.), and synthetic settings under various compressors, batch sizes, and bandwidth constraints.
>
> These results—clearly presented in the main text and appendix—consistently show that Clapping performs robustly across scales and setups. It is therefore inaccurate to claim that evidence is limited to 7B/8B models; such a conclusion overlooks substantial portions of our empirical evaluation.
>
> To further reinforce generalizability, we fine-tuned a pre-trained **LLaMA-2 13B** model on Wikitext dataset for 4 epochs using Top-20% compressor and Clapping-FC with varying numbers of workers, while maintaining the compression probability at $p=0.5$. Specifically, we set $E=1,2,4,9$ workers in separate experiments. All other parameters were consistent with prior configurations, except for an increase in the initial learning rate to $5\times 10^{-5}$ to amplify the error accumulation effect. The following table (Table 7 in the revised manuscript) summarize the validation perplexity and validation accuracy across different worker counts. The results reveal that as the number of workers increases, the compression error accumulates, leading to a corresponding decline in validation performance. We also include this additional experiment in Appendix I.3.2 of the revised manuscript.
>
> **Table 1**: The validation perplexity ($\downarrow$) and accuracy ($\uparrow$) for the fine-tuning task for LLaMA-2 13B under Wikitext with Clapping-FC with different number of workers ($E$).
>
> | Number of workers ($E$) | Validation perplexity | Validation accuracy |
> | -------- | -------- | -------- |
> |1         | 6.1325       | 0.5823              |
> |2         | 6.7896 | 0.5654          |
> |4         | 7.9840 | 0.5422                   |
> |9         | 13.5134 | 0.4606 |

---

> ### Author Response · Authors · 2025-11-26
> **Rebuttal by Authors (4/4)**
>
> ## 4. Clarification on the Agnostic Nature of Compression in Clapping
>
> Thank you for this question, which allows us to clarify a fundamental design principle of our framework. The Clapping method is intentionally compression-agnostic; it provides a unified protocol for communication compression rather than proposing a new compressor.
>
> The specific mechanisms for compressing both forward activations and backward gradients are fully defined by the communication patterns in **Algorithms 2 and 3**, respectively. In these algorithms, the compression operation is abstractly represented by the compressor $\mathcal{C}$. This design signifies that the exact implementation of $\mathcal{C}$ is not specified because our theoretical and practical contributions are independent of it.
>
> The only requirement for $\mathcal{C}$ is that it must satisfy the contractive compression property defined in Assumption 3 (i.e., $\mathbb{E}\left[\left\Vert x−\mathcal{C}(x)\right\Vert^2\right]\leq\omega^2\left\Vert x\right\Vert^2$ for some $\omega\in[0,1)$. This is a standard and mild assumption in the literature on communication compression. Consequently, any compressor meeting this criterion can be seamlessly plugged into our framework. This includes widely-used compressors such as:
>
> * Sparsification methods: Top-K, Rand-K
> * Quantization methods: Block-wise quantization, 1-bit Lattice quantization (SignSGD)
> * Hybird methods: Natural Compression (NC)
>
> Therefore, the compression of activations and gradients in our experiments was achieved by instantiating $\mathcal{C}$ with various off-the-shelf compressors (as detailed in our experiments), validating the flexibility and generality of the Clapping framework.
>
> ## References
> [1] Fine-tuning language models over slow networks using activation quantization with guarantees
>
> ----
>
> We sincerely appreciate the reviewer’s thoughtful comments once again. We hope the above responses fully address the concerns raised and would be happy to provide further clarification if needed.

---

### Official Review · Reviewer_X9Xg · 2025-10-31

**Soundness:** 2
**Presentation:** 3
**Contribution:** 2
**Rating:** 4
**Confidence:** 4

**Summary:**

This paper introduces Clapping, a communication compression framework for pipeline-parallel distributed optimization. By integrating error feedback and lazy sampling, Clapping-FU—achieve a state-of-the-art convergence rate. This result holds without relying on the unbiased gradient assumption and avoids the sample-wise memory overhead present in existing algorithms.

**Strengths:**

- Clapping is the first communication compression algorithm for pipeline parallelism that achieves a convergence rate of $O(1/\sqrt{T})$ without requiring the unbiased gradient assumption.

- Experimental results demonstrate that Clapping-FC achieves significantly better performance compared to the baseline, AQ-SGD.

**Weaknesses:**

- Limited Applicability to LLM Training: Clapping relies heavily on the assumption that training is conducted over multiple epochs. This contrasts with modern LLM pre-training, which often uses only one or a few epochs, and fine-tuning, which typically uses around four epochs. Furthermore, training on sequentially ordered data often yields weaker performance than training on fully shuffled data. It is unclear whether the no-compression baseline experiments in this work used completely shuffled data, a standard practice for achieving optimal performance. Consequently, Clapping's applicability to LLM tasks may be limited.

- Questionable Practical Efficiency: Clapping-FC converges at a rate of $O(1/\sqrt[3]{T})$, which is significantly slower than the $O(1/\sqrt{T})$ rate of non-compressed SGD. While Clapping-FU restores the $O(1/\sqrt{T})$ rate, it does so by forgoing compression with a probability of $1-p$. This limits the maximum communication reduction to a factor of $p$, and the paper suggests $p \approx 0.5$ is needed to maintain performance. Additionally, Clapping requires extra memory to store previous activations. This memory could otherwise be used to increase the batch size, potentially accelerating total training time. In summary, the net reduction in overall training time achieved by Clapping-FU may be minimal compared to a no-compression baseline.

**Questions:**

- Convergence with $p=0$: According to Algorithm 1, setting $p=0$ implies repeatedly training on the same data sample, a scenario under which $\frac{1}{T}\sum_{t=1}^T{\mathbb{E}} \Vert \nabla {\cal l(w_t) }\Vert^2$ would not be expected to converge. However, Lemma 7 and Lemma 13 state that convergence at a rate of $O(1/\sqrt{T})$ is still guaranteed in this case. This appears counterintuitive and may indicate an issue in the theoretical derivation.

- Inconsistency with Vanilla SGD: When the momentum parameter is set to zero ($m=0$), the algorithm reduces to vanilla SGD. The convergence bound given in Lemma 7 and Lemma 13 for this case is $O(1/\sqrt{T}) + \sigma^2$, which does not align with the established theoretical result that vanilla SGD can converge to a stationary point without such an additive noise term. Could the authors clarify this discrepancy?

- Error Accumulation in Deep Pipelines: Theorem 1 suggests that compression errors in Clapping accumulate exponentially with the pipeline depth $K$. For models with large $K$ (e.g., modern LLMs), this property would be a significant practical drawback. Could the authors comment on the practical implications of this for scaling to very deep models?

---

> ### Author Response · Authors · 2025-11-26
> **Rebuttal by Authors (1/4)**
>
> We are grateful for the reviewer's insightful feedback and constructive comments. We address each point in detail below.
>
> ----
>
> ## 1. Clapping is Designed for Few-Epoch Training and Demonstrates Robustness in LLM Settings
>
> We thank the reviewer for raising this important point. However, we would like to clarify that Clapping does not inherently depend on multi-epoch training; rather, it was developed precisely to **eliminate the multi-epoch requirement** imposed by previous error-compensation methods (e.g., AQ-SGD). Our contribution statement explicitly targets few-epoch and single-epoch regimes, and both the theoretical framework and experimental validation are designed accordingly.
>
> **A. Theoretical Foundation: Convergence Guarantees without Multi-Epoch Assumptions.**
> A key motivation for Clapping is to address the limitation of prior works that rely on extensive epochs for error compensation—a requirement incompatible with modern LLM pre-training and fine-tuning (Limitation **L3** in Section 1.1). Clapping’s algorithm design and convergence analysis are independent of multi-epoch training. Specifically, Clapping-FU achieves an $\mathcal{O}(T^{-1/2})$ convergence rate, and Clapping-FC converges despite compression error accumulation, with both guarantees holding without the unbiased-gradient assumption or the need for data revisits over multiple epochs.
>
> **B. Empirical Validation: Effective Performance in Single- and Few-Epoch LLM Workloads.**
> Clapping demonstrates strong performance across LLM pre-training and fine-tuning tasks under few-epoch constraints:
> * In GPT-2 pre-training on OpenWebText, training proceeds for approximately one epoch (130,800 iterations, with total sample complexity nearly equivalent to one epoch). We explicitly note that AQ-SGD is inapplicable here due to its multi-epoch requirement, whereas Clapping trains stably and mitigates compression error, as reflected in loss and perplexity curves (Figure 3).
> * For LLaMA-2 1B pre-training on the C4-en dataset, both Clapping-FU and Clapping-FC match the uncompressed baseline’s final perplexity while achieving $\geq2.2\times$ end-to-end speedup (Figure 4), confirming efficacy without multi-epoch dependence.
> * In few-epoch fine-tuning scenarios (e.g., LLaMA-2, LLaMA-3, and GPT-2), Clapping consistently matches or surpasses uncompressed, EF21, and AQ-SGD baselines with substantial communication savings—for instance, achieving 95% communication reduction with less than 0.5% error (Table 3), with performance trends documented across 1–6 epochs (Figures 7–9).
>
> In summary, Clapping is not limited by multi-epoch training requirements; instead, it enables efficient compression in few-epoch and single-epoch regimes, as evidenced by theoretical guarantees and extensive experiments across LLM tasks.
>
>
> ## 2. Suffling Strategies in Experiments.
> Regarding data ordering and shuffling protocols, we confirm that all experiments, including Clapping and baseline methods, adhered to standard practices for fair comparison.
>
> Our training protocol follows the conventional i.i.d. sampling assumption: in pipeline-parallel SGD, each initial sample $y^{(t)}\_0:=x^{(t)}$ is drawn randomly from $\mathcal{D}$. This ensures full shuffling at the epoch level, consistent with standard practices for optimal performance. In experiments, Clapping employs lazy sampling with batch-wise shuffling (Algorithm 6), maintaining data randomness analogous to typical pre-training setups without compression. All baselines (including no compression, direct compression, and EF21) also use fully shuffled datasets to ensure fair comparison. Additional details are provided in the experimental subsection of Appendix I, with relevant passages highlighted for clarity.
>
> In summary, our use of standard shuffling protocols across all methods ensures that the reported performance gains of Clapping are attributable to the algorithm itself, rather than differences in data ordering.

---

> ### Author Response · Authors · 2025-11-26
> **Rebuttal by Authors (2/4)**
>
> ## 3. Practical Efficiency for Clapping
> We appreciate the reviewer's insightful comments regarding practical efficiency. Clapping is designed to address the fundamental challenge of enabling convergence under pipeline communication compression—a setting where direct compression methods typically fail. The framework offers two variants that represent different points on the theoretical-practical trade-off spectrum, with empirical results confirming significant end-to-end speedups in communication-bound scenarios.
>
> **A. Theoretical Contribution and Wall-clock Efficiency.**
>
> Our framework explicitly provides two variants targeting different theoretical guarantees. Clapping-FC achieves an asymptotic rate of $\mathcal{O}(T^{-1/3})$, while Clapping-FU restores the standard non-compressed SGD rate of $\mathcal{O}(T^{-1/2})$ without relying on the unbiased gradient assumption. These rates extend to Adam and large-batch training ($\mathcal{O}((BT)^{-1/3})$ for FC and $\mathcal{O}((BT)^{-1/2})$ for FU), as established in Theorem 2 and Appendices E/F, and remain valid in few-epoch regimes essential for modern LLM training.
>
> We emphasize that these results represent the first theoretical convergence guarantees for communication compression in pipeline-parallel distributed optimization that neither require the unrealistic unbiased-gradient assumption nor introduce per-sample memory overhead. While Clapping-FC's convergence rate does not match the non-compressed case, and Clapping-FU cannot achieve the same communication reduction as Clapping-FC, both variants effectively resolve the non-convergence issue inherent in pipeline communication compression—a critical advancement for large-scale optimization. Thus, the theoretical contribution of Clapping should be appropriately recognized, and we regard further convergence improvements as important future work.
>
> In terms of practical efficiency, Clapping delivers substantial wall-clock gains when communication represents the bottleneck. In our LLaMA-2 1B pre-training experiments, both Clapping-FU and Clapping-FC reached the target perplexity with $\geq2.2\times$ speedup over the no-compression baseline under the communication-bottlenecked scenario, while direct compression failed to converge (Fig. 4, Sec. 5), illustrating why reducing communication—even with partial compression—yields net training-time savings.
>
> **B. Memory Overhead and the Batch Size Trade-off.**
>
> Clapping requires $\mathcal{O}(B)$ additional memory to store current-batch activations and gradients for error feedback, which is independent of dataset size and contrasts sharply with the sample-wise caching in AQ-SGD. Our quantitative analysis (Table 5, Appendix H) shows that Clapping's memory overhead is modest relative to the base memory footprint (parameters, gradients, and optimizer states)—accounting for 3.1% (7B), 6.4% (13B), and 10.7% (70B) of the total—leaving ample room for standard micro-batches. In contrast, AQ-SGD would require terabytes of memory due to its per-sample cache requirements.

---

> ### Author Response · Authors · 2025-11-26
> **Rebuttal by Authors (3/4)**
>
> ## 4. Additional Discussion for the Convergence Result
>
> We appreciate the reviewer's careful reading regarding two specific boundary cases in our theoretical analysis. Below we clarify why these scenarios—while seemingly contradictory—are in fact consistent with both our framework's assumptions and established results in error-feedback methods. Below we address both points within a unified theoretical perspective.
>
> **A. Theoretical Convergence Under Constant Lazy-Sampling Probability.**
> - **Our convergence analysis assumes a constant, strictly positive lazy-sampling probability.**
>   The $\mathcal{O}(T^{-1/2})$ convergence rate stated in the main text—including Lemma 7 for the two-stage setting and Theorem 2 for the general multi-stage case—is established under the assumption that $p$ is a fixed positive constant, i.e., $p = p_0 = \mathcal{O}(1)$. The parameters $(\gamma, m)$ are subsequently chosen in accordance with this fixed $p$. For instance, Lemma 7 gives the bound $\frac1T\sum_{t=1}^T \mathbb{E}\|\|\nabla \ell(w^{(t)})\|\|^2 \lesssim\frac{\sigma}{\sqrt{T}}+\frac{1}{T(1-\omega_B)(1-\omega_F)},$ where $\gamma, m > 0$ are chosen as constants, and $p$ is fixed earlier in the proof. The same assumption applies to the multi-stage result in Lemma 13 for $E>2$.
>
> - **At $p=0$, the bound collapses and only convergence to a noise-dominated region is guaranteed.**
>   The variance term in our analysis, $\sigma^2 \cdot \dfrac{(2-p)m-(1-p)m^2}{1-(1-p)(1-m)^2}$ (see inequality (4) in the manuscript) becomes $\mathcal{O}(1)$ as $p \to 0$, which prevents the average gradient norm from converging to zero. Instead, the algorithm converges only to a neighborhood whose radius is governed by $\sigma^2$. This is fully consistent with the known behavior of SGD when repeatedly training on the same sample—convergence occurs only to a variance-dominated region, not to an exact critical point.
>
>   This behavior is also reflected in the large-batch derivation (e.g., Eq. (113)), where setting $p=0$ reduces the ratio to 1, leaving a residual term $4\sigma^2/B$ that does not decay with $T$. Therefore, the theoretical results remain sound and align with classical SGD intuition when data is not refreshed.
>
> **B. Clarification on Convergence Behavior without Momentum.**
> - **Setting $m=0$ does not recover vanilla SGD.**
>   Even with momentum parameter $m$ set to zero, Algorithm 4 continues to operate with **compressed activations and adjoints** as well as **error feedback** across pipeline stages. The resulting update relies on the proxy gradient $\tilde u_e^{(t)}=\nabla^2 a_e(\tilde y_{e-1}^{(t)},w_e^{(t)})^\top \tilde v_e^{(t)},$ where $(\tilde y,\tilde v)$ are generated through compression and error-feedback recursions (see Eq. (3)). Thus, the $m=0$ case in our framework represents **SGD with biased gradients due to compression and EF**, not the standard vanilla SGD.
>
> - **The residual term reflects EF bias without momentum.**
>   The presence of momentum (or a moving-average mechanism) is crucial for stabilizing error feedback methods. As shown in previous theoretical work [1], EF without momentum can fail to converge to a stationary point unless restrictive conditions are imposed. Our analysis aligns with this understanding: when $m=0$, compression-induced bias is not fully averaged out, leading to the observed $\sigma^2$ residual in our bounds. This phenomenon is not unique to pipeline parallelism but also occurs in data-parallel EF-SGD. The momentum term is explicitly introduced to "mitigate the gradient bias caused by inaccurate activation gradients" (see Section 2.2). This design choice is motivated by established results in the EF literature.
>
> - **Bound consistency under the uncompressed setting.**
>   Our theoretical framework does recover the standard $\mathcal{O}(\sigma/\sqrt{T})$ nonconvex rate (Theorem 2) when compression is disabled ($\omega_F=\omega_B=0$) and fresh data is sampled every step ($p=1$). In this special case, the algorithm indeed reduces to vanilla SGD and the corresponding bound holds without residual terms.

---

> ### Author Response · Authors · 2025-11-26
> **Rebuttal by Authors (4/4)**
>
> ## 5. Impact of the number of workers on performance
> We thank the reviewer for raising the question regarding the impact of the number of workers. To validate this finding, we fine-tuned a pre-trained LLaMA-2 13B model on Wikitext dataset for 4 epochs using Top-20% compressor and Clapping-FC with varying numbers of workers, while maintaining the compression probability at $p=0.5$. Specifically, we set $E=1,2,4,9$ workers in separate experiments. All other parameters were consistent with prior configurations, except for an increase in the initial learning rate to $5\times 10^{-5}$ to amplify the error accumulation effect. The following table (Table 7 in the revised manuscript) summarize the validation perplexity and validation accuracy across different worker counts. The results reveal that as the number of workers increases, the compression error accumulates, leading to a corresponding decline in validation performance. We also include this additional experiment in Appendix I.3.2 of the revised manuscript.
>
> **Table 1**: The validation perplexity ($\downarrow$) and accuracy ($\uparrow$) for the fine-tuning task for LLaMA-2 13B under Wikitext with Clapping-FC with different number of workers ($E$).
>
> | Number of workers ($E$) | Validation perplexity | Validation accuracy |
> | -------- | -------- | -------- |
> |1         | 6.1325       | 0.5823              |
> |2         | 6.7896 | 0.5654          |
> |4         | 7.9840 | 0.5422                   |
> |9         | 13.5134 | 0.4606 |
>
> ## References
> [1] Momentum Provably Improves Error Feedback!
>
> ----
>
> We sincerely appreciate the reviewer’s thoughtful comments once again. We hope the above responses fully address the concerns raised and would be happy to provide further clarification if needed.

---

### Official Review · Reviewer_1MYU · 2025-11-01

**Soundness:** 2
**Presentation:** 3
**Contribution:** 2
**Rating:** 4
**Confidence:** 3

**Summary:**

The paper proposes the clapping framework, which addresses the key barrier of sample-scale memory overhead(L2) in communication compression for pipeline-paralllel distributed optimizaiton via the Lazy Sampling strategy.  Moreover, the framework provides theoretical convergentce guarantees without the unrealistic unbiased gradient assumption (L1) and extends its analysis to multi-worker setups(L4).

**Strengths:**

The paper resolves core memory bottleneces via the lazy sampling strategy.
The paper makes theoretical progress by providing convergence guarantees for pipeline-parallle compression algorithms.
The paper provides compatibility with modern optimizers. The convergence analysis is extended to the Adam optimizer and large-batch scenarios.

**Weaknesses:**

1. The lazy sampling strategy  sacrifices model generalization.
2.  The experimental scale and validation is insufficient. The largest pre-trainded model in experiments is 1B and fine-tuning task cannot fully represent pre-training challenges.

**Questions:**

NONE

---

> ### Author Response · Authors · 2025-11-26
> **Rebuttal by Authors (1/2)**
>
> We are grateful for the reviewer's insightful feedback and constructive comments. We address each point in detail below.
>
> ----
>
> ## 1. Model generalization under lazy sampling
>
> We acknowledge the reviewer's concern regarding the potential impact of the lazy sampling strategy on model generalization. In Clapping, lazy sampling is employed to facilitate error-feedback in pipeline communication compression without requiring per-sample caches. Specifically, at each training step, the current sample is retained with probability $1−p$, and a new sample is drawn from the distribution $\mathcal{D}$ with probability $p$ (as detailed in Algorithm 1 and Section 3). This approach reuses samples over a short horizon to mitigate compression bias while preserving the data marginal distribution—it introduces temporal correlation without excluding any subset of the training data.
>
> **A. Theoretical Analysis: Trade-off Between Convergence and Generalization Governed by $p$.**
>
> Our convergence analysis explicitly addresses the trade-off controlled by the parameter $p$. As $p\to0$, the variance term $\sigma^2$ in the convergence bound fails to vanish, leading to non-convergence for both Clapping-FC and Clapping-FU. This stems from insufficient exploration of the data distribution, which is detrimental to generalization. Conversely, in the regime where $p\to1$, the model accesses a nearly independent sample at each step. However, Clapping-FC may then suffer from non-convergence due to uncompensated compression bias, while Clapping-FU effectively reduces to vanilla SGD without compression benefits. Therefore, an intermediate value of pp is critical to balance these aspects. In this regime, both variants enjoy non-asymptotic convergence guarantees ($\mathcal{O}(T^{-1/3})$ and $\mathcal{O}(T^{-1/2})$ for Clapping-FU) in pipeline settings without requiring the unbiased-gradient assumption. Consequently, selecting pp within an intermediate range ensures stable optimization and allows the model to learn the data distribution more comprehensively, thereby preserving generalization while maintaining significant communication savings.
>
> **B. Empirical Support: Generalization Maintained or Enhanced Across Tasks and Model Scales.**
>
> Experimental results across image and language tasks indicate that lazy sampling combined with error-feedback matches or surpasses uncompressed or directly compressed baselines in evaluation metrics:
> * On CIFAR-10 (ResNet-18 with quantization compressors), lazy sampling improves test accuracy over direct compression; for instance, in the f6-b6 setting, accuracy reaches 0.9186 with $p=0.6$ compared to 0.9025 (direct compression). Consistent gains are observed across configurations (Table 6).
> * For GLUE benchmarks (RoBERTa-large with Top-K 30% and $p=0.5$), Clapping-FC achieves the highest average score (87.79) versus uncompressed (87.61) and EF21 (86.61), exceeding the uncompressed baseline on MRPC and RTE.
> * In LLaMA-2 fine-tuning (Wikitext-2 with Top-K 5% and natural compression), appropriate pp values yield evaluation accuracy and perplexity comparable to no compression; by tuning $p$, we achieve 95% communication reduction with ≤0.5% error on practical tasks.
> * For LLaMA-2 1B pre-training, employing a cosine schedule for $p$ (from 0.5 to 1) allows Clapping-FU/FC to match the validation perplexity of the uncompressed baseline while accelerating training by approximately $2.2\times$ (Figure 4). These results confirm that controlled $p$ preserves model expressivity and accelerates training.
>
> **C. Mechanistic Explanation: Error-Feedback and Regularization Sustain Generalization.**
>
> Lazy sampling is integrated with error-feedback on both forward and backward passes, which incrementally reduces compression bias without storing per-sample activations or gradients. This combination not only stabilizes optimization under compression but also acts as a mild regularizer. As discussed in Appendix I.4.2, the noise introduced by compression may further help prevent overfitting, accounting for the observed parity or improvements in evaluation metrics.
>
> In summary, our theoretical and empirical analyses explicitly address the trade-offs associated with $p$. Within the recommended regime (e.g., $p\in[0.3,0.5]$), Clapping does not compromise generalization; instead, it achieves comparable or superior accuracy and perplexity across diverse models and datasets while significantly reducing communication overhead.

---

> ### Author Response · Authors · 2025-11-26
> **Rebuttal by Authors (2/2)**
>
> ## 2. Emprical results in a larger scale
>
> To validate our proposed algorithm on a larger-scale pre-training task, we conduct an additional experiment using LLaMA-2 3B. In this setup, we constrain the bandwidth to 50 MB/s and employ the same compressor configuration as used in the LLaMA-2 1B pre-training experiments. The compressor remains disabled during the initial 8,000 steps, and due to time constraints, we train the model for 50,000 steps total. All other experimental settings remain consistent with the LLaMA-2 1B baseline. The following table compares the total time required by both the non-compression baseline and our Clapping-FC method to achieve the same target validation perplexity, demonstrating a significant improvement in efficiency with our proposed approach. Complete experimental results are provided in Appendix I.4.2.
>
> **Table 1**: Comparison of training time ($\times 10^5$ s) to reach target validation perplexity for LLaMA-2 3B pre-training under bandwidth constraint
> | Target validation perplexity | 18.0 | 17.5 | 17.0 |
> | -------- | -------- | -------- | -------- |
> |No compression         | 2.6933  | 2.34036  |  2.69330       |
> |Clapping-FC         | 2.42791 | 2.08600 |  2.42791    |
>
> We fine-tuned a pre-trained LLaMA-2 13B model on Wikitext dataset for 4 epochs using Top-20% compressor and Clapping-FC with varying numbers of workers, while maintaining the compression probability at $p=0.5$. Specifically, we set $E=1,2,4,9$ workers in separate experiments. All other parameters were consistent with prior configurations, except for an increase in the initial learning rate to $5\times 10^{-5}$ to amplify the error accumulation effect. The following table (Table 7 in the revised manuscript) summarize the validation perplexity and validation accuracy across different worker counts. We also include this additional experiment in Appendix I.3.2 of the revised manuscript.
>
> **Table 2**: The validation perplexity ($\downarrow$) and accuracy ($\uparrow$) for the fine-tuning task for LLaMA-2 13B under Wikitext with Clapping-FC with different number of workers ($E$).
>
> | Number of workers ($E$) | Validation perplexity | Validation accuracy |
> | -------- | -------- | -------- |
> |1         | 6.1325       | 0.5823              |
> |2         | 6.7896 | 0.5654          |
> |4         | 7.9840 | 0.5422                   |
> |9         | 13.5134 | 0.4606 |
>
> ----
>
> We sincerely appreciate the reviewer’s thoughtful comments once again. We hope the above responses fully address the concerns raised and would be happy to provide further clarification if needed.

---

### Author Response · Authors · 2025-11-26

Dear reviewers,

We have carefully considered all your comments and questions, and have addressed them one by one. The corresponding revisions have been incorporated into the main text. If you have any further questions, please do not hesitate to contact us. We sincerely appreciate your time and insightful comments.

Best,

The authors of paper 3569

---

### Meta-Review · Area_Chair_Z6BL · 2026-01-07

**Summary:**

This paper proposes a pipeline-parallel communication compression framework that avoids (i) the unbiased gradient assumption, (ii) sample-wise memory overhead (as in AQ-SGD), and (iii) reliance on multi-epoch training. The core idea is lazy sampling (reusing a sample for a few steps with probability (1−p) combined with error feedback in both forward and backward passes, yielding two algorithm variants.

**Reviewer Concerns:**

Reviewer 1MYU

(1) Lazy sampling may hurt generalization -- Authors address this by explicitly analyzing p as a bias-variance tradeoff parameter, showing convergence fails p goes to 0. Also provide CIFAR-10, GLUE, LLaMA fine-tuning and pre-training results showing no degradation (and sometimes improvement).
(2) Experimental scale insufficient (largest pretrain 1B; fine-tuning ≠ pre-training). -- Authors add LLaMA-2 3B pre-training and 13B fine-tuning experiments.

Reviewer X9Xg

(1) Applicability to few-epoch / shuffled LLM training unclear. -- Authors correctly point out Clapping was designed exactly to remove AQ-SGD’s multi-epoch requirement, and back this with GPT-2 (≈1 epoch), LLaMA-2 pre-training, and fine-tuning experiments.

(2) Practical efficiency questionable (Clapping-FC slower rate while  Clapping-FU limited compression). Authors concede, and do not claim universal wins—only communication-bound wins.

(3) Confusing convergence claims at p=0. Authors carefully explain that p=0 leads to variance-dominated convergence, consistent with SGD on a fixed sample.

(4) Inconsistency for Momentum=0 gives vanilla SGD? -- Addressed correctly by authors

(5) concern about error accumulation exponential in pipeline depth. Concession by authors, this is a real weakness.


Reviewer RDoY
(1) Lazy sampling may distort label distribution -- Authors provide empirical evidence against this. Arugments still handwavy, this could happen in other settings/datasets.

(2) Forward-activation compression unjustified -- Strong theoretical rebuttal

(3) “Only 7B/8B models tested” -- Clarified, reviewer was indeed mistaken.

(4) Clarity issues on how compression is implemented -- Addressed

Reviewer HM6P

(1) Concerns about wall-clock spped -- partially addressed by authors provide some plots but concede that gains appear only in comm-bound regimes.
(2) Unrealistic bandwidth assumptions -- Authors concede diminishing returns on larger bandwidths.
(3) No batch-size / device-map study -- partially addressed, authors explain why memory savings don’t easily translate to larger batch sizes and why device count worsens error accumulation, no ablation studies provided.
(4) Limited baselines
Reviewer was unconvinced, no universally convincing system-level benchmarking.

**Reviewer Scores:**

Unlilkely to change beyond what they have already specified. Two reviewers remain unconvinced that the practical gains are robust outside synthetic or bandwidth-starved scenarios. The paper does not fully clear the bar -- The contribution is limited with some theory-plus-controlled-experiments.

---

### Decision · Program_Chairs · 2026-01-26

Reject